# The Greenhouse gas Emission Monitoring network to Inform Net-zero Initiatives UK (GEMINI-UK): network design, theoretical performance, and initial data

Alexander Kurganskiy[1,2], Liang Feng[1,2], Neil Humpage[3,4], Paul I. Palmer[1,2], A. Jerome P. Woodwark[5,2], Stamatia Doniki[6], and Damien Weidmann[6]

[1]National Centre for Earth Observation, University of Edinburgh, Edinburgh, UK
[2]School of GeoSciences, University of Edinburgh, Edinburgh, UK
[3]National Centre for Earth Observation, University of Leicester, Leicester, UK
[4]School of Physics and Astronomy, University of Leicester, Leicester, UK
[5]Karn Scientific Ltd, Edinburgh, EH6 4LB
[6]Space Science and Technology Department, STFC Rutherford Appleton Laboratory, Didcot, UK

**Correspondence:** Alexander Kurganskiy (alexander.kurganskiy@ed.ac.uk), Neil Humpage (nh58@leicester.ac.uk), Paul I. Palmer (paul.palmer@ed.ac.uk), and A. Jerome P. Woodwark (jerome.woodwark@karnscientific.com)

**Abstract.** The Greenhouse gas Emissions Monitoring network to Inform Net-zero Initiatives for the UK (GEMINI-UK) includes ten Bruker EM27/SUN instruments located across the UK that collect dry average volume mixing ratios of $CO_2$ and methane ($XCO_2$ and $XCH_4$). The primary objective of GEMINI-UK is to infer regional net flux estimates of $CO_2$ and methane across the UK that can be used to provide actionable information to the UK Government. The instruments are housed in bespoke autonomous weatherproof enclosures that help maximize cloud-free data collection throughout the calendar year. The network will become fully operational in late 2025. As part of our commissioning phase, we designed the network so it would deliver the biggest uncertainty reduction in net $CO_2$ fluxes, based on prior emission inventories. The ten sites are located at UK education institutions and a national scientific research laboratory, underlining our commitment to make these data openly available to all. In this study, we use a series of closed-loop numerical experiments for the nominal calendar year of 2019 to quantify the theoretical benefit of using this new ground-based remote sensing network, accounting for cloudy scenes, to estimate spatially resolved net fluxes of $CO_2$ and methane across the UK. Based on our results, we expect that GEMINI-UK will deliver significant error reductions in $CO_2$ flux estimates, up to 51% in January and up to 59% in July (GEMINI-UK only), and up to 8% in January and up to 24% in July when combined with existing tall tower measurements collected across England and Ireland. Despite the network being optimally designed to enhance our understanding of UK $CO_2$ fluxes, we expect, based on our calculations, that GEMINI-UK will also substantially reduce uncertaintes of methane emissions, achieving *a priori* error reductions up to 55% in January and up to 75% in July (GEMINI-UK only), and up to 10% in January and up to 29% in July when combined with the existing tall tower sites. In the context of augmenting the information collected by the established tall tower network, we find that GEMINI-UK data have the greatest potential over high flux regions in the central and southern parts of the UK during winter months, and over broader southern to northern regions during the summer months. More broadly,

the data collected by GEMINI-UK will also provide the basis to evaluate satellite observations of these trace gases, thereby providing confidence in their ability to supplement data collected by GEMINI-UK and the tall tower network.

*Copyright statement.*  TEXT

## 1  Introduction

The UK Government reported emissions of 406.2 Tg of $CO_2$-equivalent for the UK in 2022, a metric that allows different
greenhouse gases (GHGs) to be combined by taking into account their global warming potential. This number represents mostly $CO_2$ (80%), with contributions from methane (14%), nitrous oxide (4%), and fluorinated gases (2%) (UK Government, 2024). The importance of emissions from individual sectors changes with the gas but because $CO_2$ represents the bulk of the total $CO_2$-equivalent emission, the UK sectors that dominate are those that emit the most $CO_2$. The largest $CO_2$-equivalent emitting sectors include domestic transport, commercial and domestic energy consumption, agriculture, and waste. Some of
these emissions are focused on cities and towns (hotspots) across the UK, but others are more peri-urban (e.g., waste) or rural (livestock and crop agriculture) and consequently more diffuse. This diversity in sources, not unique to the UK, brings its own measurement challenge in terms of accuracy and precision.

The Deriving Emissions linked to Climate Change (DECC) network (Figure 1) was established in 2012, initially funded by the UK Government, and expanded as part of the NERC Greenhouse gAs UK and Global Emissions (GAUGE) project
(Palmer et al., 2018; Stanley et al., 2018). The DECC network mainly uses telecommunication masts to collect air at various inlet heights – typically between 50 and 200 metres – that have a geographical footprint that depends on wind speed and direction. The current network of five sites is supported by UK Government funding and coordinated by the University of Bristol, with measurements of $CO_2$ methane, nitrous oxide, sulfur hexafluoride and a suite of halocarbons. These sites were originally located to enable quantification of GHG emissions from the devolved administrations, i.e. England, Northern Ireland,
Scotland, and Wales, on annual timescales. In February 2024, the DECC network was extended with the addition of a tall tower site at Jodrell Bank, which began measuring GHG concentrations at a 50 m inlet height and provides coverage over northwest England. The DECC network will be further expanded in 2025 with a new Scottish-run tower hosted by the James Hutton Institute at Invergowrie, providing additional coverage for Scotland. In addition to the tall towers in the DECC network, the Mace Head Atmospheric Research Station, located on the west coast of Ireland and operated by the National University of
Ireland, Galway, has played a crucial role in quantifying UK greenhouse gas (GHG) fluxes for decades, e.g., Manning et al. (2011); Lunt et al. (2021a). Its coastal position, with prevailing westerly winds from the Atlantic, makes it ideal for monitoring background concentrations of GHGs, helping to differentiate between local emissions and regional background levels.

Bayesian inverse methods are used to relate changes in atmospheric $CO_2$ and methane to update *a priori* knowledge of the corresponding net fluxes and emissions (White et al., 2019; Lunt et al., 2021a; Jones et al., 2021; Deng et al., 2022; Worden
et al., 2022; Byrne et al., 2023; Forstmaier et al., 2023). These methods are used widely in the community but with some

caveats. Success of this approach relies on accurate assessments of the uncertainties associated with the *a priori* knowledge and the measurements. Random errors associated with the atmospheric transport model are difficult to quantify (Simmonds et al., 2021) so typically we are left to guesstimate them; systematic errors are typically ignored. The spatial and temporal resolution of the resulting *a posteriori* estimates also relies on the distribution of the measurements so that a sparse measurement network would result in independent estimates representative of larger geographical regions over longer time periods than a denser network. One of the outstanding challenges associated with this top-down atmospheric inversion approach is developing robust independent estimates of sector-based emissions. For example, changes in atmospheric $CO_2$ represent a superposition of aged and fresh air masses with signatures from regional uptake and emission. Without additional *a priori* information it is extremely difficult to separate natural fluxes and combustion emissions. If these competing contributions to atmospheric $CO_2$ are geographically distinct then this separation may be possible but generally this is not the case. Recent studies have begun to use trace gases and their isotopologues that are co-emitted with $CO_2$ during combustion or associated with the combustion process (Wang et al., 2018; Basu et al., 2020; Pickers et al., 2022; Scarpelli et al., 2024; C. Schooling, 2024) – while this represents a promising approach, illustrated by some recent methodological advances, it is non-trivial and introduces errors associated with atmospheric chemistry.

When the DECC network was established in 2012, it was the only national GHG measurement system designed to provide data suitable for annual reporting to the United Nations Framework Convention on Climate Change (UNFCCC). Since then, the DECC network has been crucial in supporting the UK's reporting obligations, contributing valuable data to inform national GHG inventories. However, there has been increasing recognition of the need to expand the UK measurement system to enhance emission reporting and support policy makers. For example, we have not yet incorporated data from Earth-orbiting satellite missions. Data from the NASA Orbiting Carbon Observatory (OCO-2, Crisp et al. (2017)), the Japanese Greenhouse gases Observing SATellite (GOSAT; Kuze et al. (2016)), and the European TROPOspheric Monitoring Instrument (TROPOMI; Lorente et al. (2021)) have proven effective in tracking $CO_2$ and methane emissions from large-scale sources, e.g., Nassar et al. (2021); Lauvaux et al. (2022). Future satellite missions, such as ESA's Copernicus $CO_2$ monitoring mission (CO2M), are expected to further advance our ability to monitor anthropogenic emissions, complementing the data collected by ground-based networks. Nevertheless, the UK represents a challenging measurement environment for these satellites – it is geographically small and cloudy – and even with CO2M we may never be able to rely exclusively on satellite data to deliver the data needed to report reliable estimates of UK GHG emissions. A robust and reliable GHG measurement framework for the UK will need to integrate information from a diversity of sensor technologies that ensures we maximise spatial and temporal coverage.

Funded as part of the Greenhouse Gas Emissions Measurement and Modelling Advancement (GEMMA) programme by the UKRI (UK Research and Innovation) Building a Green Future theme, an initiative aimed at fostering research to support sustainability and climate goals, we are establishing a ground-based network of ten Bruker EM27/SUN Fourier Transform spectrometers that will collect dry average volume mixing ratios of $CO_2$ and methane ($XCO_2$ and $XCH_4$) across the UK. These quantities, retrieved from the spectroscopic data collected by the instruments, will be used to infer regional carbon budgets, complementing data collected by the DECC tall tower network. The network of EM27/SUN spectrometers forms the Greenhouse gas Emissions Monitoring network to Inform Net-zero Initiatives for the UK (GEMINI-UK) that will deliver

data from early 2025 in time to report on the fourth carbon budget (2023–2027) and beyond, and represents the next phase of the UK-wide GHG measurement programme. Here we describe the design of the GEMINI-UK concept, show using focused closed-loop numerical experiments the potential benefits of GEMINI-UK to estimate spatially resolved net fluxes of $CO_2$ and methane across the UK over and above the information collected by the tall towers, and report some initial comparisons between the UK Total Carbon Column Observing Network (TCCON, Wunch et al. (2011); Weidmann et al. (2025)) sites based at the Rutherford Appleton Laboratory in Oxfordshire and the colocated GEMINI-UK spectrometer. The closed-loop numerical experiments involve generating and analyzing simulated data using the same model setup to assess the system's theoretical performance.

In Section 2, we briefly describe the EM27/SUN spectrometer and the purpose-built weatherproof enclosure that together forms the basis of GEMINI-UK measurements; the existing ground-based *in situ* measurement networks in the UK; the atmospheric transport model and the associated inverse method that we use to study GEMINI-UK measurements and transform them into regional estimates of $CO_2$ fluxes and methane emissions; the method we use for network design; and a description of a series of closed-loop experiments that assess the theoretical performance of GEMINI-UK to quantify regional $CO_2$ fluxes and methane emissions across the UK. In Section 3, we report the results from those closed-loop experiments and how the performance of GEMINI-UK compares with the existing ground-based network, and present an initial comparison between the UK TCCON instrument in Oxfordshire and the co-located GEMINI-UK instrument. We conclude the paper in Section 4.

## 2 Data and Methods

In this section, we introduce the ground-based remote sensing instruments that observe near-infrared (NIR) and shortwave infrared (SWIR) spectra and the algorithm that is used to infer $CO_2$ and methane columns from those spectra. We also introduce the bespoke weatherproof enclosure for the instrument, which allows us to operate autonomously the resulting GEMINI-UK network of instruments across the UK. Additionally, we provide an overview of the ground-based *in situ* data collected across the UK and mainland Europe that complement information being collected by the ground-based remote sensing instruments. We also detail the GEOS-Chem atmospheric chemistry and transport model and the associated ensemble Kalman filter (EnKF) inverse method that describes how we infer $CO_2$ and methane fluxes from atmospheric data. We illustrate how we use these analyses tools to design an optimal network of measurements that results in the largest reduction in uncertainty of $CO_2$ and methane fluxes across the UK. Finally, we present the closed-loop numerical experiments we use to showcase the theoretical potential of the GEMINI-UK network. Our theoretical calculations are focused on the contrasting months of January and July during 2019.

### 2.1 EM27/SUN Ground-based Remote Sensing Instruments

For the GEMINI-UK network, we use Bruker EM27/SUN FTIR (Fourier Transform InfraRed) spectrometers (Gisi et al., 2012; Hase et al., 2016). We have chosen this instrument for GEMINI-UK because it was designed to provide a portable, relatively low-cost means of accurately measuring total column concentrations of greenhouse gases from the ground and it has been well

established for consistent operation within a network (Frey et al., 2019). The instrument achieves this by measuring moderate-resolution SWIR spectra of direct sunlight, with a spectral resolution of 0.5 cm$^{-1}$, using an automatic solar tracker connected
to the spectrometer. Column concentrations of $CO_2$, methane, carbon monoxide (CO), and water vapour, are then inferred from the spectral absorption of sunlight as it passes through the atmosphere.

We use the PROFFAST retrieval code (Sha et al., 2020; Frey et al., 2021; Hase, 2023), developed within the COllaborative Carbon Column Observing Network (COCCON; Frey et al. (2019); Alberti et al. (2022)), to infer the column quantities from the observed SWIR spectra. PROFFAST is a non-linear least-squares fitting algorithm, which works by scaling atmospheric
*a priori* profiles until the difference between the measured and forward-modelled spectrum is minimised. It uses a look-up table, created from HITRAN spectroscopic line lists (Gordon et al., 2022), to generate absorption cross-sections for the radiatively relevant molecules considered in the forward model. The spectra are generated from the measured interferograms using a tool called PREPROCESS, which applies a DC correction, phase correction, and a number of quality control tests before applying a fast Fourier transform to produce the spectra ready for analysis using PROFFAST. We run the PROFFAST suite of software
using PROFFASTpylot (Feld et al., 2024), a Python interface which allows for efficient, consistent processing across a network of instruments, and takes advantage of parallel processing capability where available.

Based on a long-term intercomparison of column data determined from an EM27/SUN spectrometer and a co-located reference high-resolution FTIR used for the TCCON network (IFS 125HR, Wunch et al. (2011)), the EM27/SUN was shown to demonstrate highly stable instrument characteristics on timescales of several years (Frey et al., 2019). The standard deviation
in GHG column concentrations between an ensemble of 30 EM27/SUNs tested alongside the reference instrument at the Karlsruhe Institute of Technology (KIT) between 2014 and 2018 was found to be 0.13 ppm for $XCO_2$, and 0.6 ppb for $XCH_4$. The stability and precision of the EM27/SUN instrument has also been tested over a wider geographical extent through side-by-side comparisons with the Bruker IFS 125HR instruments (Frey et al., 2015; Hedelius et al., 2016; Hase et al., 2016; Sha et al., 2020; Alberti et al., 2022), used worldwide by TCCON (Wunch et al., 2011)).
To ensure inter-comparability between the numerous EM27/SUNs operated by research groups around the world, the instrument line shape parameters for each instrument are obtained through a standard calibration procedure at KIT, where the instruments are also operated side-by-side with a reference EM27/SUN located at the Karlsruhe Institute of Technology to obtain instrument-specific scaling factors for each measured gas. The derived scaling factors are applied at the post-processing stage to their retrieved GHG column data, to enable comparability with all other EM27/SUN measurements which have been
processed in the same way using the software tools developed through the COCCON project (Frey et al., 2019; Alberti et al., 2022). The EM27/SUNs we use here include a second detector channel allowing measurement of the column concentration of carbon monoxide (Hase et al., 2016), which provides useful information to help characterise emissions sources connected to observed carbon dioxide column enhancements, e.g., Wunch et al. (2009); Silva et al. (2013); Che et al. (2022); Shan et al. (2022)).
To ensure data collected as part of GEMINI-UK are intercomparable with similar instruments worldwide, including those in the COCCON network, we link them indirectly to the relevant World Meteorological Organisation (WMO) measurement scales. Linking to these scales requires *in situ* measurements of the vertical atmospheric profiles of $CO_2$ and methane above

measurement sites, performed using airborne air-sampling instrumentation which has been calibrated to WMO standards. Previous studies (Wunch et al., 2010; Messerschmidt et al., 2011) have performed this calibration of ground-based column

measurements of $CO_2$ and methane on a number of TCCON stations worldwide. Additional uncertainties are introduced by assumptions made around the profile concentrations beyond the vertical range of the *in situ* airborne measurements. These studies demonstrated that, within these and other characterised uncertainties related to the measurements, a single global calibration factor can be used for each gas to tie TCCON total column data to the WMO scale.

Column measurements of CO provide an indication of incomplete combustion that will initially be used to help interpret

observed changes in $CO_2$ and methane across the UK, e.g., Sadiq et al. (2021). They will eventually be used more formally within the EnKF inverse method to help determine combustion sources of $CO_2$ (Super et al., 2024; Scarpelli et al., 2024) and methane so provides a way to evolve GEMINI-UK in due course.

GEMINI-UK has been designed intentionally to run autonomously so that we can maximize the number of clear-sky measurements throughout the year. This is enabled by software that allows us to check remotely on instrument performance and

data acquisition and by bespoke weatherproof enclosures that ensure the instruments can run throughout the calendar year with minimal human intervention. These are described in Appendix A.

## 2.2   Other Relevant UK and European Measurement Networks

In this study, we perform theoretical error reduction calculations using simulated GEMINI-UK and DECC observations. Real observational data are not assimilated in the results presented here but will be used in future inversion experiments. For these

theoretical calculations, we also consider continuous *in situ* concentration measurements of $CO_2$ and methane collected at a fixed elevation as part of the UK DECC network (Stanley et al., 2018), which currently includes five sites. These data have been used to produce data-driven UK estimates that supplement inventory estimates reported annually to the UNFCCC. To simulate these observations, we use the mean values of the five lowest model levels (from the GEOS-Chem 3-D Model of Atmospheric Chemistry and Transport, described in the next section), as the station heights (ranging from 56 m to 380 m above sea level)

and the top inlet heights (ranging from 45 m to 248 m above ground level) fall within the altitude range of these levels. We also include the surface measurement site at Mace Head, western Ireland (a few metres above local terrain), and new tall towers at Jodrell Bank in northwest England and at Invergowrie in east Scotland. For future inverse calculations using real data, we will consider only tall tower measurements collected at the highest inlet heights, typically 90-248 m above ground, during local hours of 10:00-17:00 to avoid the influence of the nocturnal boundary layer, when measurements may be skewed or localized

due to thinner and more stratified conditions.

When we analyse real data, we anticipate also using $CO_2$ and methane data collected across mainland Europe as part of the Integrated Carbon Observing System (ICOS; Heiskanen et al. (2022)) to provide lateral boundary conditions for the UK. The ICOS sites are shown in Figure 1b for illustrative purposes, and their data will be used in the inversion together with the GEMINI-UK observations once GEMINI-UK becomes fully operational in late 2025. The current ICOS network comprises

170 sites in 16 European countries.

## 2.3 GEOS-Chem 3-D Model of Atmospheric Chemistry and Transport

We use version 14.3.1 of the 3-D GEOS-Chem atmospheric chemistry transport model to describe changes in atmospheric $CO_2$ and methane. We run the model at a horizontal resolution of $0.25 \times 0.3125°$ for a regional European domain (-15 to 35°E and 34 to 66°N), as shown in Figure 1b, with 47 vertical levels ranging from the surface to 0.01 hPa, described by a hybrid-sigma coordinate system. This nested simulation is driven by boundary conditions from three independent sources, described below. Meteorological and surface fields are provided by GEOS FP reanalysis fields from the NASA Global Modeling and Assimilation Office (GMAO) at NASA Goddard.

Our $CO_2$ and methane model simulations include anthropogenic emissions from all 12 sectors provided in the TNO inventory (Super et al., 2020; Kuenen et al., 2022). For $CO_2$, these are grouped into nine categories, following Scarpelli et al. (2024): public power, industry, road and off-road transport, shipping, aviation, fugitive emissions, 'other' combustion, and 'non-combustion'. For methane, we include the same 12 sectors without grouping, and thus sources such as waste, solvents, livestock, and other agricultural sources are used explicitly. However, only the total emissions are estimated in our error reduction calculations; sector-specific fluxes are not resolved. We also include biomass burning emissions of $CO_2$ and methane from the GFAS v1.2 inventory (Kaiser et al., 2012) and the Global Fire Emissions Database (GFED) version 4.1 (Giglio et al., 2013), respectively. To describe the land biosphere exchange of $CO_2$, we use hourly fluxes of gross primary production (GPP) and respiration (RESP) taken from a pan-European simulation of the VPRM model (Gerbig, 2021). Ocean fluxes of $CO_2$ are taken from Mercator Ocean's NEMO PISCES model (Lefèvre et al., 2020). Oceanic and coastal methane fluxes are not included in our simulations. While such fluxes exist (e.g., Weber et al. (2019)), their contribution to atmospheric methane over the European and UK domain is relatively small compared with the terrestrial and anthropogenic sources considered here.

We include the lateral exchange of carbon, associated with the removal of crops from production sites to consumption sites, as described by Deng et al. (2022). We use wetland emissions of methane from v1.0 of the JPL WetCHARTs inventory (Bloom et al., 2017). We also include minor European methane sources from geological seeps (Etiope et al., 2019) and termites, taken from the Copernicus Atmospheric Monitoring Service (CAMS) dataset (Doubalova, 2018). To describe the main methane loss process, we use monthly pre-computed three-dimensional fields of the hydroxyl radical, consistent with observed values for the lifetime of methyl chloroform, from the GEOS-Chem HOx-NOx-Ox chemistry simulation (Wecht et al., 2014). We also include a minor soil sink of methane based on output from the MeMo model (Murguia-Flores et al., 2018).

We spin up the nested model using lateral boundary conditions from the equivalentglobal version of the model run to form our forward baseline calculations, which represent the prior model state before applying perturbations to *a priori* emissions and before data assimilation. The global model is run at a horizontal resolution of $2° \times 2.5°$ and have been fitted to satellite observations and surface mole fraction observations of column methane and $CO_2$ using an EnKF (Feng et al., 2017, 2023). To understand the sensitivity of our results to assumed lateral boundary conditions, we used two alternate datasets. The first is taken from vCAMS-73 of the CAMS dataset that is available every three hours on a horizontal resolution of $1.9° \times 3.75°$ and every six hours on a horizontal resolution of $2° \times 3°$ for $CO_2$ and methane, respectively. This model is fitted to surface mole fraction observations of $CO_2$ and methane (CAMS, 2020). The second alternate set of boundary conditions is taken from the

220 CAMS EGG4 model that makes additional use of satellite column observations of $CO_2$ and methane (Agustí-Panareda et al., 2023). This model output is available every three hours on a horizontal resolution of $0.75° \times 0.75°$.

We report an evaluation of this model configuration using real data collected by the DECC network for which we sample the data at the time and location of each observation. For *in situ* $CO_2$ and methane mole fraction data, we use a one hour averaging time for model and observed time series. We report the comparison between model and measurement $CO_2$ and methane mole

fraction data in Figures 2 and 3, respectively, which is described later. We then filter the time series so that we only consider data collected at local times of 10:00–17:00 to avoid instances when the nighttime boundary layer is below the height of the highest inlet, and during well-mixed atmospheric conditions. We consider conditions to be well-mixed when the standard deviation of concentrations across the lowest five model vertical layers – approximately lowest 600 metres – is less than 5 ppm for $CO_2$ and less than 25 ppb for methane. These threshold values were chosen based on expert judgement, guided by an analysis of the

observed time series and the number of data points retained under different candidate thresholds.

### 2.4 Local Ensemble Transform Kalman Filter Inverse Method

We employ a variant of the EnKF to demonstrate the potential benefits of the GEMINI-UK network to infer regional flux estimates of UK emissions of $CO_2$ and methane over and above the information provided by existing *in situ* measurement networks. Even though we do not report flux estimates inferred from our synthetic data in this study, we use some of the same

numerical machinery to design the GEMINI-UK network and to determine flux uncertainty reductions when we consider the GEMINI-UK data.

We use the Local Ensemble Transform Kalman Filter (LETKF; Hunt et al. (2007); Liu et al. (2016)), which uses a comparatively small ensemble of perturbations to represent the *a priori* error covariance and localises observation constraints to suppress adverse effects from any resulting artificial long-distance correlations. This approach is generally considered to be

more computationally efficient than other inverse methods such as 4D-var (Chevallier et al., 2010) or the conventional Ensemble Kalman Filter (Feng et al., 2009). As such, the LETKF has been widely used to infer $CO_2$ flux estimates (e.g., Scarpelli et al. (2024)) and methane emission estimates (e.g., Lunt et al. (2021b)).

Details of our LETKF framework are described in Scarpelli et al. (2024). For brevity, here we outline only the specifics of the GEMINI-UK $CO_2$ inverse problem. First, we construct an ensemble ($n$=100) of the *a priori* $CO_2$ emissions and sinks (as

described above) with random perturbations to represent the assumed $CO_2$ *a priori* (background, $b$) error covariance matrix which we assume to have a uniform value of 50% ($\sigma$=0.5) across all model grid boxes. We also assume an error correlation length of 100 km. We also estimate the four lateral boundary conditions by applying a relative perturbation, using a distribution with a mean of 1.0 and a standard deviation of 0.05. The perturbation ensemble of the state vector $\mathbf{\Delta x}^b$ is defined as:

$$\mathbf{\Delta x}^b = \mathbf{x}^b - \overline{\mathbf{x}}^b, \tag{1}$$

where $\overline{\mathbf{x}}^b$ is the mean state of the emission ensemble. Here and elsewhere we adopt the convention that enboldened variables in upper and lower case roman denote vector and matrix quantities, respectively.

We use a nested version of the GEOS-Chem model, described above, to simulate the atmospheric transport of the emitted gases for each of the ($n$=100) perturbed sources across Europe. We also solve for the four lateral boundary conditions of our nested model every 15-day assimilation window (see below). We then sample the resulting 4-D model atmospheric concentrations to get the vertical compostion profile at the time and location of each observation. To compare the model with the tall tower measurements (DECC and ICOS), we use the mean values of the lowest five model layers to represent the mean planetary boundary layer value. To compare the model with dry-air $CO_2$ and methane columns (XCO$_2$ and XCH$_4$) retrieved from EM27/SUN instruments, we calculate the model column that is convolved with the instrument averaging kernel that describes the instrument vertical sensitivity to changes in the two gases. In practice, these kernels are scene- and time-dependent but for the purposes of our theoretical calculations, we use a values that corresponds to two specific scences representative for winter and summer months (see Section 2.4 for details). Collectively, these sampling and convolution steps describe the projection of the ensemble state vector $\mathbf{x}$ (time-dependent distributions of fluxes) to observation space $\mathbf{y}$ (the quantities being observed):

$$\mathbf{y}^b = H(\mathbf{x}^b), \tag{2}$$

where $H$ denotes the forward operator that includes the sample and convolution steps describe above and links the state vector and the observation vector.

We estimate the mean *a posterior* (analysis, $a$) flux estimates ($\overline{\mathbf{x}}^a$) by optimally fitting the model to the observations:

$$\overline{\mathbf{x}}^a = \overline{\mathbf{x}}^b + \mathbf{K}(\boldsymbol{y}^{obs} - \overline{\boldsymbol{y}}^b), \tag{3}$$

where $\mathbf{K}$ is the Kalman gain matrix, $\boldsymbol{y}^{obs}$ is the observation vector and $\overline{\boldsymbol{y}}^b$ is the ensemble mean of the GEOS-Chem model concentrations. The Kalman gain matrix, $\mathbf{K}$ (equation 3) governs the extent to which discrepancies between model predictions and observations will be reduced by adjusting the state vector. Calculating $\mathbf{K}$ involves the observation and *a posteriori* error covariances:

$$\mathbf{K} = \boldsymbol{\Delta}\mathbf{x}^b \mathbf{P}^a (\boldsymbol{\Delta}\mathbf{y}^b)^T \mathbf{R}^{-1}, \tag{4}$$

where $\boldsymbol{\Delta}\mathbf{y}^b$ represents the mean ensemble perturbations in the measurement space, defined as the difference between the ensemble state $\mathbf{y}^b$ and its mean value $\overline{\mathbf{y}}^b$, analogous to equation 1. $\mathbf{R}$ and $\mathbf{P}^a$ denote the observation and local *a posteriori* error covariance matrices, respectively. Our state vector includes emissions from $80 \times 64$ grid cells at $0.5° \times 0.625°$ resolution across the UK and scaling terms for our four lateral boundary condition of our nested transport model at a temporal resolution of 15 days.

For the synthetic EM27/SUN data, we use only data at a solar zenith angle of $\leq 78°$. We discard data where clear-sky observations are unavailable. The clear-sky condition is determined by evaluating the cloud cover fraction from ECMWF Reanalysis v5 (ERA5). For each hour, we calculate the probability of clear skies (1 - cloud fraction) and use N=15 random sounding samples to estimate the number of cloud-free scenes. Only hours with at least one clear-sky observation are retained. We also discard data with aerosol optical depth (AOD) > 0.3. For our experiments, we use hourly AOD data from ECMWF Atmospheric Composition reanalysis (EAC4) with a spatial resolution of $0.25° \times 0.25°$ and hourly cloud cover fraction data

from ERA5 that has a spatial resolution of $0.75° \times 0.75°$. To translate the resulting vertical profiles of $CO_2$ and methane to XCO$_2$ and XCH$_4$, respectively, we use scene-specific averaging kernels derived from the EM27/SUN instrument at UCL, London. For simplicity, we apply one kernel for January (based on retrievals from 10 December 2021) and one for July (16 June 2021). These kernels reflect the actual scene parameters, including solar zenith angle, on those days, but are representative of the values in those contrasting months.

The observation error covariance matrix, $\mathbf{R}$, include measurement uncertainties and atmospheric transport model error, which we add in quadrature. For $CO_2$ and methane, we assume a uniform model transport error of 3 ppm and 15 ppb, respectively, and prescribe the observation error using the scene-dependent standard deviation for each observed value. Observations collected far from fluxes typically have a weaker constraint on the flux estimates than closer observations. The model transport error is invariant so the importance of the observation error is significantly reduced with the distance downwind. In the LETKF approach (Lunt et al., 2021b; Scarpelli et al., 2024), we introduce a dampening factor as a function of the distance between each observation and the grid box, to suppress analysis increment from those remote observations, and estimate the posterior estimate at grid box $m$ as the sum of re-scaled increment from each single observation $i$:

$$\overline{\mathbf{x}}_m^a = \overline{\mathbf{x}}_m^b + \sum_i e^{-(l_{mi}/l_c)^2} (K_{mi}[\boldsymbol{y}_i^{obs} - \overline{y}_i^b]). \tag{5}$$

For the weighting coefficient $e^{-(l_{mi}/l_c)^2}$, $l_{mi}$ denotes the distance between observation $i$ and grid box $m$, and $l_c$ denotes the localization factor.

For our inverse model calculations (not shown), we use a 15-day assimilation window and a one-month lag window to account for the influence of emissions sampled downwind. In a full sequential inversion, *a posteriori* scale factors are evaluated for each assimilation window to update the *a priori* for the corresponding lag window, which requires perturbed runs for the preceding periods (e.g. December 2018 and June 2019 for the January 2019 and July 2019 flux estimation calculations). The inversion calculations in this full setup are localized, with the state vector influenced by observations only within a specified radius (e.g., 1000 km). In our study, we apply a simplified approach as clarified in the following sections.

**Error Characterisation of *a posteriori* Solution**

After applying the LETKF inverse method, we analyze the *a posteriori* solution to understand the impact of observations on the estimated emissions and to characterize the associated errors. The local posterior error covariance matrix $\widetilde{\mathbf{P}}^a$ accounts for the uncertainty in the emissions estimates after assimilating observational data:

$$\widetilde{\mathbf{P}}^a = [(N-1)\mathbf{I} + \mathbf{y}^b \mathbf{R}(\mathbf{y}^b)^T]^{-1}, \tag{6}$$

where $\mathbf{I}$ is the identity matrix and $N$ (100) denotes the number of ensemble members. The corresponding *a posteriori* error covariance matrix $\mathbf{P}^a$ describes the uncertainty in the *a posteriori* emissions:

$$\mathbf{P^a} = \boldsymbol{\Delta}\mathbf{x}^a (\boldsymbol{\Delta}\mathbf{x^a})^T (N-1)^{-1}, \tag{7}$$

where $\boldsymbol{\Delta}\mathbf{x}^a$ denotes the *a posteriori* emission perturbation ensemble.

## Metric to Assess the Theoretical Performance of GEMINI-UK

Since we use synthetic data, the performance of GEMINI-UK is assessed using error-reduction calculations based on the LETKF method described above, but representing a simplification of the full LETKF inverse calculations that will be applied to the real GEMINI-UK observations.

We use closed-loop numerical experiments to assess the theoretical potential of GEMINI-UK (plus the EM27/SUN based at UCL) on its own and of the added value of these data to the existing DECC tall tower data. For $CO_2$ and methane we run three sets of calculations using: 1) GEMINI-UK data alone; 2) GEMINI-UK and DECC data (plus Mace Head); and 3) DECC (plus Mace Head). Subtracting model run 3 from model run 2 provides us with an assessment of the added value of GEMINI-UK to the DECC network. We run these calculation once with the four operational DECC sites and then again also with Jodrell Bank and Invergowrie that will become operational in 2025.

We assess the improvement of adding GEMINI-UK by calculating the percentage error reduction $\eta$ for each element $j$ of the state vector corresponding to flux estimates:

$$\eta_j = 100 \left[ 1 - \left( \frac{\mathbf{P}_{j,j}^a}{\mathbf{P}_{j,j}^b} \right)^{1/2} \right], \tag{8}$$

where $\mathbf{P}_{j,j}^b$ denotes the *a priori* (background) error covariance matrix and all other variables are as previously defined.

The original inventories used in the GEOS-Chem simulation represent the synthetic "true" emissions/fluxes. The prior emissions/fluxes used in the inversion are generated by perturbing this true inventory to represent the initial uncertainty. Thus, the perturbed inventory serves as the prior estimate, while the original inventory acts as the synthetic truth for evaluation purposes. We use an atmospheric transport model error of 3 ppm for $CO_2$ and 15 ppb for methane at the *in situ* DECC sites. For the total column observations from the GEMINI-UK network, we adopt smaller atmospheric transport model errors (0.5 ppm for $CO_2$ McNorton et al. (2020), and 5 ppb for XCH4, consistent with Stanevich et al. (2021), reflecting their reduced sensitivity to local variability near the surface. In addition, we assess the impact of uncertainty assumptions by repeating the analysis using values that are half and double of our assumed atmospheric transport model errors for XCO2 and XCH4 (Fig. B11-B16, Tables B1-B2).

### 2.5 GEMINI-UK Network Design

We designed the GEMINI-UK network to fulfill a number of objectives. Above all, as described below, the network has been designed to maximise the reduction in uncertainty of $CO_2$ fluxes based on *a priori* knowledge. We show below that this optimized network also works well for estimating methane emission estimates. The locations of two EM27/SUN sites were chosen specifically so they link with other measurement networks.

We chose the Weybourne Atmospheric Observatory (Table 1) because they also host an *in situ* methane and $CO_2$ sensors and because it is within 60 km of the Tacolneston tall tower that is part of the DECC network. This close proximity to *in situ* sensors allows us to study the relationship between $CO_2$ and methane columns and the surface data. We chose the Rutherford

Appleton Laboratory at Harwell in Oxfordshire because they also host the IFS120/5 HR Bruker spectrometer that is currently the only UK contribution to TCCON (Weidmann et al., 2023, 2025). This allows us to compare our GEMINI-UK instrument with a higher-resolution instrument that is linked indirectly with WMO scale calibrated working standards via TCCON and whose data are vetted using TCCON's data quality assurance protocols. We are working closely with COCCON so that our instrument data are integrated into a wider network of similar sensors.

Generally, individual sites are hosted by or are affiliated with educational or research institutes so that data can be used for teaching as well as research. In our experience, this also attracts an enhanced level of ownership by the host institution. Because we have chosen to host GEMINI-UK spectrometers with educational institutes, we have been able to distribute the sensors more evenly across the devolved administrations than possible with tall towers. To promote transparency and data openness, all our data will be freely available for academic research purposes, subject to an embargo period to check the data passes through quality control/assurance protocols.

To support the network design and site selection process, we use the nested GEOS-Chem simulation, described above, to model $CO_2$ emissions and atmospheric concentrations across the British Isles, including the UK and Ireland. Our reference model is driven by *a posteriori* flux estimates from our global model inferred from OCO-2 retrievals of column-averaged dry-air mixing ratio of carbon dioxide ($XCO_2$) and surface flask mole fraction data (Feng et al., 2017). We divide the landmass of the UK and Ireland into 58 grid boxes each with an area of $1° \times 1°$. We use the nested GEOS-Chem model simulation to evaluate the contributions of $CO_2$ flux from each of the 58 grid boxes to simulated $XCO_2$ values "observed" at a long list of 40 geographical locations across the UK colocated with further or higher education institutes. We systematically evaluate this contribution by sampling the 4-D model fields corresponding to our perturbation run (without an emission at the $m$th grid box) with our reference model that emissions for each of the 58 grid boxes. We convert the vertical profiles at each measurement site $i$ into $XCO_2$ for the reference ($XCO_2^r(i)$) and perturbed ($XCO_2^p(i,m)$) model runs by applying an averaging kernel from an EM27/SUN, corresponding to an appropriate solar zenith angle, and then take the difference:

$$\Delta XCO_2(i,m) = XCO_2^p(i,m) - XCO_2^r(i). \tag{9}$$

We estimate the overall sensitivity $S(m)$ of any chosen subset of candidate measurement sites to the flux at grid box $m$ by summing its contribution to $XCO_2$ values at the subset of candidate sites:

$$S(m) = \sqrt{\sum_i \Delta XCO_2(i,m)^2}. \tag{10}$$

Finally, we chose the optimal subset from all feasible options based on the distribution of $S(m)$ over UK and Ireland during January and July, 2019. Based on our calculations, and the underlying approach we adopted, the final selection of 10 sites for the GEMINI-UK network are shown in Table 1 and Figure 1. For the purposes of the initial network, we also assume a site at University College London (UCL) currently operated by the NERC Field Spectroscopy Facility.

## 3 Results

First, we evaluate the performance of the nested GEOS-Chem model to reproduce data collected across the DECC measurement network. In particular, we assess the influence of different lateral boundary conditions on determining atmospheric $CO_2$ and methane across the UK. Second, we use observations of cloud cover and AOD to demonstrate theoretical data coverage provided by individual sites across GEMINI-UK. Third, we report results from our closed-loop experiments that show the individual and collective theoretical performance of the GEMINI-UK network and the DECC data to determine UK $CO_2$ and methane fluxes. Finally, we report an initial comparison of the GEMINI-UK instrument deployment at Harwell and the colocated TCCON site.

### 3.1 Baseline $CO_2$ and Methane Model Performance against the DECC and Mace Head Mole Fraction Data

Our baseline nested model uses three sets of lateral boundary conditions that are informed by a global model that has been fitted to *in situ* or satellite remote sensing data, as described above. Fig. 2 shows a statistical comparison of model and observed $CO_2$ mole fraction data sampled at the four operational DECC sites and Mace Head (Fig. 1) for January and July 2019. Fig. 3 shows a similar comparison for atmospheric methane.

The model shows good agreement with $CO_2$ and methane mole fraction observations, with Pearson correlation coefficients corresponding to the model capturing 58%–59% of the variations in the diurnal cycle during January but only 30%–33% of the variation in the diurnal cycle during July. For both months, biases range ±4 ppm for $CO_2$ and ±40 ppb for methane, depending on the lateral boundary condition dataset used. Methane is particularly sensitive to the assumed boundary conditions. The biases for methane using the CAMS-EGG4 boundary conditions are particularly large, with a mean bias of -36.5 ppb at Heathfield in January and -43.4 ppb at Ridge Hill in July. In contrast, the GEOS-Chem global and CAMS *in situ* boundary conditions show smaller biases for which methane and $CO_2$ are typically within ±20 ppb and ±4 ppm for January and July 2019, respectively.

Fig. 2 shows that for $CO_2$ there is very little difference in the overall performance reproducing DECC data using the three lateral boundary conditions during January 2019, in terms of the Pearson correlation or the bias. For July 2019, while the Pearson correlation is lower and the magnitude of the bias is larger there is little between using the competing boundary conditions, with the CAMS EGG4 boundary condition marginally better. Fig. 3 shows less of a difference between the contrasting months and between the different assumed lateral boundary conditions.

Assessing model performance at individual sites reveals a different picture (Fig. 4). For $CO_2$, we find the smallest biases during January 2019 are typically associated with the lateral boundary conditions that are determined using *in situ* data, i.e. GEOS-Chem and CAMS. These outperform the CAMS EGG4 product at all sites. In contrast, during July 2019 we find that the EGG4 product outperforms the other lateral boundary condition products at all sites. For methane, we see much smaller differences between the two contrasting months, with the lateral boundary conditions informed by *in situ* data outperforming EGG4 at all sites. Overall, we find that the CAMS *in situ* lateral boundary conditions provide one of the lowest biases at the DECC sites (Fig. 2-3, Fig. B1-B10), particularly in comparison to CAMS-EGG4, and therefore we use them for the subsequent calculations in the perturbed GEOS-Chem model runs.

## 3.2 Synthetic EM27/SUN observations sampled from the baseline model

To realistically examine the theoretical potential of GEMINI-UK to quantify UK $CO_2$ and methane fluxes, we filter the data for excessive cloud cover and AOD values, and for SZA$\leq78°$. For this we use cloud and AOD reanalysis fields, as described above. We consider 10 GEMINI-UK sites and the EM27/SUN installation at UCL.

Figure 5 shows the result of our filtering criteria for 1-15 January and 1-15 July 2019, which reflects our 15-day assimilation cycle, described earlier. As expected, data coverage over the UK is significantly lower in January than July. This is due mainly to the SZA constraint but also due to cloud cover during winter months. During 1-15th of January 2019, we estimate a total of 264 observations were generated. In contrast, during 1-15th of July 2019, this value increases substantially to 1400 observations. In January we lose the majority of data to our SZA constraint, with an additional 171 scenes (39%) discarded because of clouds. Cloudy scenes are responsible for most of the 815 scenes (37%) we discard in July.

On a location basis, we find that Aberdeen and Glasgow – the most northerly sites in GEMINI-UK – record the lowest number (0, 4 repectively) of observations during January, and Cardiff and London record the largest number (38, 39 respectively). During July, Glasgow records the lowest number (92) of observations while Guernsey records the largest number (178) observations.

## 3.3 Theoretical Performance of GEMINI-UK

Figures 6-8 show the theoretical error reduction associated with fitting *a priori* flux estimates of $CO_2$ and methane to atmospheric observations of $CO_2$ and methane from the GEMINI-UK and DECC data.

If we consider the information provided exclusively by the GEMINI-UK network (Fig. 6, Table 2), we find that during January the largest reductions in uncertainty is over the Midlands and southern England, with foci coincident with the GEMINI-UK sites (e.g., Cardiff) that have error reductions of up to 50.9% for $CO_2$ and 54.9% for methane. Averaged over the UK and Ireland area, the reductions amount to 9.4% for $CO_2$ and 15.8% for methane (Table 2). The lack of information in Scotland and Ireland is unsurprising given the data coverage during this month (Fig. 5). During July, we find the error is more uniform across the British Isles, with much higher values that peak at 58.9% for $CO_2$ and 74.7% for methane, with average reductions over the UK and Ireland region of 28.0% for $CO_2$ and 39.0% for methane (Table 2). This calculation highlights the effectiveness of GEMINI-UK data on its own, particularly during summer months.

The added value of GEMINI-UK data to the DECC network is shown by Figure 7. During January, we find that GEMINI-UK data contributes to an additional error reduction of up to 7.9% for $CO_2$ and up to 9.0% for methane (Table 2), with foci in locations away from DECC sites (Fig. 1), e.g., Northern Ireland, Lancashire, West Yorkshire, and Ayrshire, and with mean reductions over the UK and Ireland region of 1.0% for $CO_2$ and 1.1% for methane (Table 2). During July, GEMINI-UK data play a much larger role, contributing an additional error reduction of up to 24.3% for $CO_2$ and up to 28.7% for methane, particularly across Scotland, Antrim, Derry, Oxfordshire, Greater London, and Devon, with mean reductions over the UK and Ireland region of 7.2% for $CO_2$ and 6.0% for methane (Table 2). This calculation shows that data collected by the GEMINI-UK network provides information about $CO_2$ and methane fluxes over and above that provided by the DECC network.

We also estimate the added value of GEMINI-UK to the DECC data when it also includes data collected at Jodrell Bank and Invergowrie. Figure 8 shows that the impact of GEMINI-UK is decreased, as expected. The maximum error reductions are
445 7.6% for $CO_2$ and 7.2% for methane in January (mean reductions of 0.8% for both species), and 17.7% for $CO_2$ and 19.2% for methane in July (mean reductions of 4.8% for $CO_2$ and 3.7% for methane). Overall, the error reductions (shown in Fig. 6- 8) correspond spatially to regions with substantial *a priori* fluxes, as illustrated in Fig. 9. This demonstrates that the GEMINI-UK data improve knowledge particularly over areas with high net $CO_2$ and methane fluxes.

The sensitivity tests with increased and decreased assumed atmospheric transport errors for $CO_2$ and methane led to corre-
450 sponding decreases and increases in the error reduction values (Tables B1- B2, Figs. B11- B16), while the main spatial patterns were preserved.

The theoretical performance quantified here relies on calculations using the relative error reduction between prior and posterior error covariance matrices. This approach allows evaluation of the potential constraints provided by the observing system, but it does not capture all sources of uncertainty, including systematic errors arising from model biases or representation errors.
Such limitations are typical of closed-loop experiments, where the same transport model is used to generate the synthetic truth and to perform the error reduction calculations by perturbing the prior fluxes and applying LETKF.

## 3.4 Initial Comparison with the Harwell TCCON site

Ahead of the full GEMINI-UK network installation, we deployed one of the GEMINI-UK EM27/SUN instruments in a weatherproof enclosure (see Appendix A) at the Rutherford Appleton Laboratory in Harwell, Oxfordshire, at the end of February
2025. This is also the location of the Harwell TCCON site, which has provided ground-based remote sensing observations of averaged dry column concentrations of atmospheric greenhouse gases since September 2020 (Weidmann et al., 2025). By locating one of the GEMINI-UK instruments next to the Harwell TCCON, we can demonstrate (and continuously evaluate) the performance of the EM27/SUN and enclosure setup that will be deployed at each GEMINI-UK location, relative to that of the TCCON data, thus providing an indirect link to WMO concentration scales. This initial comparison between the two,
covering a time period from February 24th to June 16th 2025, gives us a first indication of the GEMINI-UK data quality, using the TCCON data as a baseline from which we can quantify the errors and uncertainty in the GEMINI-UK observations.

For this comparison, we use data from EM27/SUN serial number 197 which has been processed using the PROFFAST retrieval algorithm as described in Section 2.1. The TCCON data here have been processed using the standard GGG2020 algorithm and methodology, and are publicly available (Weidmann et al., 2023; Laughner et al., 2024). The same *a priori*
profiles, obtained from the GEOS FP-IT global climatological model (Goddard Earth Observing System Forward Product for Instrument Teams, Laughner et al. (2023)). For each dataset, we first calculate the median values for every 30-minute interval, rejecting those intervals where at least one of the two instruments provided fewer than five successful retrievals. Figure 10 shows scatter plots of the median values of $CO_2$ and methane, with Pearson correlations of 0.78 and 0.92, and mean differences ($\pm$ one standard deviation) of 0.19$\pm$0.32 ppm and 1.5$\pm$2.5 ppb respectively. The histogram plots in Figure 10
show that these differences generally fall within one standard deviation of zero. Small differences between the two datasets are expected, since the two instruments measure the solar absorption spectra at different spectral resolutions, and different retrieval

algorithms are used in each case (Frey et al., 2019). An in-depth, systematic comparison between the Harwell EM27/SUN and TCCON greenhouse gas column concentrations, which will investigate the differences and errors introduced by different spectral resolutions (e.g., by truncating the TCCON interferograms to match the EM27/SUN resolution), retrieval algorithms,
and the assumed a priori atmospheric profiles, will be the subject of a dedicated paper once a longer time period of data is available for analysis.

## 4 Closing Remarks

GEMINI-UK is the first national-scale network of ground-based remote sensing instruments that has been designed to quantify net fluxes of $CO_2$ and methane. It forms one component of a UK measurement verification support system to deliver actionable
information to government and complements measurements collected by the Deriving Emissions linked to Climate Change (DECC) tall tower network. GEMINI-UK comprises ten Bruker EM27/SUN spectrometers that collect measurements of incoming short-wave IR data that are sensitive to changes in $CO_2$ and methane in the lower atmosphere. The spectrometers are operated within bespoke weatherproof enclosures that enable a level of autonomy to collect the maximum volume of clear-sky data with infrequent human interaction.
We have designed the network using a Bayesian approach to ensure we locate the sensors so they deliver the biggest reduction in *a priori* $CO_2$ flux uncertainty. The resulting network is hosted by further (post-secondary, non-degree) and higher (university degree) education institutions, underlining our commitment to delivering open data for all to use. Using a series of closed-loop numerical experiments, we find our network reduces uncertainties in $CO_2$ flux estimates by up to 15%–63% in January and 29%–72% in July 2019. Our network also delivers substantial uncertainty reductions in methane emissions, ranging from
13%–70% in January to 32%–87% in July 2019. This capability also provides redundancy for existing networks such as DECC, ensuring we continue to collect a timeseries of atmospheric $CO_2$ and methane. Data collected by GEMINI-UK also provide the basis to evaluate satellite observations of $CO_2$ and methane, particularly in the context of upcoming data from the Copernicus Carbon Dioxide Monitoring mission (CO2M). In doing so, we provide confidence in being able to use CO2M data to further improve our ability to quantify changes in UK $CO_2$ and methane emissions. Because CO2M also includes $NO_2$
column measurements, we should also be able to improve our ability to estimate combustion emissions of $CO_2$.

GEMINI-UK is one component of the broader UK Greenhouse Gas Emissions Measurement and Modelling Advancement (GEMMA) framework. The objective of the first phase of GEMMA is to deliver regular, reliable, and robust knowledge of greenhouse gas (GHG) emissions in a form that can be digested by UK Government and other stakeholders. GEMMA uses data collected by established networks like TCCON, ICOS, and DECC, and to translate these atmospheric data into spatially
and temporally resolved GHG emission estimate we use atmospheric chemistry transport models and inverse methods. This integrative approach helps GEMMA address observational gaps and to support advancements in emissions measurement and modelling.

Subsequent phases of GEMMA will focus on our ability to estimate GHG emissions from individual sectors, e.g., agriculture for methane. They will also progressively improve emission estimates on smaller spatial scales and shorter timescales that

provide a stronger link between climate legislation and emission reductions, ultimately advising on the efficacy of climate policies and strategies. These improvements will be achieved in part by adopting new technologies, which help improve the sustainability and resilience of the observing network, and analysis techniques that translate the data into actionable information for stakeholders.

*Code and data availability.* The DECC (O'Doherty et al., 2020) and TCCON (Weidmann et al., 2023) data are available from the Centre for Environmental Data Analysis (https://www.ceda.ac.uk/; last accessed 8th Jan, 2025). The gridded ERA5 cloud cover data (Her) are available from Copernicus Climate Change Service (C3S) Climate Data Store (https://cds.climate.copernicus.eu; last accessed 8th Jan, 2025). The CAMS global reanalysis EAC4 AOD data (Inness et al., 2019), the CAMS $CO_2$ and methane concentrations (Chevallier, 2023; Segers, 2023), and CAMS EGG4 reanalysis data (Agustí-Panareda et al., 2023) are available from the Atmosphere Data Store (ADS) archive (https://ads.atmosphere.copernicus.eu; last accessed 8th Jan, 2025). The GEOS-Chem atmospheric chemistry and transport model is maintained centrally by Harvard University (https://geoschem.github.io/; last accessed 8th January, 2025) and is available on request. The ensemble Kalman filter code is available on request.

*Author contributions.* AK, LF, and PIP designed the study. AK adapted and implemented the EnKF code to work with GEOS-Chem v14.3.1, performed the calculations, with inputs from LF (network design), conducted the analysis and prepared the figures (Figures 1-9) using Python. AK and PIP wrote the manuscript, with inputs from NH (Sections 2.1, 3.4, Figure 10), LF (Section 2.4), and AJPW (Appendix A). SD and DW supported the installation and operation of the Harwell EM27/SUN instrument and runs the Harwell TCCON site providing the comparison data. All authors helped to revise the paper.

*Competing interests.* The authors declare that they have no competing interests

*Acknowledgements.* We thank the University of Bristol for maintaining the $CO_2$ and methane mole fraction data record from the DECC tall tower network. We also thank the GEOS-Chem community, particularly the team at Harvard University who help maintain the GEOS-Chem model, and the NASA Global Modeling and Assimilation Office (GMAO) who provide the MERRA2 data product. AK, NH, and PIP are funded by the GEMMA programme via NERC and the UKRI Building a Green Future theme (NE/Y001788/1), and in-kind contributions from NPL and the Met Office. We also acknowledge funding for LF, NH, and PIP from the UK National Centre for Earth Observation (NCEO) funded by the National Environment Research Council (NE/R016518/1). AJPW acknowledges funding by the University of Edinburgh. DW and SD acknowledge support from the GEMMA programme to install and maintain the EM27/SUN at Harwell, and from STFC RAL and NCEO to support the continued operation of the Harwell TCCON observatory.

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

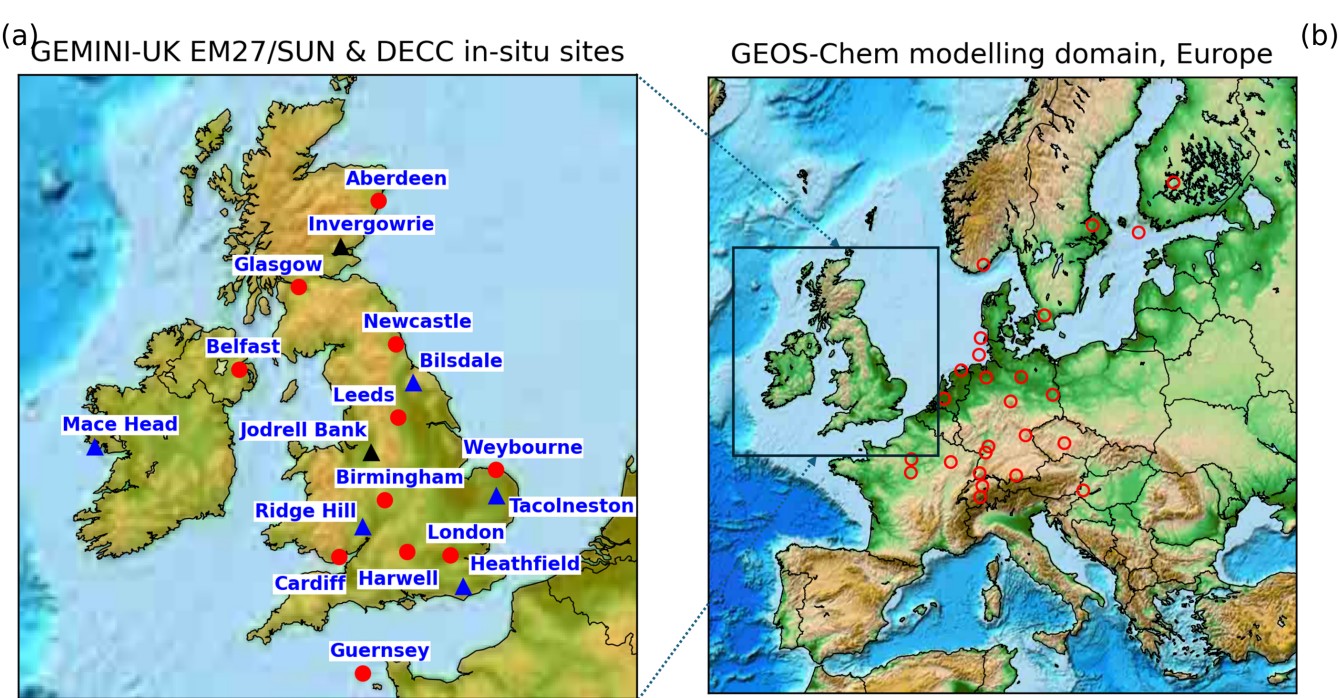

**Figure 1. (a)** Location of GEMINI-UK site (red dots), DECC tall towers over the UK and Mace Head in Ireland (triangles), with black triangles representing the forthcoming Jodrell Bank and Invergowrie sites. **(b)** The GEOS-Chem nested model domain over Europe. Circles denote the locations of ICOS site that will be used when we interpret $CO_2$ and methane data over the UK.

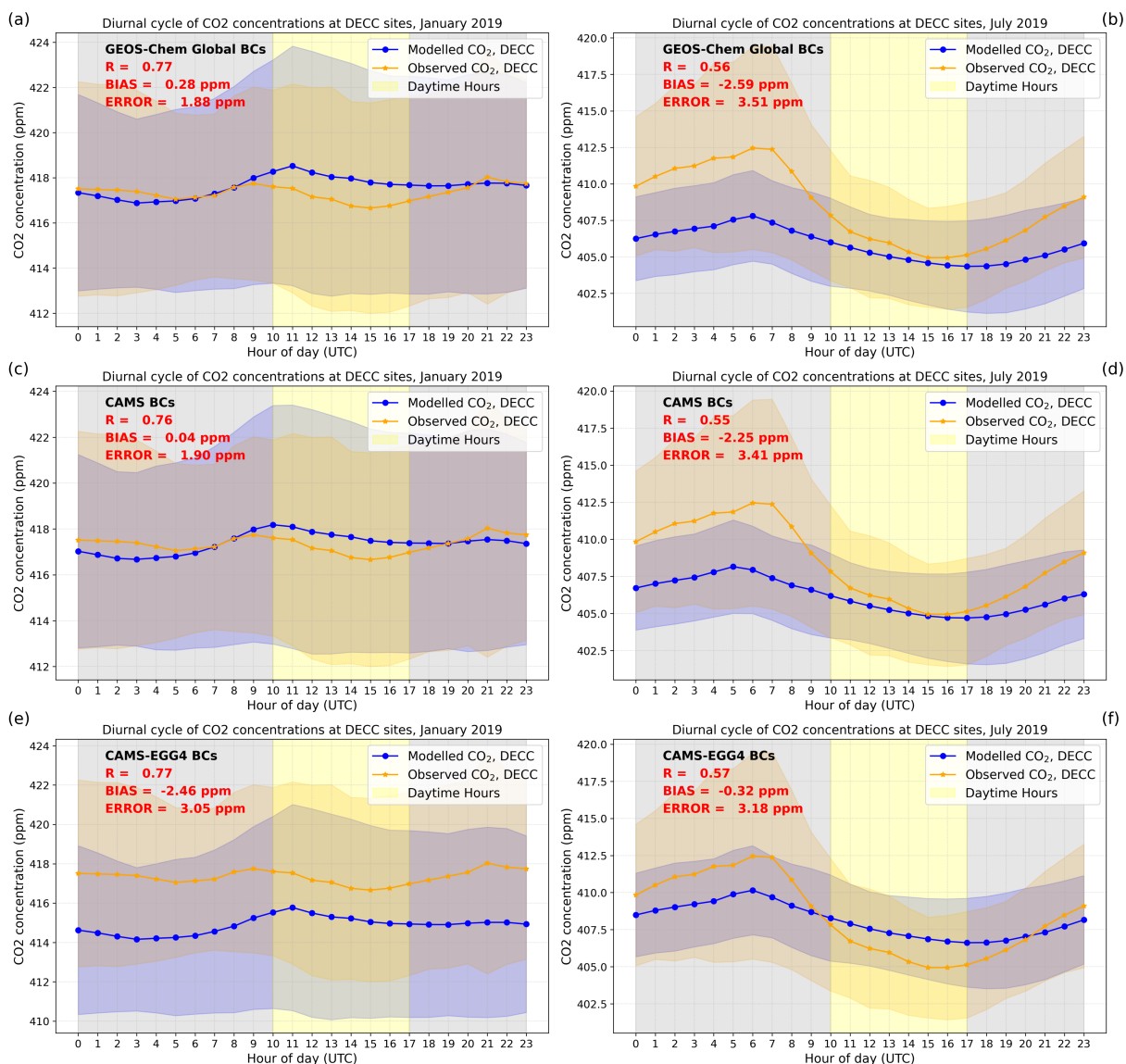

**Figure 2.** Mean diurnal cycle of observed (orange) and GEOS-Chem model (blue) $CO_2$ mole fraction values at four DECC tall tower sites and Mace Head for January (**(a)**, **(c)** and **(e)**) and July (**(b)**, **(d)** and **(f)**) 2019. Shaded envelopes denote the $1$-$\sigma$ value about the mean value. The top, middle, and bottom rows denote results from using lateral boundary conditions for the nested GEOS-Chem model from the corresponding global GEOS-Chem simulation, the CAMS model that is constrained by *in situ* data, and the CAMS EGG4 model reanalysis that is constrained by satellite data, respectively. The model is sampled at the time and at the top inlet height of each tall tower. Daytime hours (10-17 UTC) are denoted by the vertically region highlighted in yellow when the DECC data are typically used for the inverse calculations. Note, R = Pearson correlation coefficient, BIAS = mean bias, ERROR = Mean Absolute Error (MAE), calculated over the full time series.

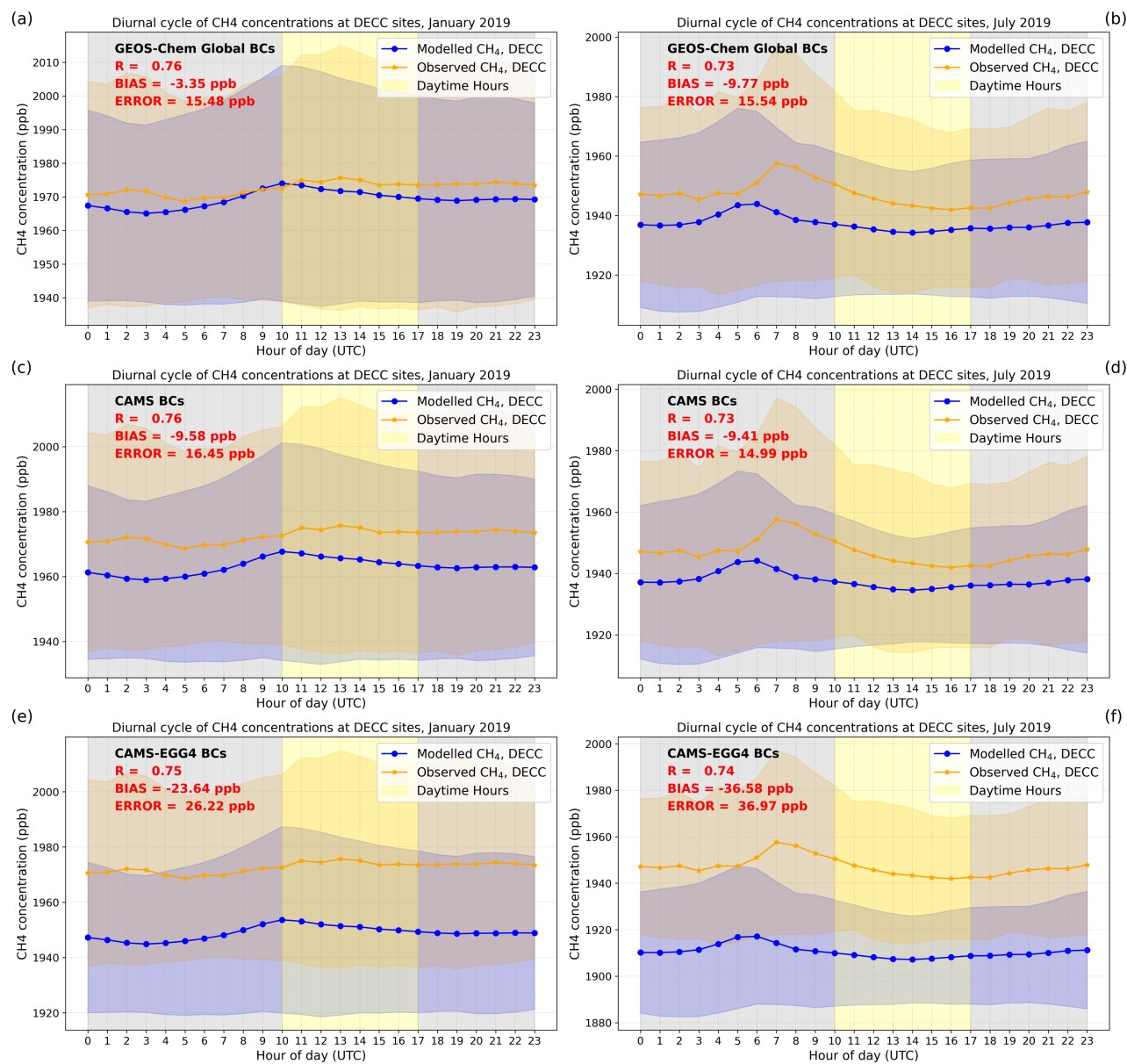

**Figure 3.** As Figure 2 but for methane.

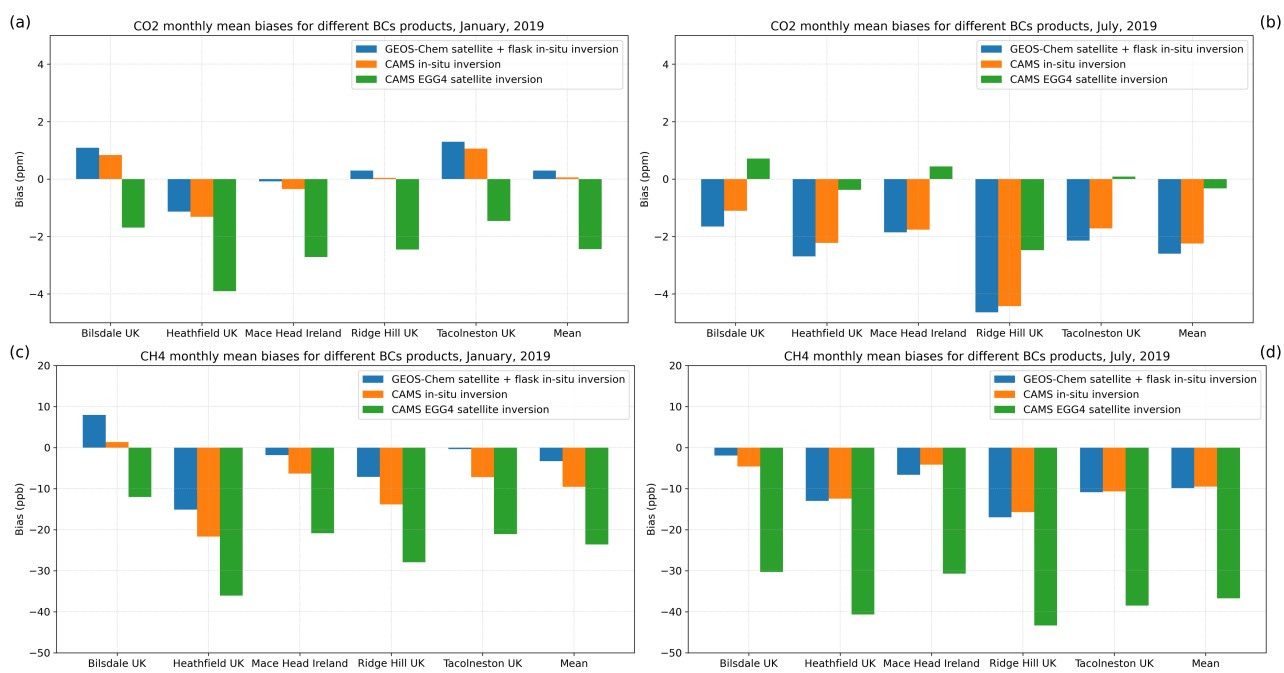

**Figure 4.** Mean bias between observed and GEOS-Chem $CO_2$ (**(a)** and **(b)**) and methane (**(c)** and **(d)**) mole fractions for January (**(a)** and **(c)**) and July (**(b)** and **(d)**). The three different lateral boundary condition data products are denoted by different colours: GEOS-Chem global model constraints by *in situ* and satellite data (blue); CAMS model constrained by *in situ* data (orange); and CAMS EGG4 reanalysis constrained by satellite data (green).

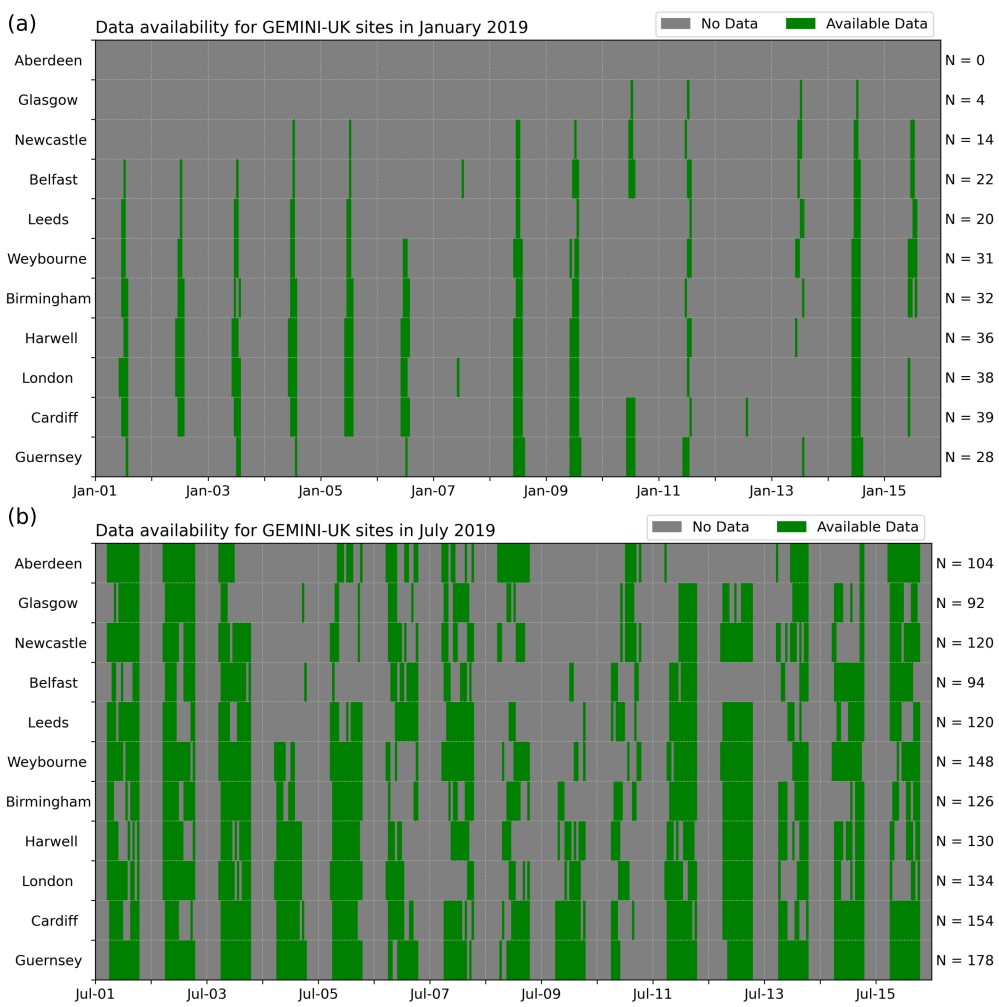

**Figure 5.** Data availability for the synthetic EM27/SUN $CO_2$ column observations across the GEMINI-UK network for the first 15-day assimilation window of January **(a)** and July **(b)** 2019, accounting for changes in solar zenith angle, cloud cover and aerosol optical depth (AOD). N represents the total number of records sampled at each site during the considered period.

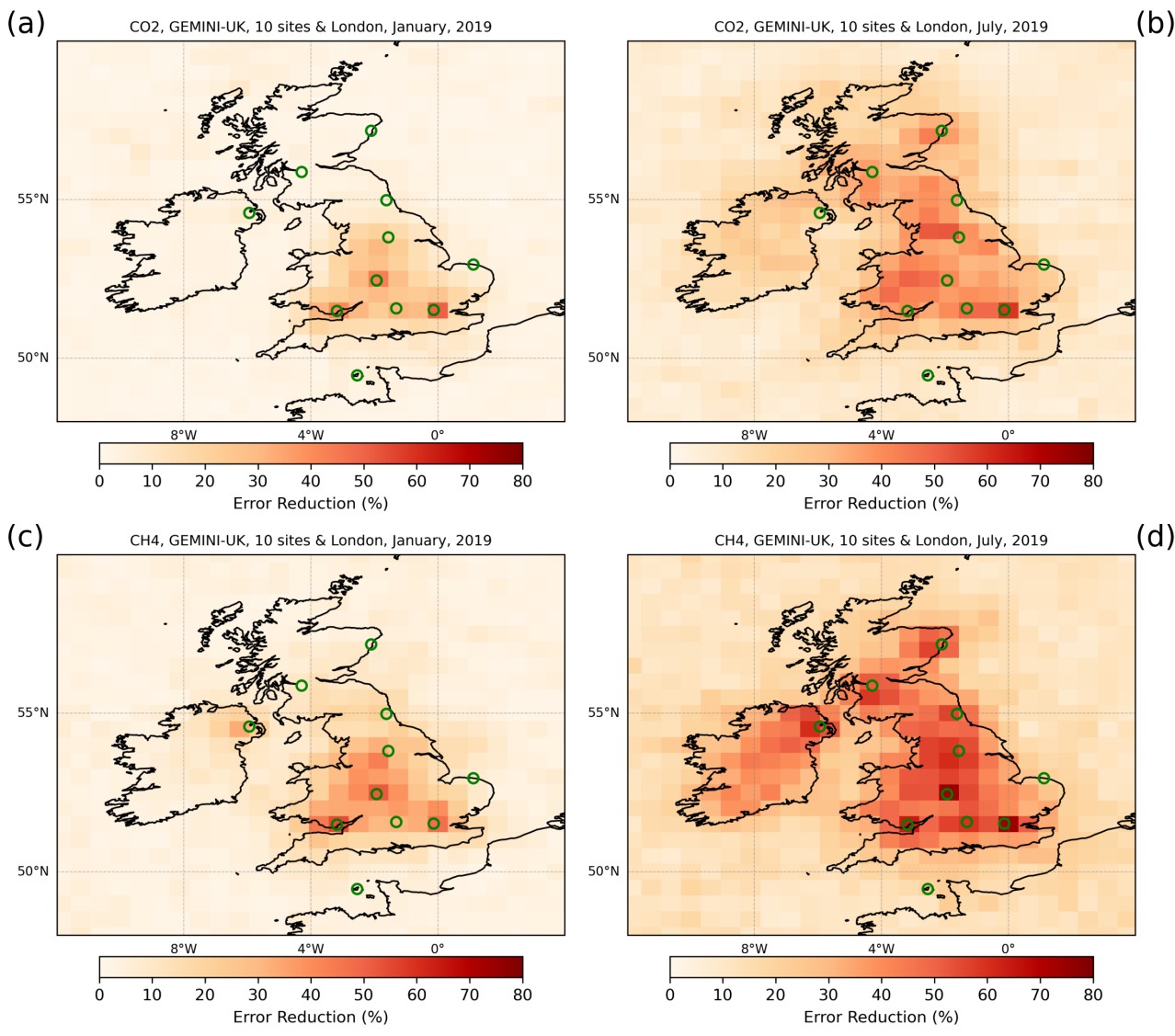

**Figure 6.** *A posteriori* error reduction of *a priori* net $CO_2$ ((**a**) and (**b**)) and methane ((**c**) and (**d**)) flux uncertainties from GEMINI-UK data (site locations: green circles) for the first 15-day assimilation window of January ((**a**) and (**c**)) and July ((**b**) and (**d**)) 2019.

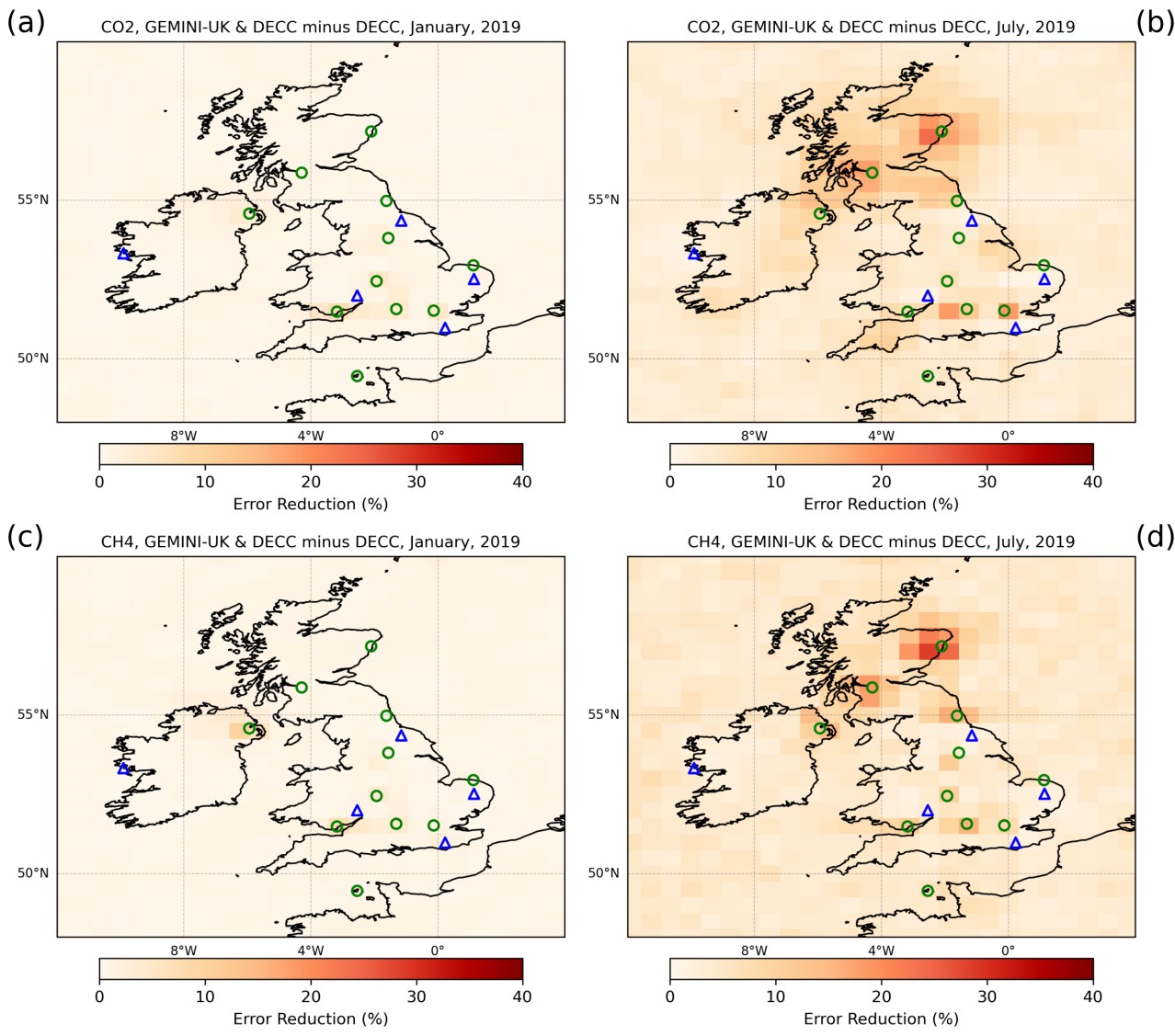

**Figure 7.** Same as Figure 6 but relative to the information already provided by the DECC tall tower data and Mace Head (site locations: blue triangles).

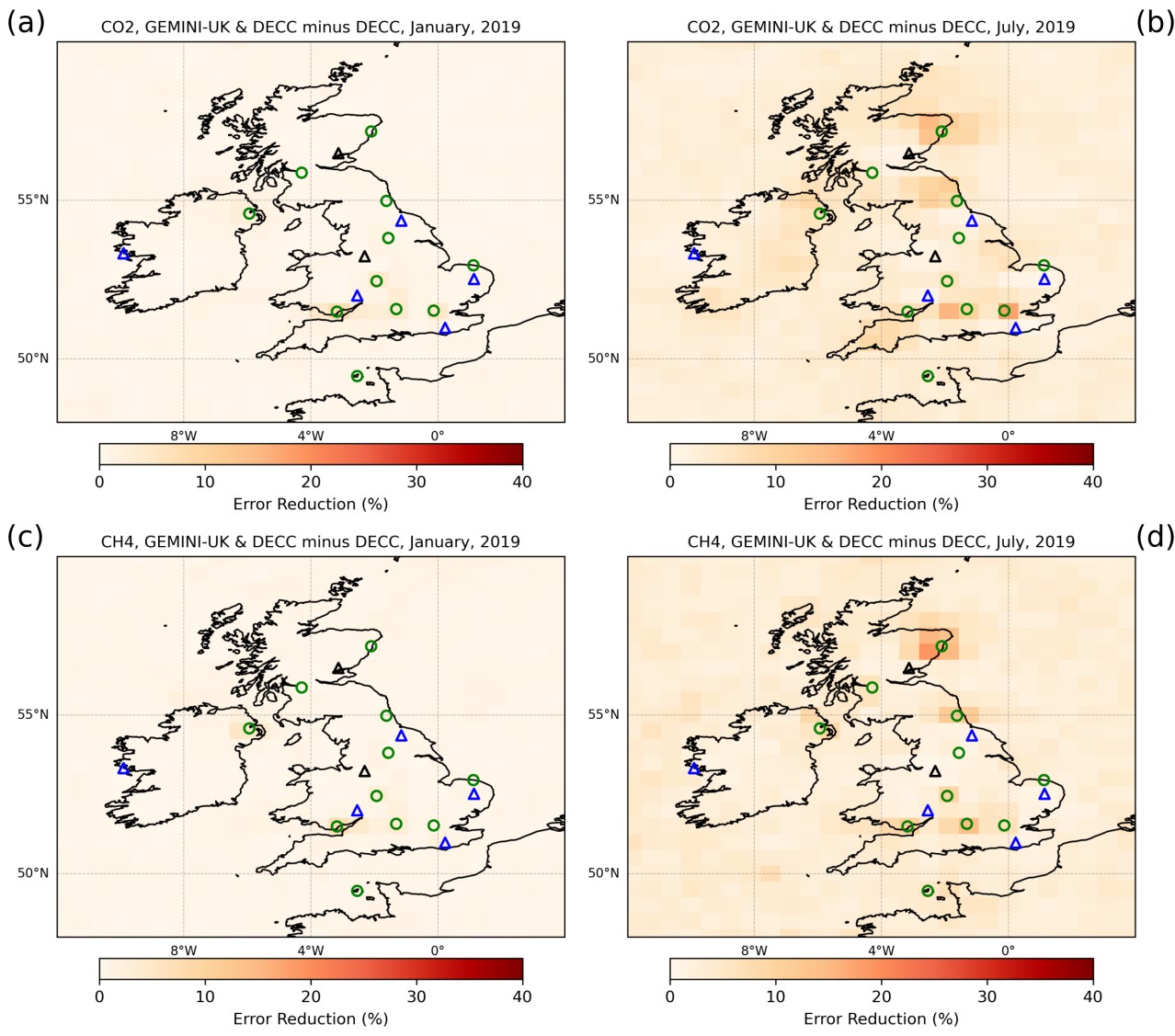

**Figure 8.** Same as Figure 7 but with the DECC tall tower data also including sites at Jodrell Bank and Invergowrie (site locations: black triangles) that will become operational in 2025.

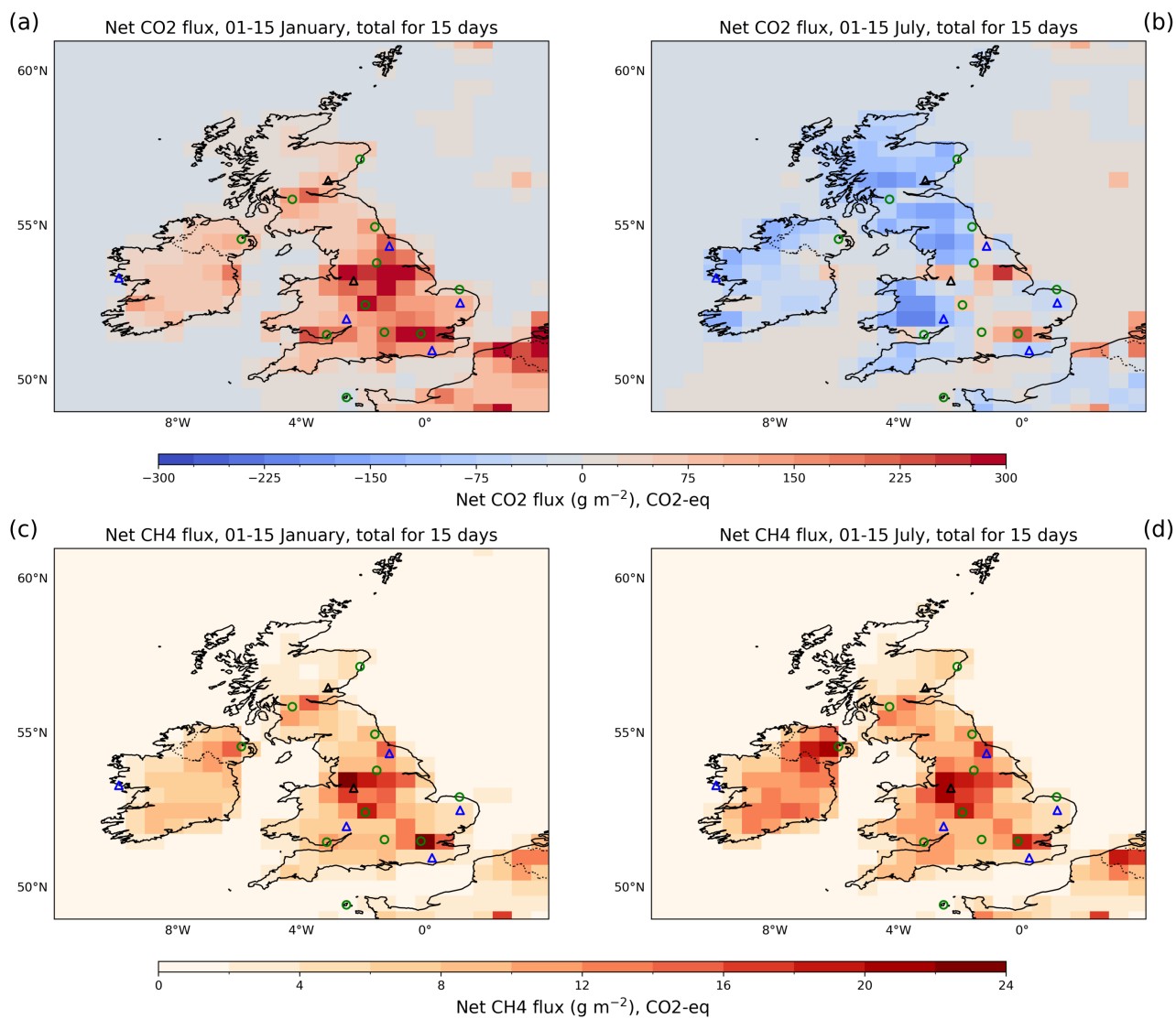

**Figure 9.** Net *a priori* fluxes of $CO_2$ ((**a**) and (**b**)) and methane ((**c**) and (**d**)) for the first 15-day assimilation window of January ((**a**) and (**c**)) and July ((**b**) and (**d**)) 2019. The methane is plotted as $CO_2$-equivalent. Site locations are shown as green circles (GEMINI-UK), blue triangles (DECC sites), and black triangles (Jodrell Bank and Invergowrie).

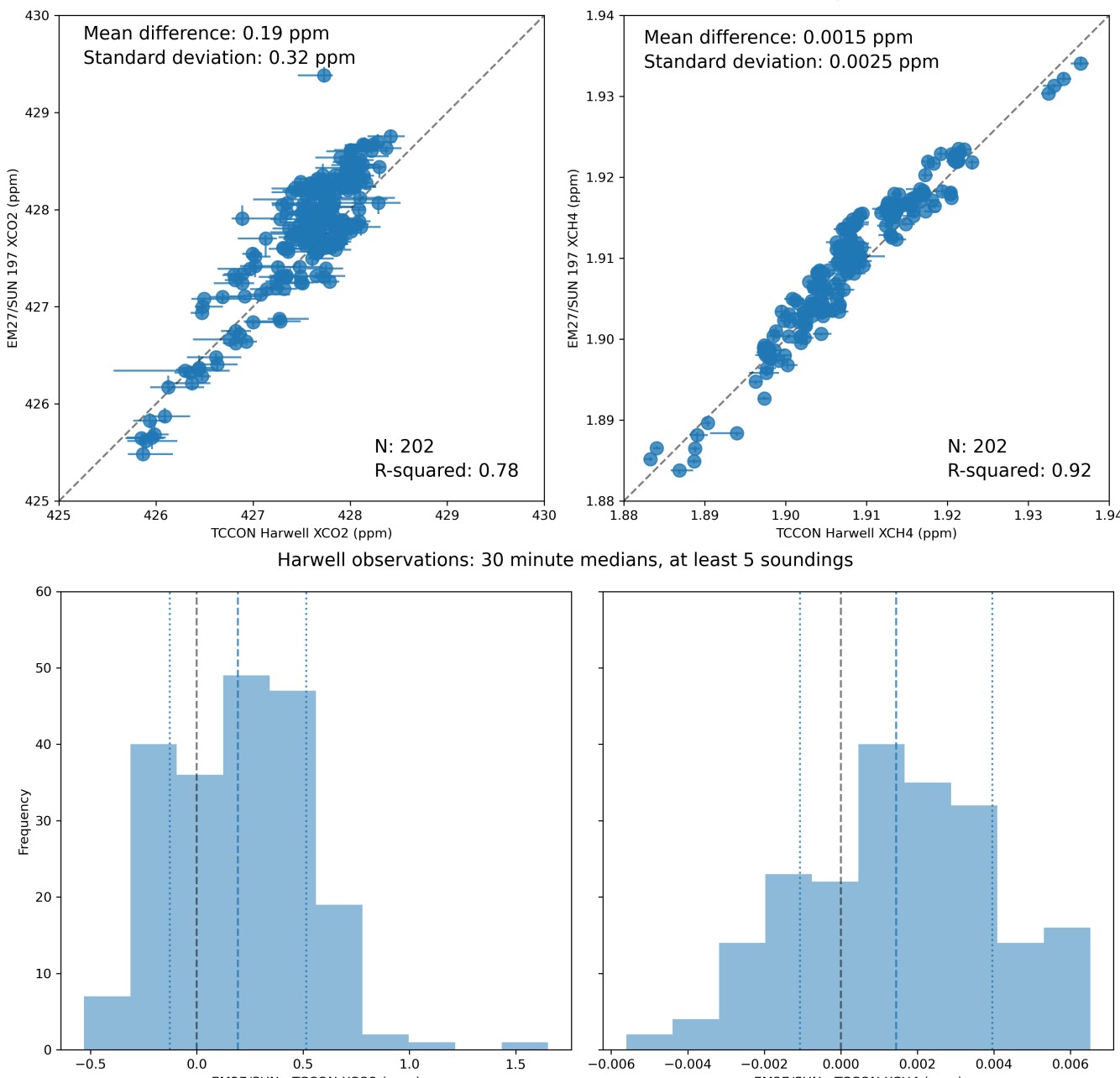

**Figure 10.** Scatterplots and histograms of 30-minute median values of $XCO_2$ (**(a)** and **(c)**) and $XCH_4$ (**(b)** and **(d)**) collected by the colocated EM27/SUN (serial number 197) and TCCON (IFS120/5 HR) instruments at Harwell, 24th February 2025 to 16th June 2025. The horizonal and vertical blue lines for each data point shown in the scatterplots denote the standard deviation of the median values. The dashed grey line denote the 1:1 line. Mean statistics are shown inset. For the histograms, the dashed blue line indicates the mean difference and the dotted blue lines show the mean $\pm$ the standard deviation. The vertical dashed black line denotes zero difference.

**Table 1.** Locations chosen for the GEMINI-UK EM27/SUN network.

| Location | Host institute | Latitude (°N) | Longitude (°E) | Height above sea level (m) |
|---|---|---|---|---|
| **Aberdeen** | University of Aberdeen | 57.17 | -2.10 | 33 |
| **Glasgow** | University of Glasgow | 55.87 | -4.29 | 42 |
| **Newcastle** | Northumbria University | 54.98 | -1.62 | 55 |
| **Belfast** | Queen's University Belfast | 54.58 | -5.94 | 73 |
| **Leeds** | University of Leeds | 53.87 | -1.32 | 55 |
| **Weybourne** | NCAS/University of East Anglia | 52.95 | 1.12 | 18 |
| **Birmingham** | University of Birmingham | 52.45 | -1.93 | 150 |
| **Harwell** | Rutherford Appleton Laboratory | 51.57 | -1.31 | 142 |
| **London** | University College London | 51.52 | -0.13 | 68 |
| **Cardiff** | Cardiff University | 51.49 | -3.18 | 31 |
| **Guernsey** | Elizabeth College | 49.46 | -2.54 | 47 |

**Table 2.** Error reduction (%) for $CO_2$ and $CH_4$ fluxes over the UK + Ireland for different network configurations. Results are shown for GEMINI-UK alone and GEMINI-UK combined with 5 or 7 DECC sites. Statistics include the minimum, mean, and maximum error reduction for January and July. All results assume an atmospheric transport model error of 0.5 ppm for $XCO_2$ and 5.0 ppb for $XCH_4$. Instrumental uncertainties were assumed to be 1 ppm for $XCO_2$ and 11 ppb for $XCH_4$ and were scaled by the number of observations per site per hour (N=25). The corresponding figure panels are indicated in the last column.

| Region | Spec | Network | # sites | Month | Min (%) | Mean (%) | Max (%) | Figure |
|--------|------|---------|---------|-------|---------|----------|---------|--------|
| UK + Ireland | $CO_2$ | GEMINI-UK | 11 | January | 0.7 | 9.4 | 50.9 | 6a |
| UK + Ireland | $CO_2$ | GEMINI-UK | 11 | July | 8.4 | 28.0 | 58.9 | 6b |
| UK + Ireland | $CH_4$ | GEMINI-UK | 11 | January | 1.2 | 15.8 | 54.9 | 6c |
| UK + Ireland | $CH_4$ | GEMINI-UK | 11 | July | 10.0 | 39.0 | 74.7 | 6d |
| UK + Ireland | $CO_2$ | GEMINI-UK & DECC | 16 | January | 0.2 | 1.0 | 7.9 | 7a |
| UK + Ireland | $CO_2$ | GEMINI-UK & DECC | 16 | July | 0.7 | 7.2 | 24.3 | 7b |
| UK + Ireland | $CH_4$ | GEMINI-UK & DECC | 16 | January | 0.2 | 1.1 | 10.0 | 7c |
| UK + Ireland | $CH_4$ | GEMINI-UK & DECC | 16 | July | 1.4 | 6.0 | 28.7 | 7d |
| UK + Ireland | $CO_2$ | GEMINI-UK & DECC | 18 | January | 0.1 | 0.8 | 7.6 | 8a |
| UK + Ireland | $CO_2$ | GEMINI-UK & DECC | 18 | July | 0.5 | 4.8 | 17.7 | 8b |
| UK + Ireland | $CH_4$ | GEMINI-UK & DECC | 18 | January | 0.1 | 0.8 | 7.2 | 8c |
| UK + Ireland | $CH_4$ | GEMINI-UK & DECC | 18 | July | 0.5 | 3.7 | 19.2 | 8d |

# Appendix A: Autonomy of GEMINI-UK

## A1  Software that Supports Automated Instrument Operation

We automate operation of the GEMINI-UK EM27/SUN instruments using a Python program called Pyra, developed at TU Munich (Dietrich et al., 2021). Pyra acts as a wrapper for the software included with the EM27/SUN that controls the spectrometer and the solar tracker components (OPUS and CamTracker, respectively), providing the means to start and stop them automatically when the solar zenith angle passes a threshold value. A detailed description of the current version of Pyra can be found in Aigner et al. (2023).

We use remote desktop software to monitor the running of the enclosures, allowing us to remotely troubleshoot any issues encountered with the software or the control components of the enclosure system.

## A2  EM27/SUN weatherproof enclosure

For GEMINI-UK we use a bespoke weatherproof enclosure for our EM27/SUN instruments, which we designed and built by Karn Scientific. Using these enclosures enables year-round deployment of the EM27/SUNs with minimal manual intervention associated with changes in weather conditions. Other EM27/SUN enclosures havebeen developed by academic institutions including TUM (Heinle and Chen, 2018), Heidelberg University (Knapp et al., 2021), University of Wollongong (Velazco et al., 2019), and LSCE (Lopez et al., 2025). Aside from the enclosure developed by Knapp et al. (2021), which is designed for ship deployment and includes additional fast slew mirrors and full waterproofing, these other enclosures are not fully weatherproof and expose the instrument to the local environment while observations are being taken. As of this year the LSCE enclosure design is being manufactured and distributed by eloneo (ELONEO -Instrumentation environnementale performante), however the other designs are not commercially available.

The enclosure consists of a fully-sealed Peli-branded IP67 rated case, optical dome assembly and protective cover, thermal management system, internal sub-frame for mounting the EM27/SUN, and power supply and control systems. The various components of the system can be seen in Figure A1. A photograph of a completed enclosure is shown in Figure A2. The unit has operational heritage with the UK's NERC Field Spectroscopy Facility (FSF), for whom the system was originally designed. FSF deployed three units during a long-term measurement campaign (April 2021–September 2022) around London where the enclosures experienced atmospheric temperatures ranging from -2°C to 31°C.

The optical dome assembly provides a fully weatherproof window with high transmission across the SWIR spectrum, through which EM27/SUN observations can be taken when environmental conditions are suitable, while protecting the instrument from dust, precipitation, and wildlife throughout the year. The dome itself is a spherical cap with surface diameter of 450 mm, cap height of 95 mm, and thickness of 5 mm, manufactured from BK7 glass. A witness sample of BK7 with matching curvature and thickness was tested during early prototyping. The spectral transmission was characterised using a Cary 5000 spectrophotometer, and the impact on the EM27/SUN's solar tracker was evaluated, and found to have no adverse affects on sun-tracking performance. The design of the optical dome assembly, consisting of the optical dome, interface plate, and clamping ring, enable observations at solar zenith angles as low as 80 degrees. Although the optical dome is fixed in position and

forms part of the weatherproof sealing envelope of the enclosure, a motorised retractable cover is used to protect the dome from dust and debris when observations are not being performed. The motorised cover operates autonomously based on the

output of a rain and light level sensor, closing when light levels drop below 2000 lux, or when precipitation is detected.

The enclosure thermal management system consists of two redundant pressure, temperature, and humidity sensors internally, up to four 30 W peltier coolers, and a 60 W internal resistive heater. The control system aims to maintain the enclosure internal temperature between the EM27/SUN operating limits with a safety margin. Dessicant is used to dry the internal air volume to around 20% RH. A 24V power supply is used to run all internal electronics, and mains connection is via a waterproof industrial

power connector. An Uninterruptible Power Supply is also provided, enabling short-term operation off a 12 Ah battery during a power outage or temporary disconnection of the mains supply. An industrially hardened microcontroller manages operation of the optical dome cover and thermal management system. The microcontroller reports data over a serial connection to a compact industrial PC, which runs the EM27/SUN operating software. Wi-Fi and ethernet connectivity is provided, and weatherproof HDMI and USB ports are included to allow ease of field installation and setup.

The enclosures for the GEMINI-UK project are being manufactured under license by the University of Leicester. Further information on the units can be found at www.karnscientific.com/em27-sun-weatherproof-enclosure.

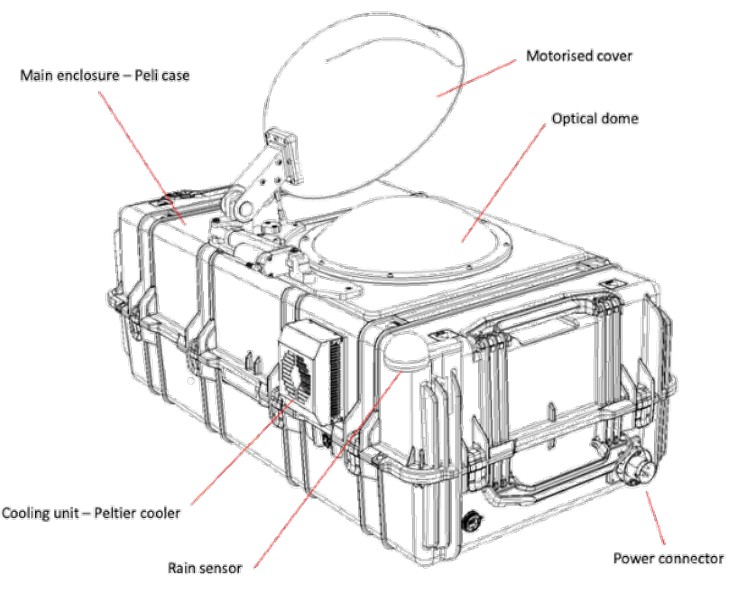

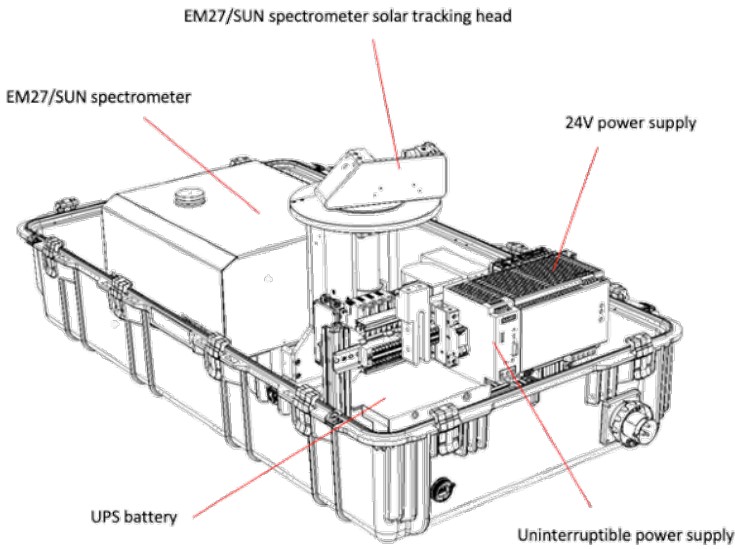

**Figure A1.** Computer Aided Design rendering of the EM27/SUN weatherproof enclosure. The top diagram shows the optical dome, motorised cover, Peltier cooler unit, rain sensor, and main power connector. The bottom diagram shows the enclosure with the casing lid removed, showing the installed EM27/SUN instrument, power supply, UPS, and UPS battery.

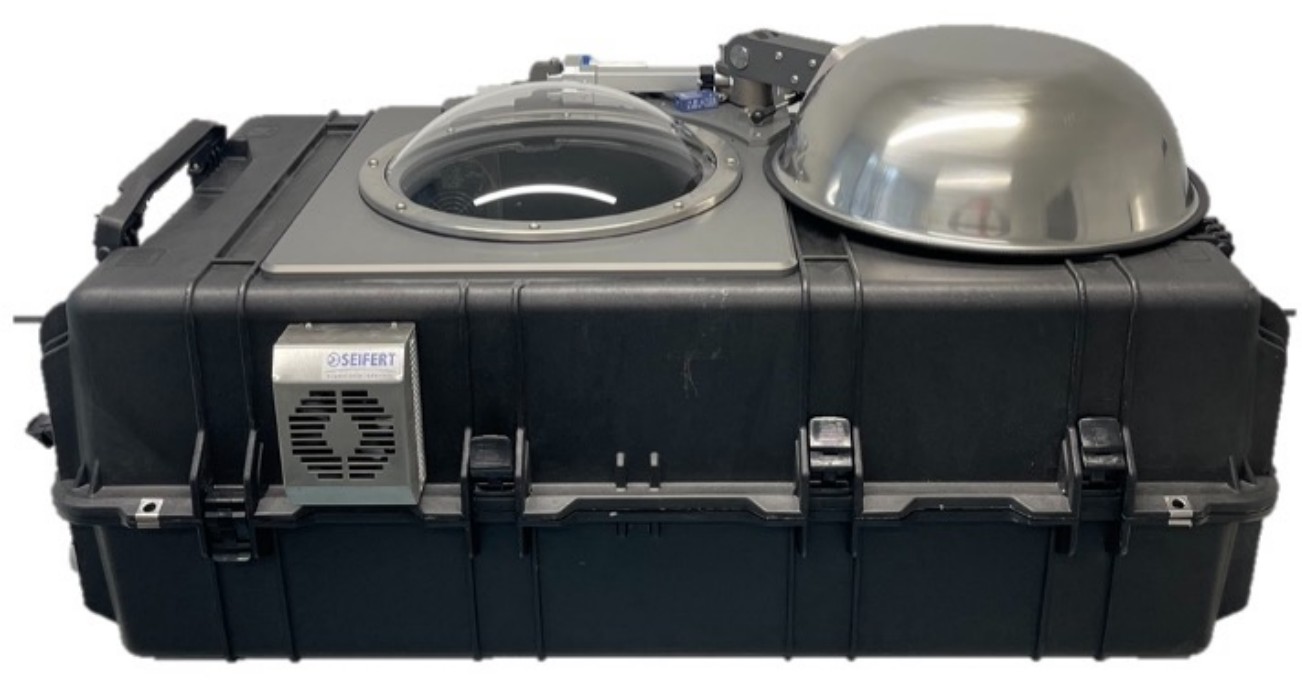

**Figure A2.** Photograph of a fully assembled EM27/SUN weatherproof enclosure.

# Appendix B: Supplementary Figures and Tables

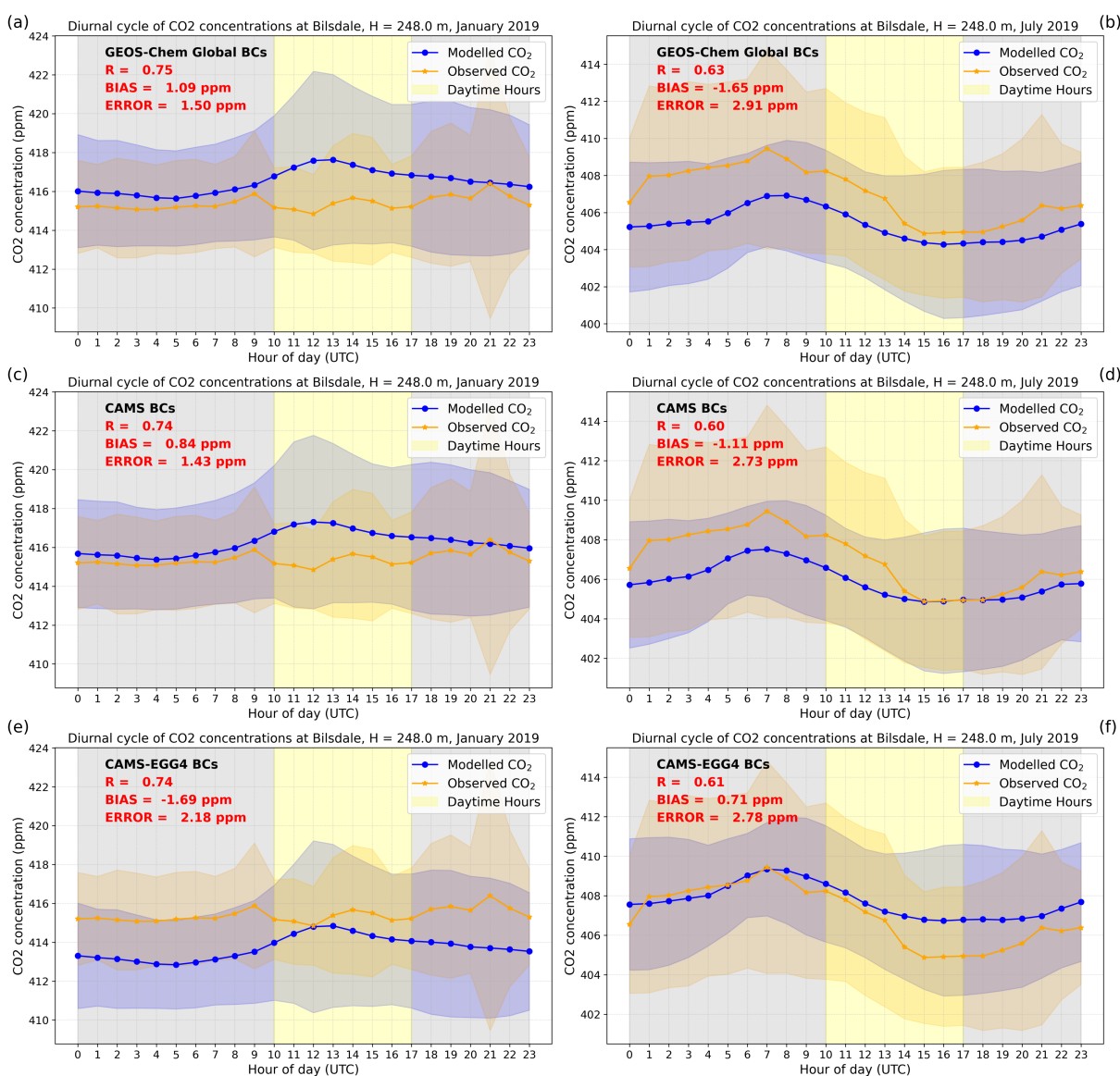

**Figure B1.** Mean diurnal cycle of observed (orange) and GEOS-Chem model (blue) $CO_2$ mole fraction values at Bilsdale for January ((**a**), (**c**) and (**e**)) and July ((**b**), (**d**) and (**f**)) 2019. Shaded envelopes denote the 1-$\sigma$ value about the mean value. The top, middle, and bottom rows denote results from using lateral boundary conditions for the nested GEOS-Chem model from the corresponding global GEOS-Chem simulation, the CAMS model that is constrained by *in situ* data, and the CAMS EGG4 model reanalysis that is constrained by satellite data, respectively. The model is sampled at the time and at the top inlet height of the tall tower. Daytime hours (10-17 UTC) are denoted by the vertically region highlighted in yellow when the DECC data are typically used for the inverse calculations.

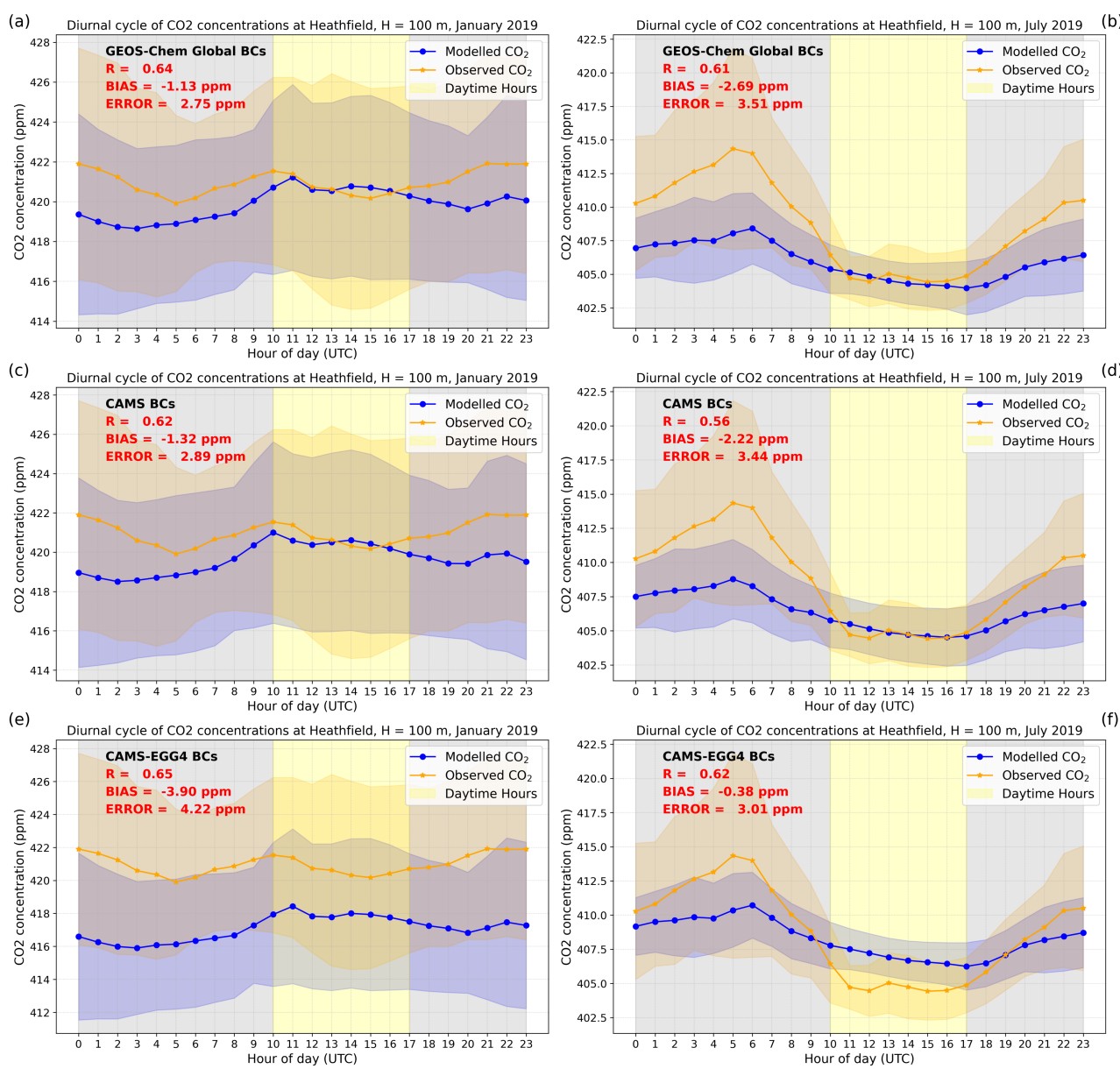

**Figure B2.** As Figure B1 but for Heathfield.

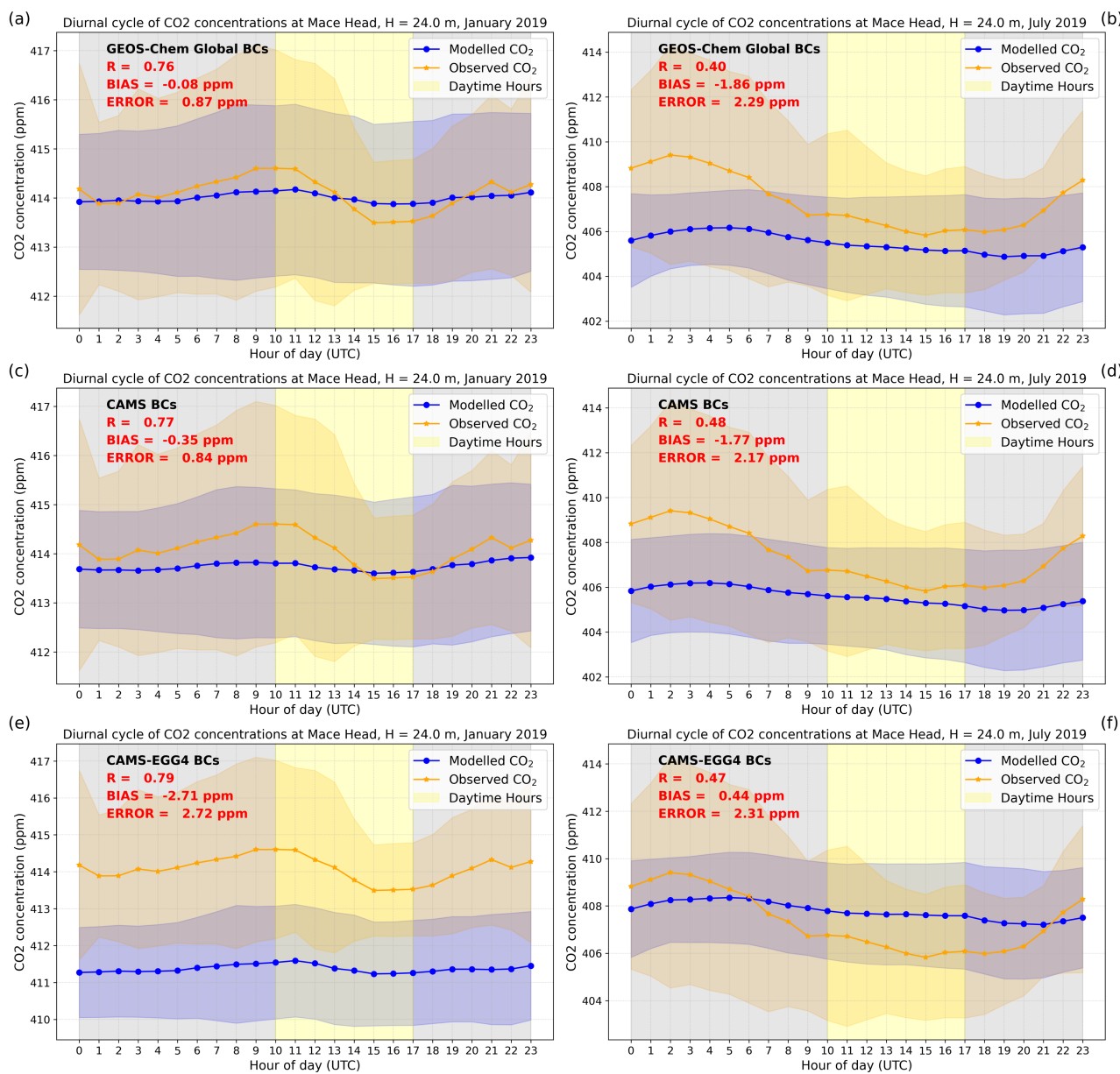

**Figure B3.** As Figure B1 but for Mace Head.

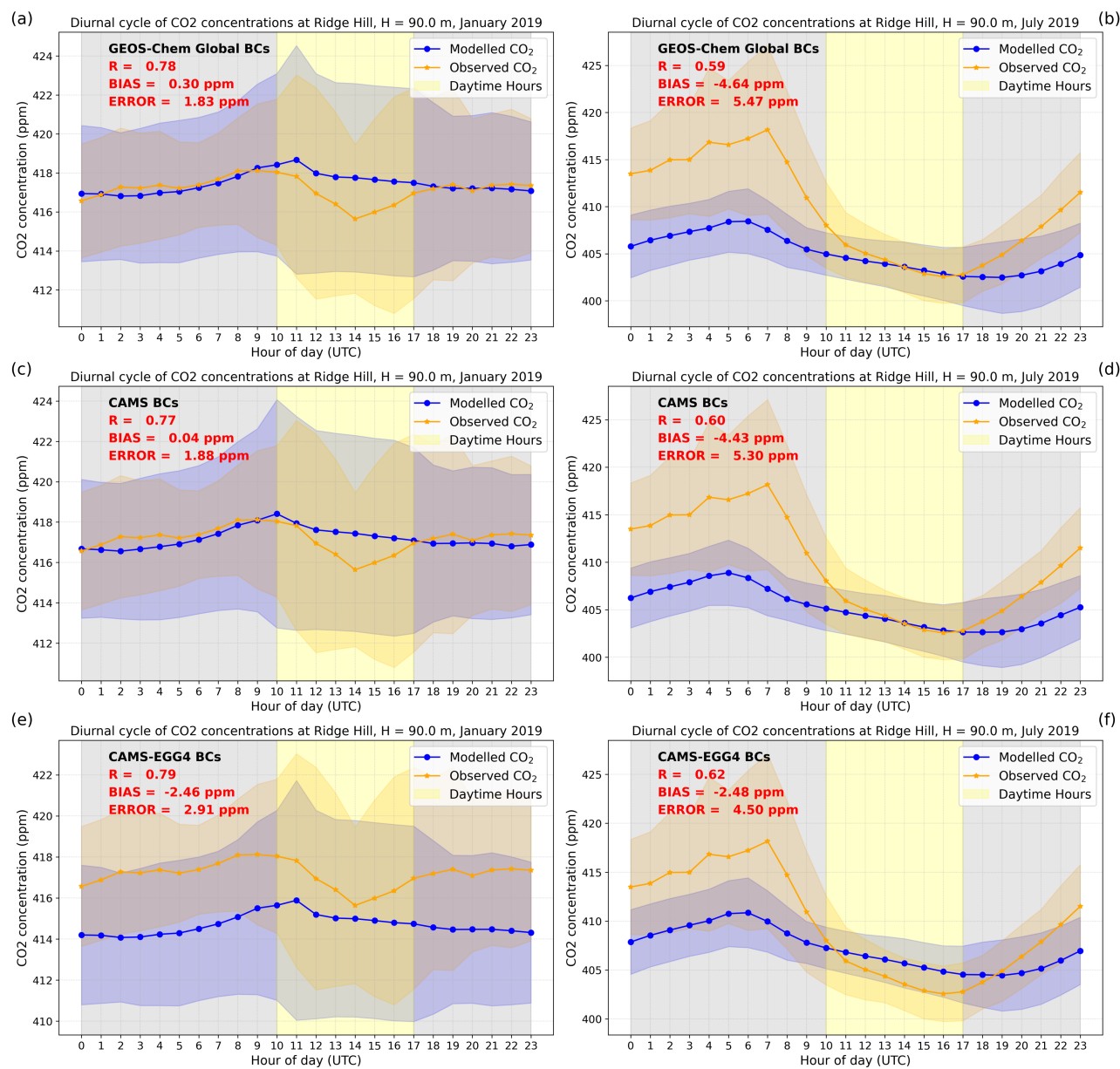

**Figure B4.** As Figure B1 but for Ridge Hill.

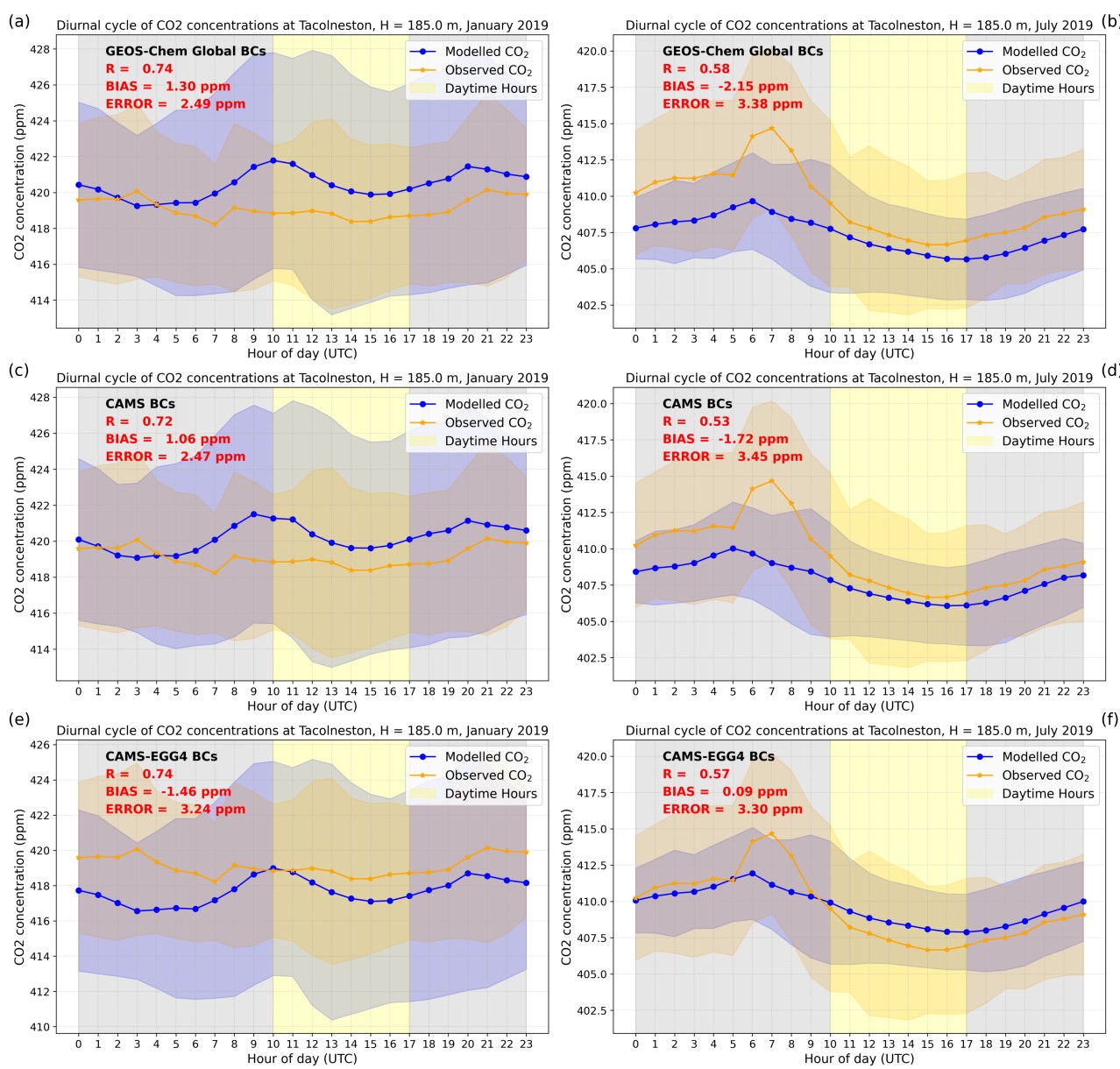

**Figure B5.** As Figure B1 but for Tacolneston.

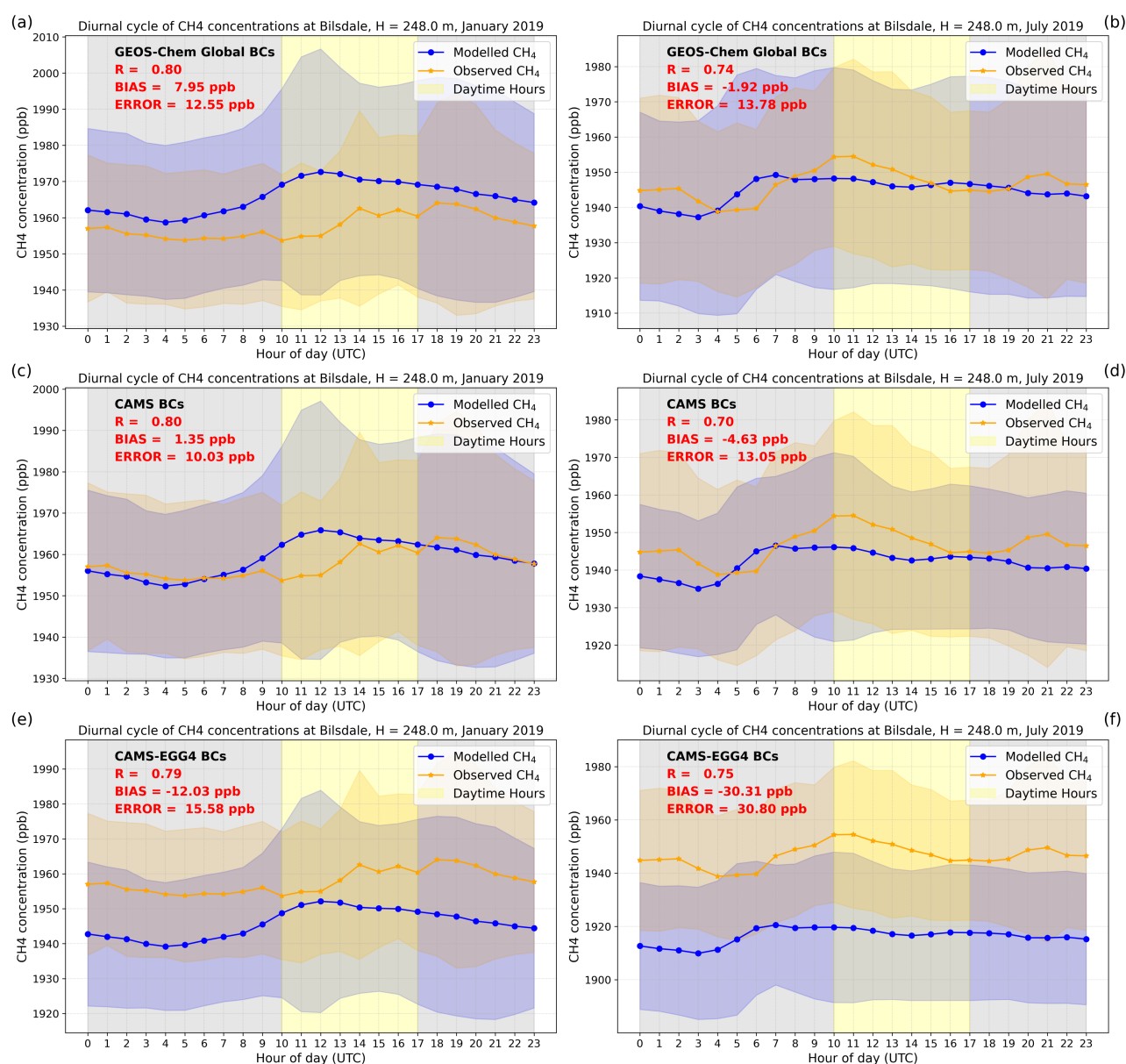

**Figure B6.** As Figure B1 but for methane.

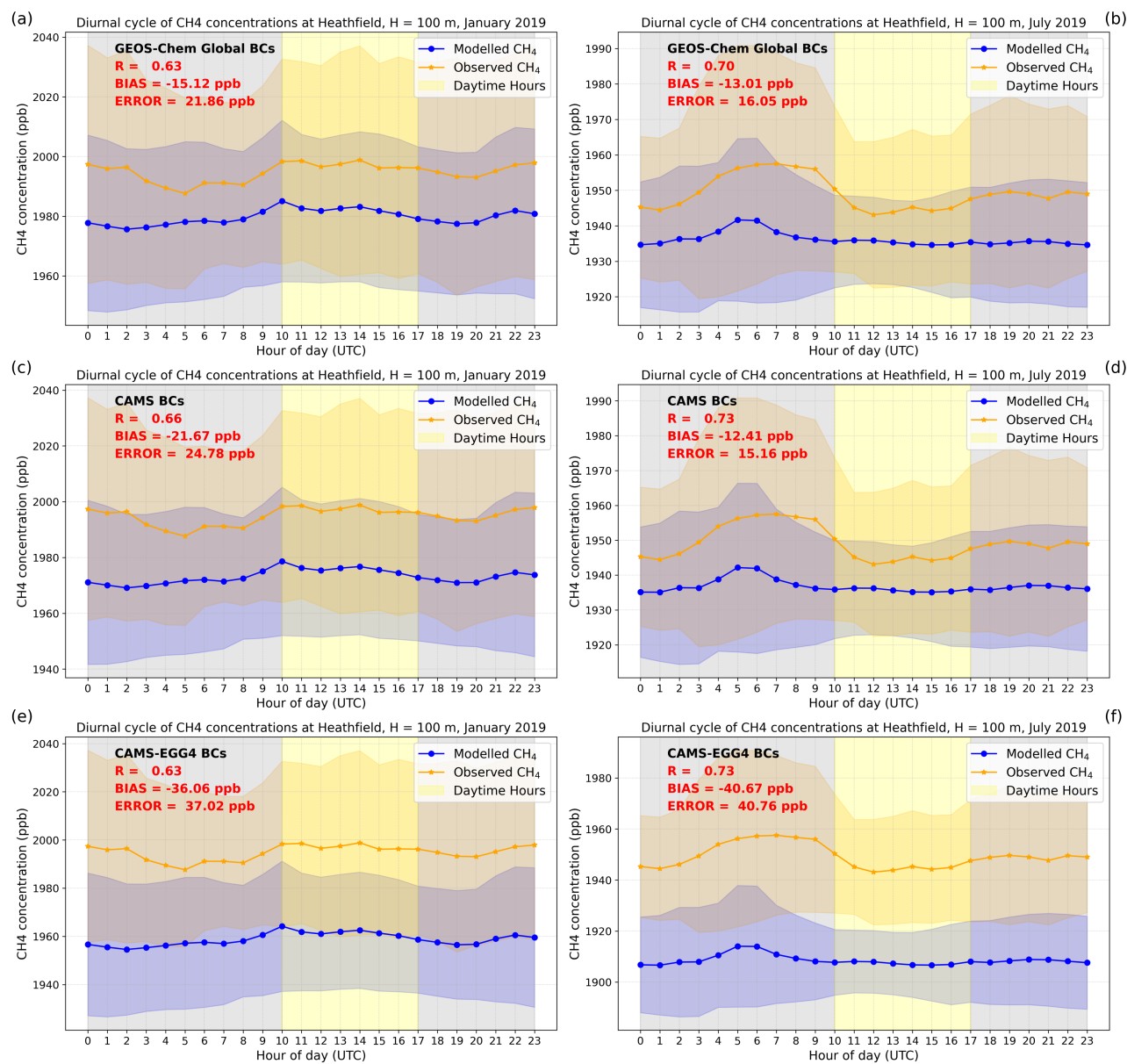

**Figure B7.** As Figure B2 but for methane.

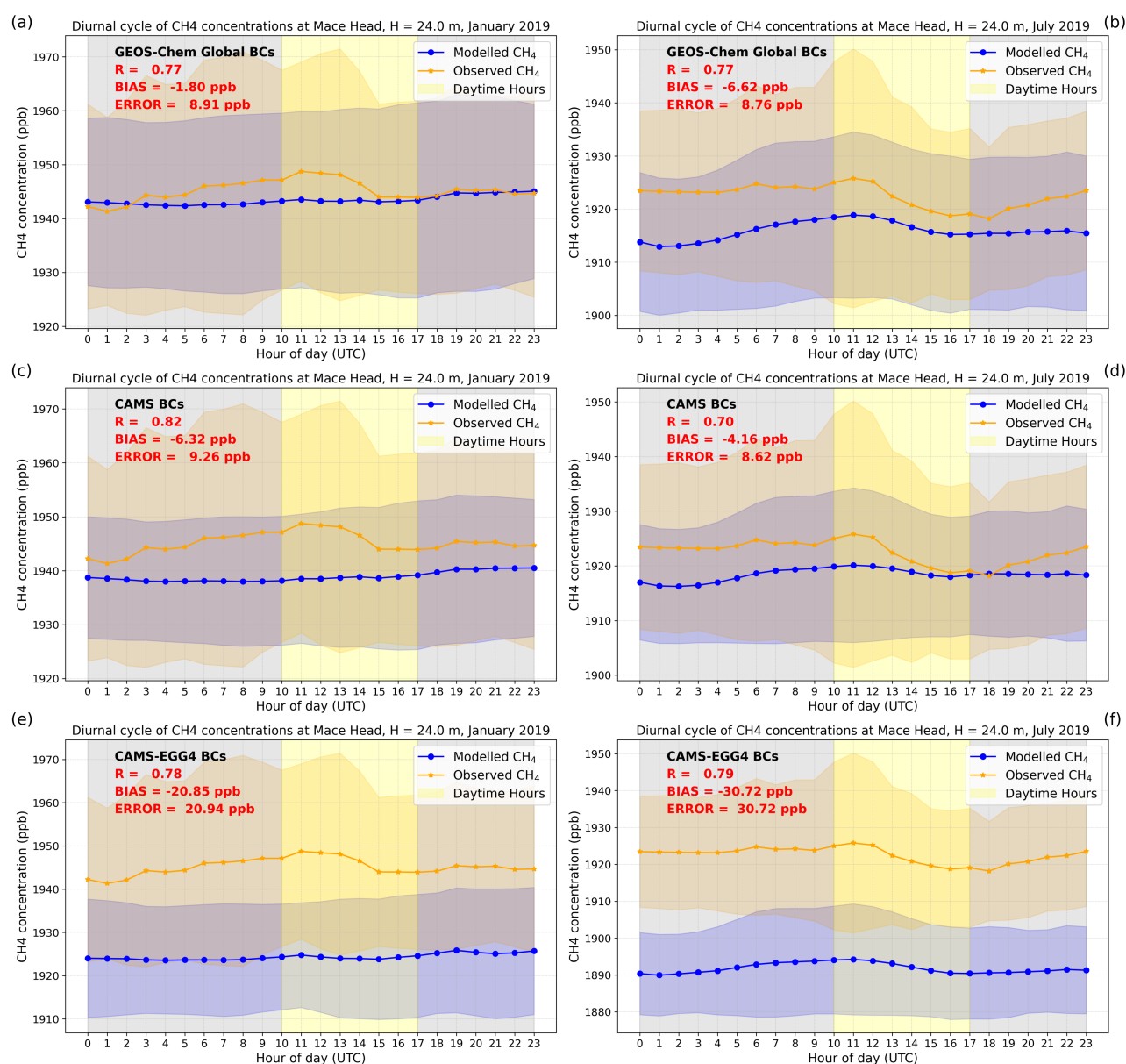

**Figure B8.** As Figure B3 but for methane.

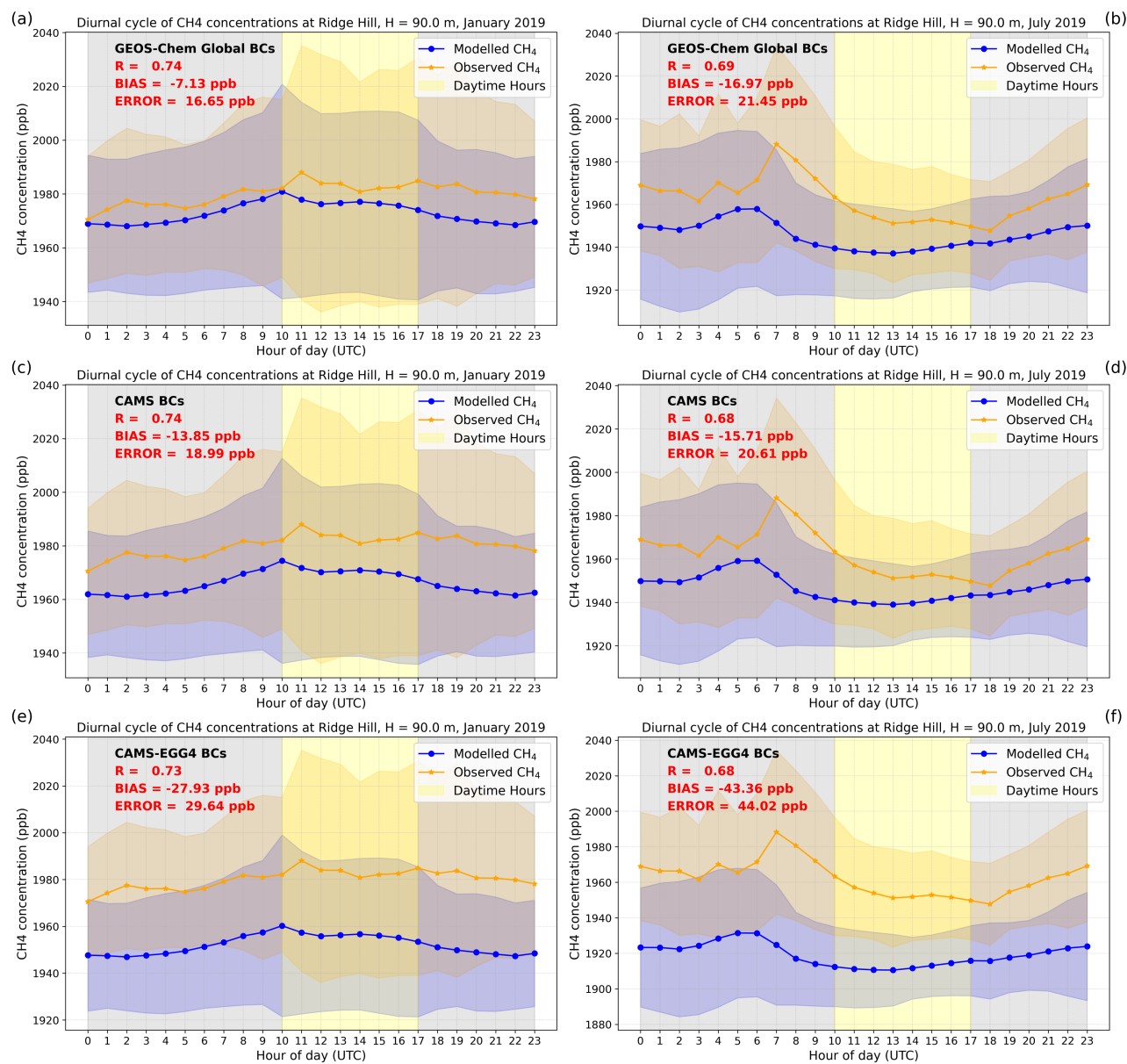

**Figure B9.** As Figure B4 but for methane.

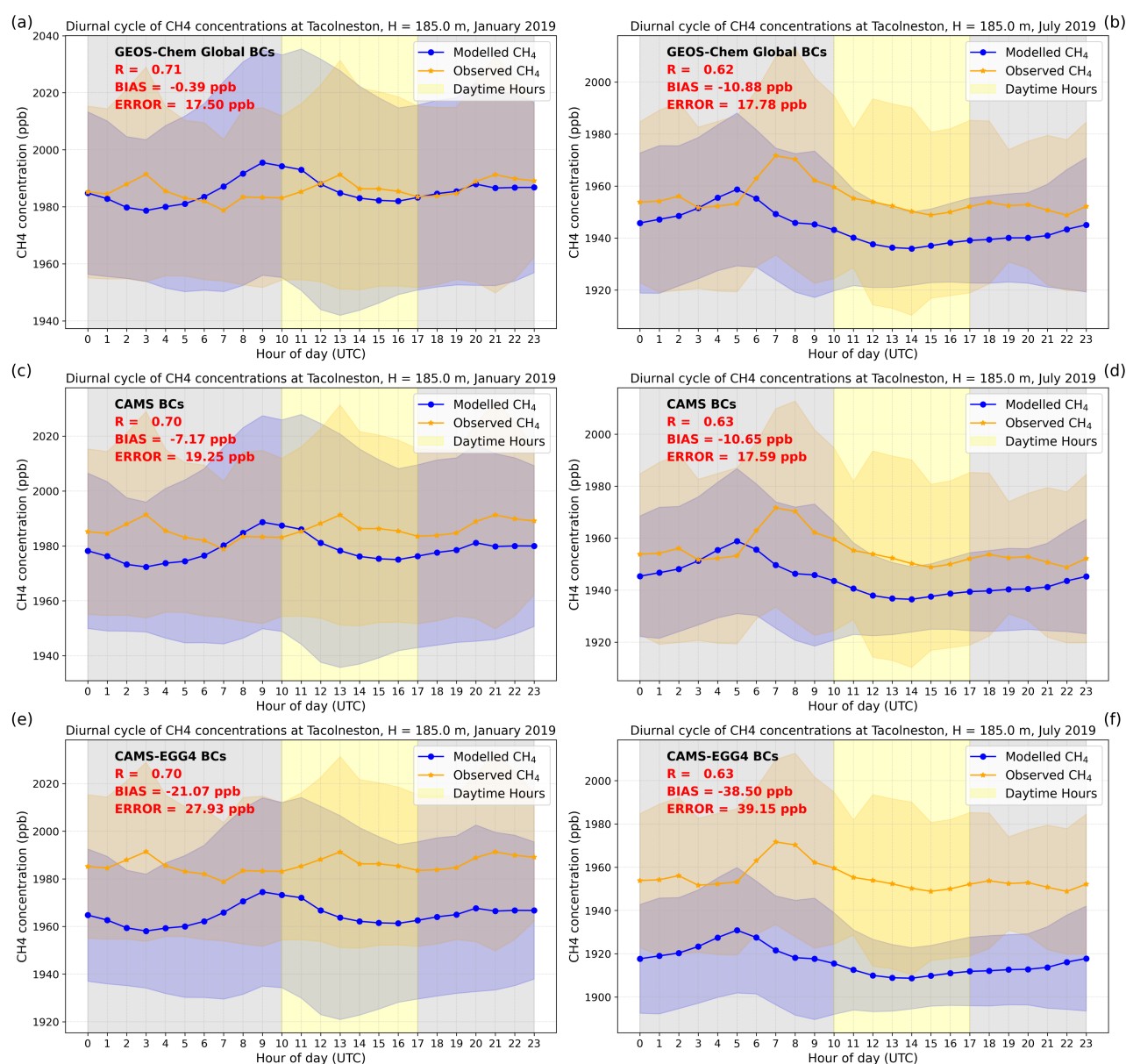

**Figure B10.** As Figure B5 but for methane.

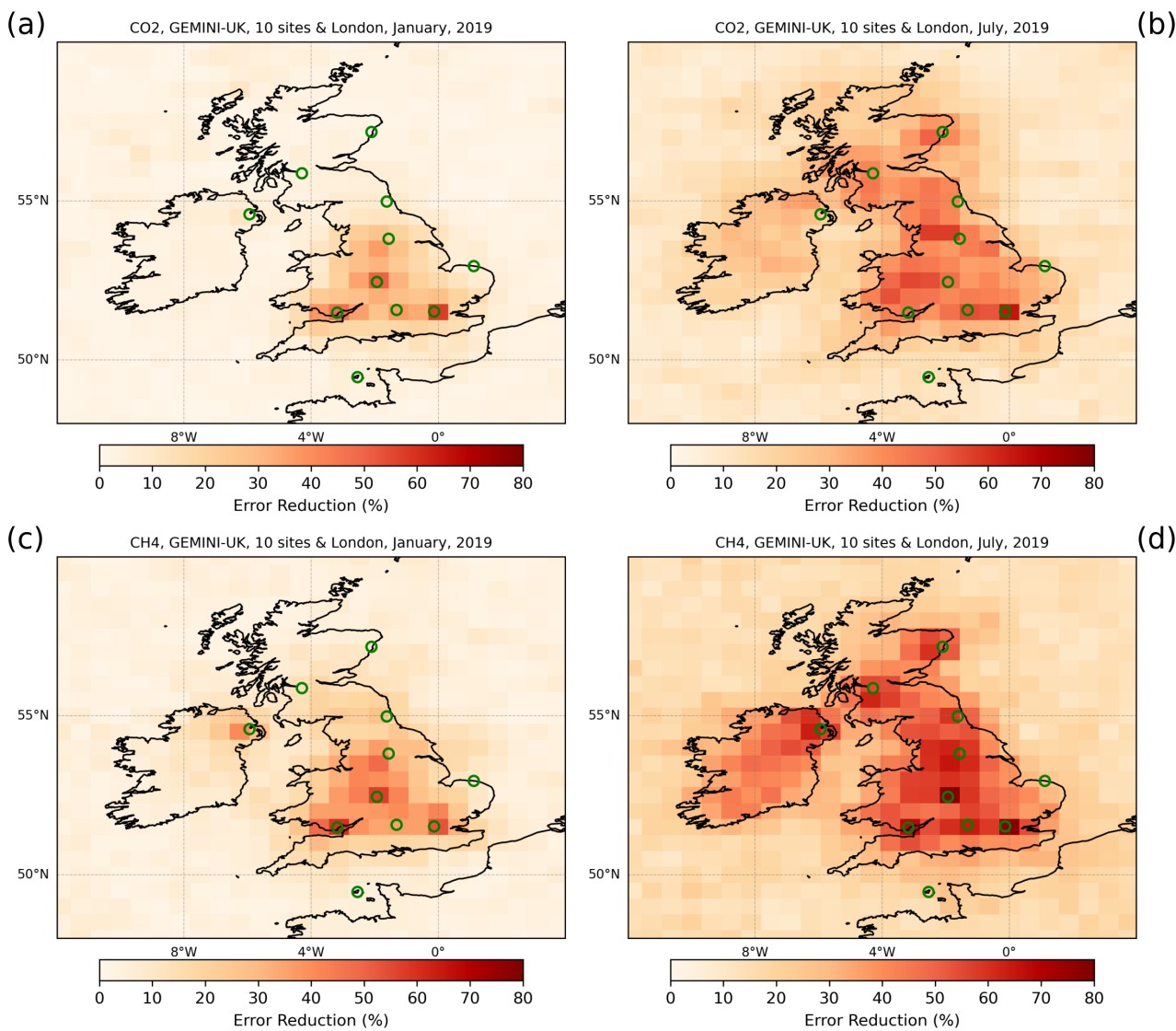

**Figure B11.** *A posteriori* error reduction of *a priori* net $CO_2$ ((**a**) and (**b**)) and methane ((**c**) and (**d**)) flux uncertainties from GEMINI-UK data (site locations: green circles) for the first 15-day assimilation window of January ((**a**) and (**c**)) and July ((**b**) and (**d**)) 2019. All results assume a transport error of 0.25 ppm for XCO2 and 2.5 ppb for XCH4.

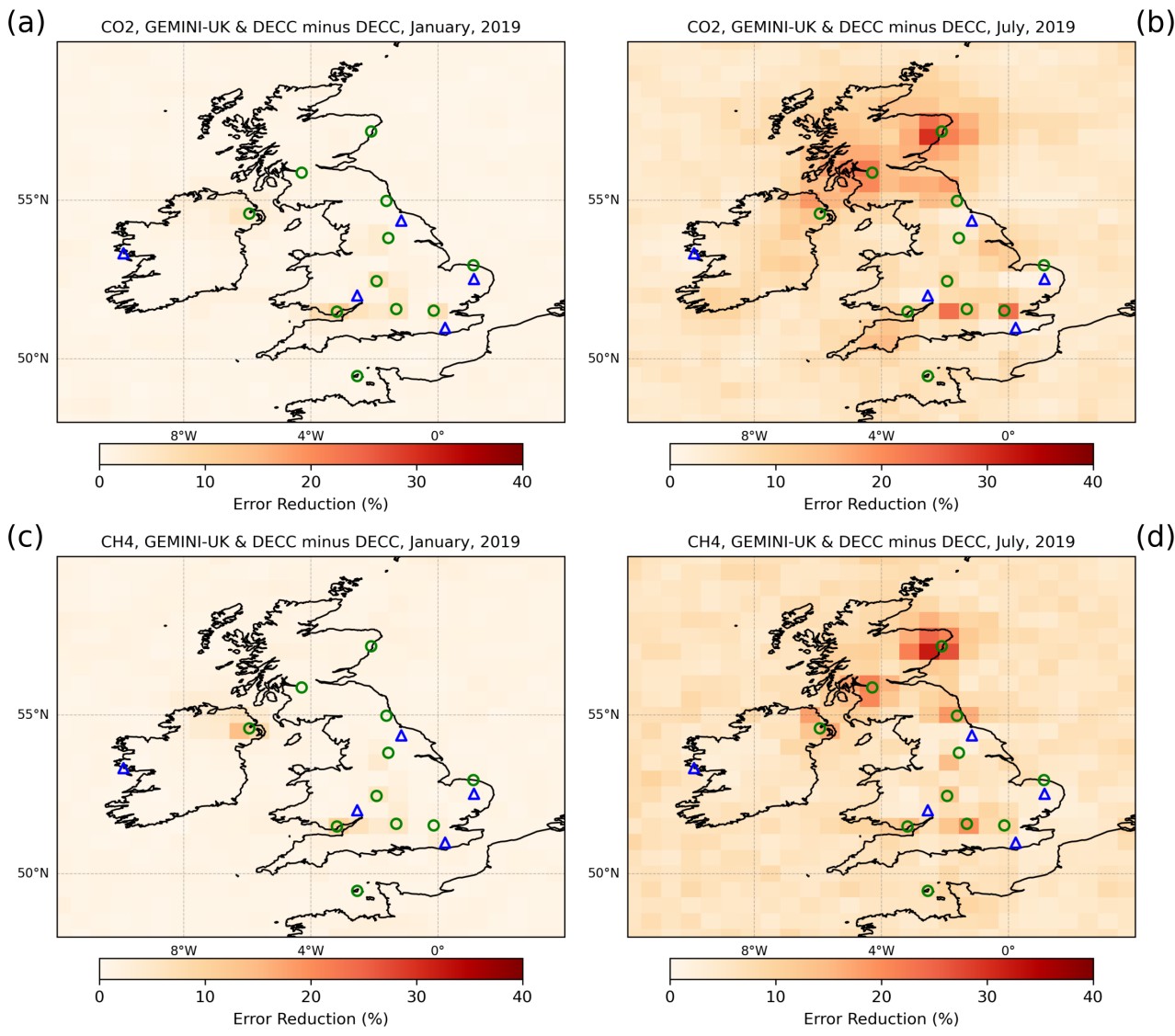

**Figure B12.** Same as Figure 6 but relative to the information already provided by the DECC tall tower data and Mace Head (site locations: blue triangles).

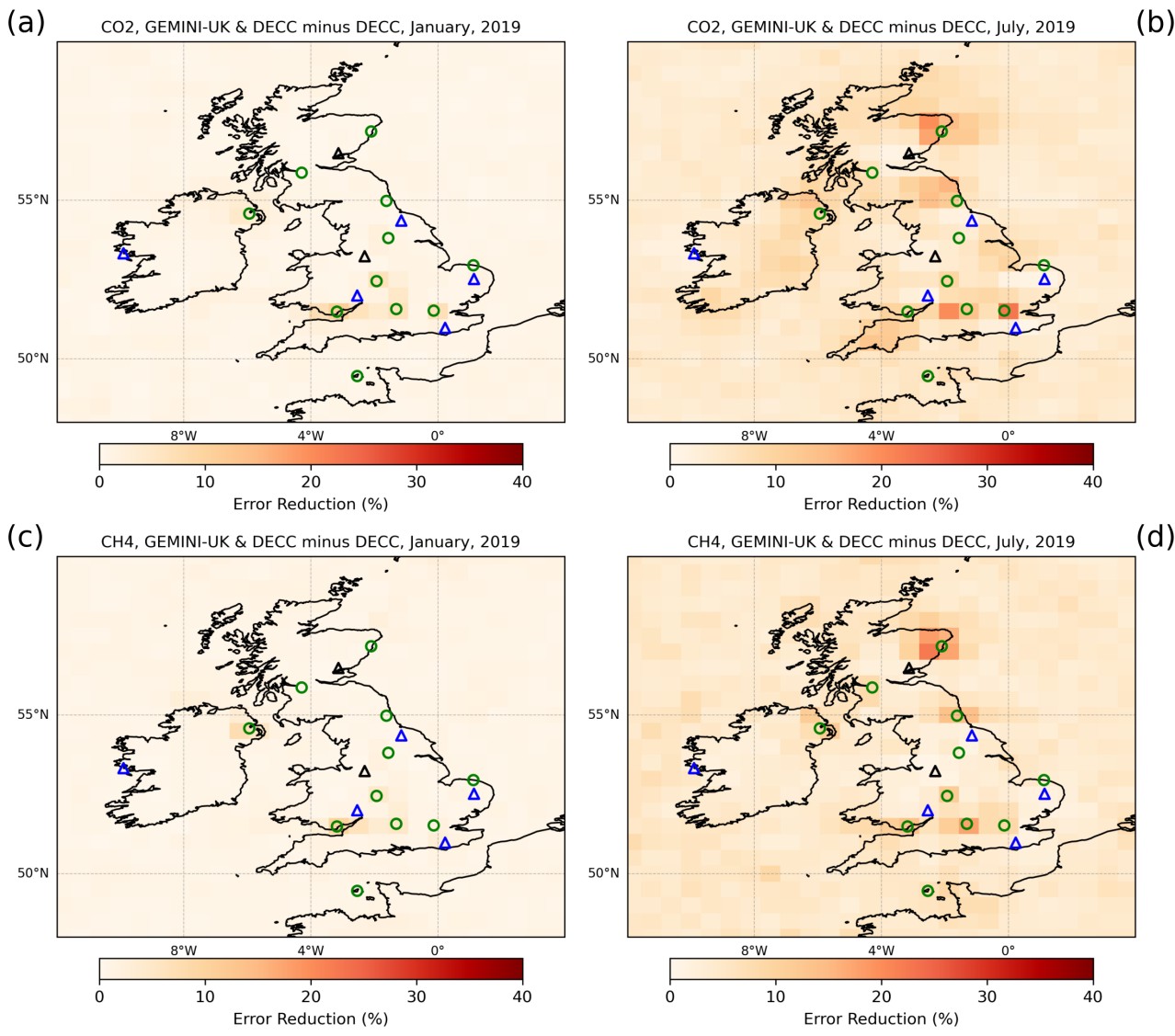

**Figure B13.** Same as Figure 7 but with the DECC tall tower data also including sites at Jodrell Bank and Invergowrie (site locations: black triangles) that will become operational in 2025.

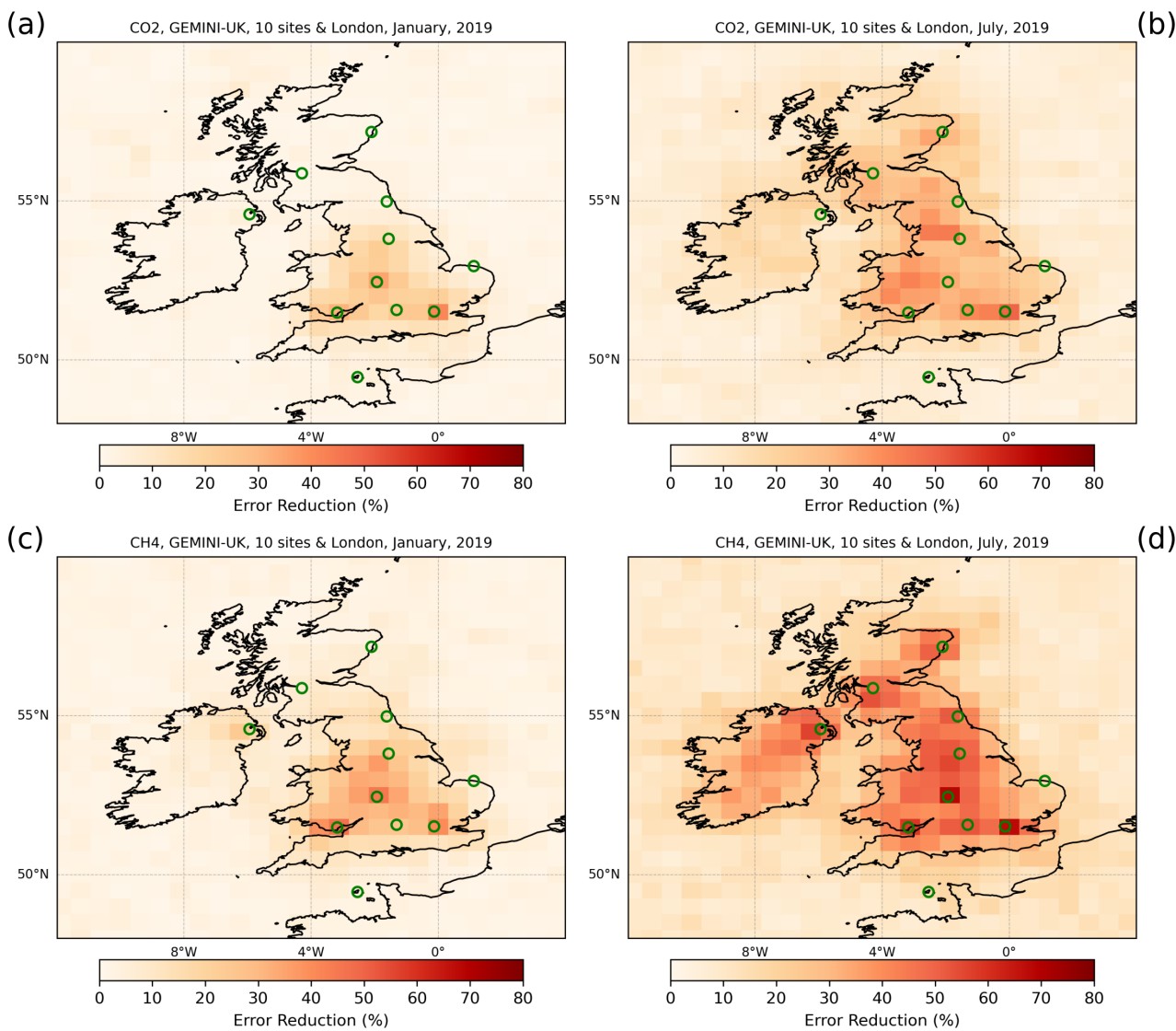

**Figure B14.** *A posteriori* error reduction of *a priori* net $CO_2$ ((**a**) and (**b**)) and methane ((**c**) and (**d**)) flux uncertainties from GEMINI-UK data (site locations: green circles) for the first 15-day assimilation window of January ((**a**) and (**c**)) and July ((**b**) and (**d**)) 2019. All results assume a transport error of 1.0 ppm for XCO2 and 10.0 ppb for XCH4.

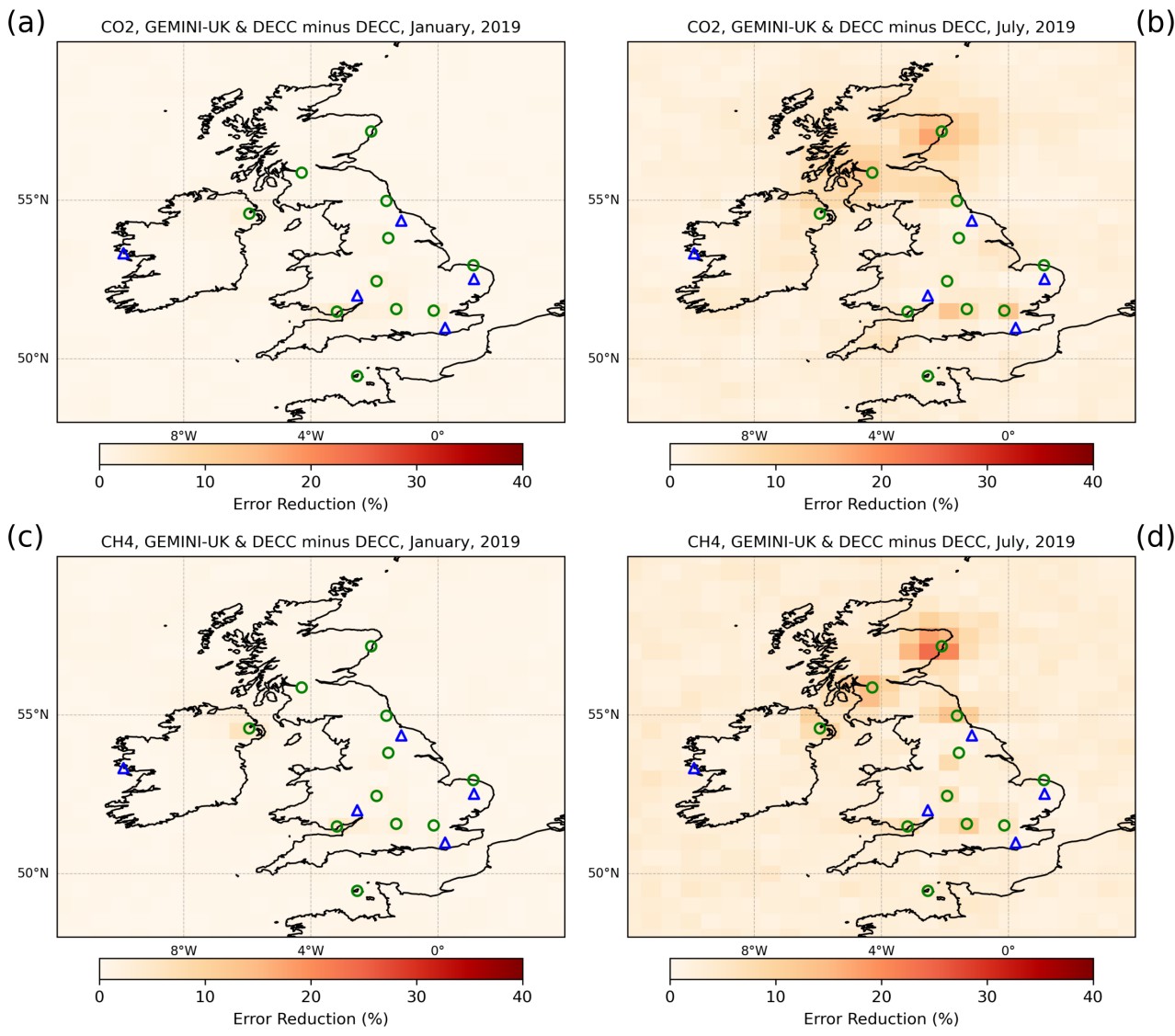

**Figure B15.** Same as Figure 6 but relative to the information already provided by the DECC tall tower data and Mace Head (site locations: blue triangles).

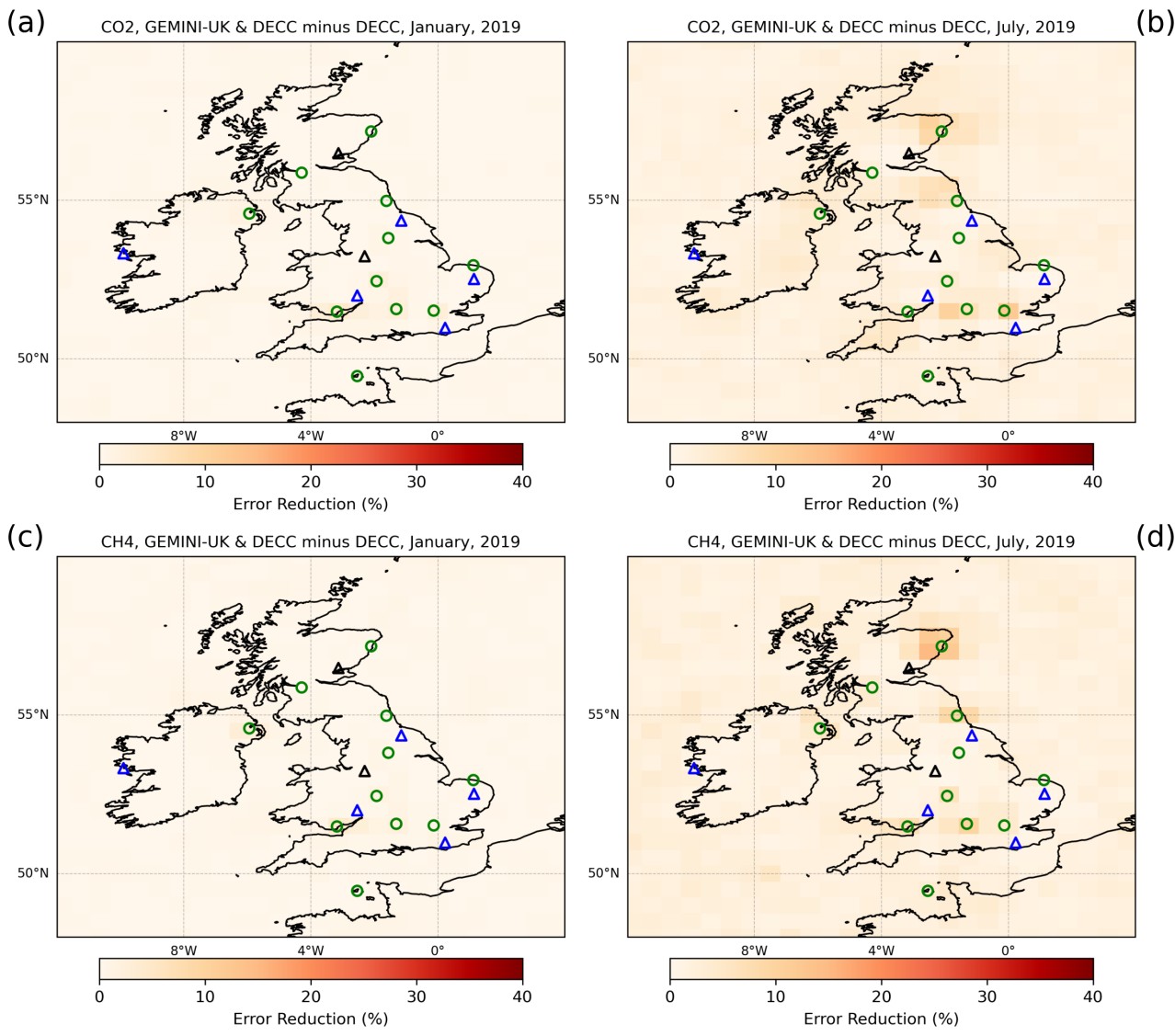

**Figure B16.** Same as Figure 7 but with the DECC tall tower data also including sites at Jodrell Bank and Invergowrie (site locations: black triangles) that will become operational in 2025.

**Table B1.** Same as Table 2 but for an atmospheric transport model error of 0.25 ppm for $XCO_2$ and 2.5 ppb for $XCH_4$.

| Region | Spec | Network | # sites | Month | Min (%) | Mean (%) | Max (%) | Figure |
|---|---|---|---|---|---|---|---|---|
| UK + Ireland | $CO_2$ | GEMINI-UK | 11 | January | 1.1 | 11.3 | 57.8 | B11a |
| UK + Ireland | $CO_2$ | GEMINI-UK | 11 | July | 10.7 | 32.6 | 65.6 | B11b |
| UK + Ireland | $CH_4$ | GEMINI-UK | 11 | January | 1.7 | 18.2 | 60.3 | B11c |
| UK + Ireland | $CH_4$ | GEMINI-UK | 11 | July | 12.3 | 42.7 | 78.4 | B11d |
| UK + Ireland | $CO_2$ | GEMINI-UK & DECC | 16 | January | 0.3 | 1.5 | 11.4 | B12a |
| UK + Ireland | $CO_2$ | GEMINI-UK & DECC | 16 | July | 0.9 | 9.7 | 29.9 | B12b |
| UK + Ireland | $CH_4$ | GEMINI-UK & DECC | 16 | January | 0.3 | 1.6 | 13.9 | B12c |
| UK + Ireland | $CH_4$ | GEMINI-UK & DECC | 16 | July | 2.0 | 7.7 | 32.1 | B12d |
| UK + Ireland | $CO_2$ | GEMINI-UK & DECC | 18 | January | 0.1 | 1.2 | 10.9 | B13a |
| UK + Ireland | $CO_2$ | GEMINI-UK & DECC | 18 | July | 0.7 | 6.7 | 22.7 | B13b |
| UK + Ireland | $CH_4$ | GEMINI-UK & DECC | 18 | January | 0.2 | 1.1 | 10.0 | B13c |
| UK + Ireland | $CH_4$ | GEMINI-UK & DECC | 18 | July | 0.8 | 5.0 | 22.4 | B13d |

**Table B2.** Same as Table 2 but for an atmospheric transport model error of 1.0 ppm for $XCO_2$ and 10.0 ppb for $XCH_4$.

| Region | Spec | Network | # sites | Month | Min (%) | Mean (%) | Max (%) | Figure |
|---|---|---|---|---|---|---|---|---|
| UK + Ireland | $CO_2$ | GEMINI-UK | 11 | January | 0.4 | 7.3 | 41.5 | B14a |
| UK + Ireland | $CO_2$ | GEMINI-UK | 11 | July | 6.1 | 22.4 | 49.7 | B14b |
| UK + Ireland | $CH_4$ | GEMINI-UK | 11 | January | 0.8 | 12.9 | 47.1 | B14c |
| UK + Ireland | $CH_4$ | GEMINI-UK | 11 | July | 7.5 | 34.3 | 69.1 | B14d |
| UK + Ireland | $CO_2$ | GEMINI-UK & DECC | 16 | January | 0.1 | 0.6 | 4.7 | B15a |
| UK + Ireland | $CO_2$ | GEMINI-UK & DECC | 16 | July | 0.5 | 4.7 | 17.4 | B15b |
| UK + Ireland | $CH_4$ | GEMINI-UK & DECC | 16 | January | 0.1 | 0.6 | 6.1 | B15c |
| UK + Ireland | $CH_4$ | GEMINI-UK & DECC | 16 | July | 0.9 | 4.2 | 24.0 | B15d |
| UK + Ireland | $CO_2$ | GEMINI-UK & DECC | 18 | January | 0.1 | 0.5 | 4.5 | B16a |
| UK + Ireland | $CO_2$ | GEMINI-UK & DECC | 18 | July | 0.3 | 3.0 | 12.0 | B16b |
| UK + Ireland | $CH_4$ | GEMINI-UK & DECC | 18 | January | 0.1 | 0.4 | 4.4 | B16c |
| UK + Ireland | $CH_4$ | GEMINI-UK & DECC | 18 | July | 0.3 | 2.5 | 15.0 | B16d |