# Peer review of "The Greenhouse gas Emission Monitoring network to Inform Net-zero Initiatives UK (GEMINI-UK): network design, theoretical performance, and initial data"

_EGUsphere, 2025_

## Referee Comment (RC1)

This paper presents GEMINI-UK sensor network and its network design, which consists of 10 EM27/SUNs. The authors have run numerical simulations to assess the potential error reduction given by incorporating the sensor network. However, the study is very theoretical and can be achieved by only tuning the parameters of the inversion. Therefore, a major revision is recommended. The paper can be published after considering the following comments.

**Major comments:**

1. The Authors assess the theoretical performance of GEMINI-UK only via the relative error reduction via prior and posterior error. You don't discuss any other uncertainties which will influence the performance/results.

2. The authors just mention briefly the uncertainty assumptions (transport error, measurement error in line 273 and prior error in line 230). All the error reduction values in chapter 3.3 and figure 6-8 (and in the abstract) depend strongly on those uncertainty assumptions (mainly the measurement/transport error, R).

3. Chapter 3.1 the statistical analysis is not quite clear, hard to understand the numbers, also a repetition of the numbers in line 367. Write it clearer or include a table. Explain what the ERROR and BIAS is (in figure 2/3), either in the figure description or in the text. In the text you write the biases in ppm/ppb and in the figure in %. Line 369 also a repetition of 361. In figure 2/3, it would be easier to understand if you write in the title what is the difference between the three figures (the boundary conditions). The chapter could be improved to make the statistics clearer and easier to understand, both in the text and in the figures.

4. Throughout the manuscript it is written as "CO2 and methane" rather than CO2 and CH4 or carbondioxide and methane. If there is no specific reason for this, I recommend to change all appearances to either one of the two mentioned options.

5. Line 273: transport errors of 3ppm and 15ppb. How did you select these values? What is the reason for choosing exactly these numbers?

6. Line274: scene dependent observational error: Can you please specify the numbers or give examples of these scene dependent standard deviations and explain the difference to the theoretical limit of the EM27/SUN or other studies? For example, Jones et al. 2021 (https://acp.copernicus.org/articles/21/13131/2021/) assumed the uncertainty to be 0.2 ppb based on the Allan analysis to the measured column difference when the instruments were co-located (Chen et al. 2016, https://acp.copernicus.org/articles/16/8479/2016/).

7. Line 415: whole section 3.4: It is not quite clear why you do this comparison and why it is relevant here? There are existing EM27/SUN vs. TCCON comparisons. What is the benefit of comparing them in your setup, in particular because you mention that the comparison has its weaknesses of different spectral resolutions, as you mention

in line 431. To my knowledge, it is possible to imitate EM27/SUN recordings with an HR125 by reducing the spectral resolution with the HR125. This was not considered here, why?

8. Line 432: Does this mean, that you used different profiles for TCCON and EM27/SUN retrievals? If so, why?

9. Line 442: Can you please specify "higher and further"?

10. Figure 6: It is not quite clear what the units of error reduction are. Please specify. The "chessboard" pattern further implies a high spatial resolution. This would further imply that single grid cells are affected by error reduction through additional observations differently in small spatial scales. However, it appears to be noise. The amount of additional EM27/SUN observations does most likely not influence single grid cells given the spatial resolution you show here. Why are you sure that this is not noise but a true error reduction? It would be helpful to see the simulated EM27/SUN observations in a plot to make further assumptions.

11. Figure 9: The colorbar is capped for a lot of data points. It is difficult to estimate the relevance of certain areas/pixels given that the surrounding has the same value. Further there are pixels capping the colorbar as maximum positive flux in winter but also cap as maximum negative flux in summer (e.g. around Dublin). If there is no other physically relevant explanation, this might point towards an overfitting of these regions. Can you exclude overfitting for your setup and why?

12. A more careful color classification would increase the information content. It is not possible to distinguish flux values within certain areas (e.g. almost the whole of Ireland for CH4 in summer).

13. Similar to figure 6 it should be made clear, to what degree the visualisation might be noise or related to noise and what content of the figure is most likely based on true signal. Similar to figure 6, the pixels seem to mimic a much higher spatial resolution than the setup is being capable of resolving. A better visualisation of single regions including some visual interpretation of the noise or the errors would give a better understanding of the scenes.

14. Line 172-174: The data from the ICOS network already exists and is public. Furthermore, what would you use this data for

15. Line 182-185: Why are the excluded sectors are not relevant/not included.

16. Line 188-189: The oceans, especially coastal areas are also a large methane source. Since at least half of the modelling domain is ocean including a lot of coasts, this should be considered. https://www.geomar.de/en/news/article/how-much-methane-is-released-by-the-ocean

17. Line 344: Optimal regarding that the network-sensitivity should be the same everywhere? How do you know whether it is optimal - did you brute-force it or use some optimization algorithm?

18. Line 396: Have these apriori estimates been introduced in the paper? The inventories for the GEOS-CHEM simulation have been introduced, but if they are the synthetic "true" emissions, what is the prior?

19. Line 407-408: It would be good to have this additional error reduction as an average over the UK land area not just their maximum values.

20. Line 413-414: How does this statement connect to the figure. It just shows one posterior, not the improvement of the posterior.

21. Line 441-442: The optimality criterion for the sensor placement you mentioned was the sensitivity coverage. Is that equivalent?

22. Line 444-445: These are the values if there were not other measurements. You should use the values from figure 8. Please also adjust the numbers in the abstract.

23. Figure 2: The definition of R, bias and error should be mentioned in the text. Are they calculated only using the daytime hours?

24. Figure 2: It would be good to see these fits at the different sites to get a better idea of where each boundary conditions performs better.

**Minor comments:**

1. 144: instrumentsorldwide

2. 190: described **by** Deng

3. 195: from

4. 197: 2° x 2.5° make the space consistent (e.g. 201)

5. 201:

6. 213: methane**.** These

7. 279:

8. 286: July instead of June

9. Line 81: "EM27/SUN"

10. Line 87: You should cite TCCON here since it is the first mentioning of it

11. Line 91: "in situ" or "in-situ"? The text uses both variants

12. Line 111: You could also cite https://doi.org/10.5194/amt-9-2303-2016

13. Line 118: You could consider to also cite the PROFFAST Software release itself. This is how the PROFFASTpylot cited it: https://gitlab.eudat.eu/coccon-kit/proffastpylot/-/blob/joss/paper.bib?ref_type=heads#L234-L246. Maybe they have a more updated way of citing it by now.

14. Line 122: https://hitran.org/citepolicy/

15. Line 144: "instruments worldwide"

16. Line 168: The model has not been introduced at this point

17. Line 177-178: What is the nesting? Is there an inner domain? Mention Figure 1b here again.

18. Line 182: Include the "CAMS-REG-v4" name in the text.

19. Line 189: Lateral exchange from where to where?

20. Line 197-198: This should be mentioned before, when presenting the "nested model".

21. Line 213: Dot is missing.

22. Line 326: How long is the embargo?

23. Line 462: New technologies like?

24. Figure 6: Where does this checkered pattern come from? It would be helpful to have the Gemini-UK locations plotted here as well.

25. Abstract line 10: these … network. Shouldn't it be this network?

26. Line 46: you could consider to cite also Bayesian inversion studies that incorporate EM27/SUN measurements, e.g.:
https://acp.copernicus.org/articles/21/13131/2021/
https://acp.copernicus.org/articles/23/6897/2023/

27. Line 388: what is considered an observation here? 15min averages or 1min averages? What is your single measurement sample frequency of the EM27/SUN?

28. Line 421: You mention TCCON data is shown but it is only 3 rows later that you explain where it is shown. This is confusing.

29. Line 428: two times "within"

30. Appendix A2: For readers to get more context it could be helpful to briefly mention existing enclosure designs, e.g. https://amt.copernicus.org/articles/11/2173/2018/

---

## Author Comment (AC1)

**Response to referee comments on *"The Greenhouse gas Emission Monitoring network to Inform Net-zero Initiatives UK (GEMINI-UK): network design, theoretical performance, and initial data"***

**by**

**Kurganskiy et al.**

We thank both referees for their constructive comments, which have led to improvements in the original manuscript. The referee comments are included in this document in black, with our responses highlighted in blue. Where changes to the manuscript text are described, we show the revised text in *blue italics*.

**Anonymous Referee #1**

This paper presents GEMINI-UK sensor network and its network design, which consists of 10 EM27/SUNs. The authors have run numerical simulations to assess the potential error reduction given by incorporating the sensor network. However, the study is very theoretical and can be achieved by only tuning the parameters of the inversion. Therefore, a major revision is recommended. The paper can be published after considering the following comments.

**Major comments:**

1. The Authors assess the theoretical performance of GEMINI-UK only via the relative error reduction via prior and posterior error. You don't discuss any other uncertainties which will influence the performance/results.

    The reviewer is correct: using OSSEs we assess the theoretical performance of the GEMINI-UK system via the relative error reduction between the prior and posterior error covariance matrices. We adopt a lightweight approach, but nevertheless it allows us to evaluate the potential constraints provided by the observing system. However, we acknowledge that other important sources of uncertainty are not fully captured, including systematic errors. These limitations are inherent in the OSSE design, where the same model is used to generate the synthetic truth and perform the error reduction calculations by perturbing the prior fluxes and applying LETKF.

    To address this reviewer comment, we provide in the revised manuscript a more comprehensive discussion of random and systematic errors that may affect the performance of the measurement system to estimate regional flux estimates for CO2 and methane.

2. The authors just mention briefly the uncertainty assumptions (transport error, measurement error in line 273 and prior error in line 230). All the error reduction values in chapter 3.3 and figure 6-8 (and in the abstract) depend strongly on those uncertainty assumptions (mainly the measurement/transport error, R).

We agree with this reviewer that our results, including the error reduction values shown in Figures 6-8 and discussed in Section 3.3, are sensitive to the uncertainty assumptions (mainly the measurement/transport error, described in the R matrix).

For our baseline calculations based on the perturbed runs, we used an atmospheric transport error of 0.5 ppm for XCO2, corresponding to the middle of the range quantified by McNorton et al. (2020, GMD), and 5 ppb for XCH4, consistent with the bias estimates reported by Stanevich et al. (2021, ACP). To assess the robustness of our results and to address this reviewer comment, we have now performed additional sensitivity tests for which we assume an atmospheric transport uncertainty that is a factor of 2 lower and higher than the baseline values (0.25 and 1.0 ppm for XCO2; 2.5 and 10 ppb for XCH4). The results are shown in Figures A3-A8 and confirm that while the absolute error reduction values vary with the assumed atmospheric transport errors, the spatial patterns and qualitative conclusions remain robust.

3. Chapter 3.1 the statistical analysis is not quite clear, hard to understand the numbers, also a repetition of the numbers in line 367. Write it clearer or include a table. Explain what the ERROR and BIAS is (in figure 2/3), either in the figure description or in the text. In the text you write the biases in ppm/ppb and in the figure in %. Line 369 also a repetition of 361. In figure 2/3, it would be easier to understand if you write in the title what is the difference between the three figures (the boundary conditions). The chapter could be improved to make the statistics clearer and easier to understand, both in the text and in the figures.

We thank the referee for this helpful suggestion. In the revised manuscript, we have improved the clarity of the statistical analysis in Section 3.1. Specifically:
- We now report the bias and error both in percentage and in concentration units (ppm/ppb) in the text and in Figures 2 and 3.
- We clarified the definitions of bias and error in the figure captions.
- We removed the repeated lines (previously L361 and L369) to avoid redundancy.
- The panels of Figures 2 and 3 have been updated to explicitly indicate the type of boundary condition used for each panel, making the comparison clearer.

These changes aim to improve readability and ensure that the statistics and their context are easy to interpret.

4. Throughout the manuscript it is written as "CO2 and methane" rather than CO2 and CH4 or carbondioxide and methane. If there is no specific reason for this, I recommend to change all appearances to either one of the two mentioned options.

We thank the reviewer for this suggestion. However, the choice to write "CO2 and methane" is intentional: CO2 is often spoken as "C–O–two," whereas methane is usually referred to by name rather than as "CH4." This convention has been used consistently in our previous publications; therefore, we have retained it in this manuscript.

5. Line 273: transport errors of 3ppm and 15ppb. How did you select these values? What is the reason for choosing exactly these numbers?

These values are based on our expert judgement and were informed by sensitivity tests within the LETKF inversion framework, consistent with previous studies (e.g. Lunt et al. 2020, Scarpelli et al., 2024) using the in-situ data. Atmospheric transport model error plays a dominant role in R, as we discuss in the manuscript, and we find that significantly increasing this error beyond these values led to the observational signal being dominated by noise, reduces the effectiveness of the error reduction calculations.

Since our study focuses mainly on the GEMINI-UK sites measuring XCO2 and XCH4, we kept the 3 ppm and 15 ppb transport errors fixed for the DECC sites, while providing additional tests with varying XCO2 and XCH4 transport errors as described above.

6. Line274: scene dependent observational error: Can you please specify the numbers or give examples of these scene dependent standard deviations and explain the difference to the theoretical limit of the EM27/SUN or other studies? For example, Jones et al. 2021 (https://acp.copernicus.org/articles/21/13131/2021/) assumed the uncertainty to be 0.2 ppb based on the Allan analysis to the measured column difference when the instruments were co-located (Chen et al. 2016, https://acp.copernicus.org/articles/16/8479/2016/).

The observational error for each hourly value was calculated assuming 25 individual retrievals per hour. We used a single-retrieval uncertainty of 1.0 ppm for CO2 and 11 ppb for methane (Sha et al., 2020). This results in effective hourly uncertainties of 0.2 ppm and 2.2 ppb, respectively. These values are larger than the theoretical instrument precision limits reported in studies such as Jones et al. (2021) and Chen et al. (2016). But our assumed values are consistent with a side-by-side comparison of six EM27/SUNs during the commissioning phase of the Edinburgh urban experiment. We have clarified this point in the revised manuscript.

7. Line 415: whole section 3.4: It is not quite clear why you do this comparison and why it is relevant here? There are existing EM27/SUN vs. TCCON comparisons. What is the benefit of comparing them in your setup, in particular because you

mention that the comparison has its weaknesses of different spectral resolutions, as you mention in line 431. To my knowledge, it is possible to imitate EM27/SUN recordings with an HR125 by reducing the spectral resolution with the HR125. This was not considered here, why?

*The purpose of this comparison is to demonstrate the performance of the EM27/SUN instrument and enclosure set up that will be deployed at each of the GEMINI-UK locations, relative to that of the TCCON instrument at Harwell. The intention is to give the reader an indication that the EM27/SUN located at each GEMINI-UK site can provide data of a comparable quality to TCCON. This is also a first attempt to quantify the GEMINI-UK observation errors, using the TCCON observations as a baseline, which are needed when the GEMINI-UK column concentrations are included in inversion models used to calculate the UK's greenhouse gas emissions. An in-depth, systematic comparison between the EM27/SUN and TCCON greenhouse gas column concentrations, which will properly investigate the differences and errors introduced by different spectral resolutions, retrieval algorithms and other assumptions, are beyond the scope of this study and will be the subject of a dedicated paper. We have included a further paragraph in Section 3.4 to clarify the purpose of the comparison shown.*

8. Line 432: Does this mean, that you used different profiles for TCCON and EM27/SUN retrievals? If so, why?

   *The two retrieval schemes use the same a priori profiles, obtained from the GEOS-IT global climatological model. We will include a line in the manuscript to explicitly state this and provide a reference for the model used.*

9. Line 442: Can you please specify "higher and further"?

   *We have clarified what we mean by "higher and further education institutes" in the revised text:*

   *"The resulting network is hosted by further (post-secondary, non-degree) and higher (university degree) education institutions, underlining our commitment to delivering open data for all to use."*

10. Figure 6: It is not quite clear what the units of error reduction are. Please specify. The "chessboard" pattern further implies a high spatial resolution. This would further imply that single grid cells are affected by error reduction through additional observations differently in small spatial scales. However, it appears to be noise. The amount of additional EM27/SUN observations does most likely not influence single grid cells given the spatial resolution you show here. Why are you sure that this is not noise but a true error reduction? It would be helpful to see the simulated EM27/SUN observations in a plot to make further assumptions.

The "chessboard" pattern in Fig. 6 is visible due to the difference in resolution between the ensemble perturbations used to define the prior uncertainty and the resolution of the model grid. Specifically, the prior flux uncertainties were generated at 0.5°×0.625° resolution, while the model simulations and the error reduction calculations were performed at 0.25°×0.3125° resolution. This mismatch leads to structured artefacts in the a posteriori error reduction fields, particularly when computing relative reductions.

To address the "chessboard" issue, we have redone the calculations by re-aggregating the prior perturbed fluxes to the same resolution (0.5°×0.625°) as the a priori uncertainties. This led to changes in the values shown in Figures 6-9 but did not alter the spatial structure or affect the main conclusions of our initial results. For consistency, the error reduction maps are now plotted in % and the site locations are added to the maps, illustrating that the signal is dominated at the site locations. The updated figures are shown below:

[Figure]

Figure 6. *A posteriori* error reduction of a priori net CO2 (a) and (b) and methane (c) and (d) flux uncertainties from GEMINI-UK data for the first 15-day assimilation window of January (a) and (c) and July (b) and (d) 2019.

[Figure]

Figure 7. Same as Figure 6 but relative to the information already provided by the DECC tall tower data and Mace Head.

[Figure]

Figure 8. Same as Figure 7 but with the DECC tall tower data also including sites at Jodrell Bank and Invergowrie that will become operational in 2025.

11. Figure 9: The colorbar is capped for a lot of data points. It is difficult to estimate the relevance of certain areas/pixels given that the surrounding has the same value. Further there are pixels capping the colorbar as maximum positive flux in winter but also cap as maximum negative flux in summer (e.g. around Dublin). If there is no other physically relevant explanation, this might point towards an overfitting of these regions. Can you exclude overfitting for your setup and why?

Fig. 9 shows the prior fluxes only, not the posterior or the error reduction directly. We refer to this figure to illustrate that the regions where the error reductions occur (shown in Figs 6-8) spatially correspond to areas with substantial prior fluxes shown in Fig. 9. The negative fluxes are present for CO2 in July (e.g. over Dublin) when uptake typically dominates the flux while in winter it appears to be positive due to higher anthropogenic activities and less active vegetation.

Thus, this pattern is not related to any overfitting. We have clarified this in the text where Figure 9 is referenced.

12. A more careful color classification would increase the information content. It is not possible to distinguish flux values within certain areas (e.g. almost the whole of Ireland for CH4 in summer).

Agreed. We have added a more detailed colour classification in Fig. 9 and changed the colour scheme for methane. The figure has also been re-plotted to account for the changes in spatial resolution mentioned above and the updated version has the observational site locations used in the study:

[Figure]

Figure 9. Net fluxes of CO2 (a, b) and methane (c, d) for the first 15-day assimilation window of January (a, c) and July (b, d) 2019. The methane is plotted as CO2-equivalent. The green dots represent the GEMINI-UK site locations, the blue triangles – the DECC site locations (operational) and the black triangles – Jodrell Bank and Invergowrie that will become operational in 2025.

13. Similar to figure 6 it should be made clear, to what degree the visualisation might be noise or related to noise and what content of the figure is most likely based on true signal. Similar to figure 6, the pixels seem to mimic a much higher spatial resolution than the setup is being capable of resolving. A better visualisation of

single regions including some visual interpretation of the noise or the errors would give a better understanding of the scenes.

We kindly refer you to our response to comment 10, where we address the interpretation of Fig. 6.

14. Line 172-174: The data from the ICOS network already exists and is public. Furthermore, what would you use this data for

Yes, the reviewer is correct that the ICOS network already exists and is public. The ICOS site are shown for illustrative purposes, and the data will be used in the inversion calculations using the real GEMINI-UK observations.  This data will also be useful as boundary conditions (BC) for the UK. We have clarified this point in the revised manuscript where Figure 1 is mentioned for the first time.

15. Line 182-185: Why are the excluded sectors are not relevant/not included.

Thank you for the comment. We confirm that no sectors were excluded from the anthropogenic emissions in our simulations. We use all 12 sectors available in the TNO inventory for both CO2 and methane. In our setup, some sectors were grouped under broader labels (e.g., "other combustion" and "non-combustion") for CO2, following the configuration from a previous study done by Scarpelli et al., 2024. For methane, we include the same 12 sectors without any grouping. These differences are purely technical and do not affect our results, as the study focuses on total emissions and fluxes rather than sectoral attribution. We have revised the manuscript to clarify this point:

*"Our $CO_2$ and methane model simulations include anthropogenic emissions from all 12 sectors provided in the TNO inventory (Kuenen et al., 2022). For $CO_2$, these are grouped into nine categories, following Scarpelli et al. (2024): public power, industry, road and off-road transport, shipping, aviation, fugitive emissions, 'other' combustion, and 'non-combustion'. For methane, we include the same 12 sectors without grouping, and thus sources such as waste, solvents, livestock, and other agricultural sources are used explicitly. However, only the total emissions are estimated in our error reduction calculations; sector-specific fluxes are not resolved. "*

16. Line 188-189: The oceans, especially coastal areas are also a large methane source. Since at least half of the modelling domain is ocean including a lot of coasts, this should be considered. https://www.geomar.de/en/news/article/how-muchhttps://www.geomar.de/en/news/article/how-much-methane-is-released-by-the-oceanmethane-is-released-by-the-ocean

Thank you for pointing this out and for sharing the reference to the gridded global dataset of oceanic methane emissions. We acknowledge that ocean and coastal

waters can be a source of atmospheric methane and that such datasets are becoming increasingly available (e.g., Weber et al., 2019). However, we did not include ocean methane fluxes in our simulations because the methane ocean source is ignored in the GEOS-Chem by default and contribution of ocean methane flux is less significant in comparison with the rest emission sources we are focused on, especially for the European region and UK selected in our study. We have clarified this point in our study:

*"Oceanic and coastal methane fluxes are not included in our simulations. While such fluxes exist (e.g., Weber et al., 2019), their contribution to atmospheric methane over the European and UK domain is relatively small compared with the terrestrial and anthropogenic sources considered here. Future studies could include oceanic emissions as high-resolution datasets become more widely available."*

17. Line 344: Optimal regarding that the network-sensitivity should be the same everywhere? How do you know whether it is optimal - did you brute-force it or use some optimization algorithm?

We had 45 candidate locations to host a total of 7 (of 10; other three sites have been fixed) EM27/Sun instruments. The final selection of the 7 sites was determined by evaluating overall spatial coverage by the network. This was done by comparing the contribution of each site (i.e., 'with' versus 'without') to the overall network sensitivity. The network sensitivity is calculated by using the GEOS-Chem model to track the emission perturbations from each of 60 1°×1° grid boxes over Ireland and the UK. For each of these grid boxes, the network sensitivity to UK $CO_2$ (or $CH_4$) emissions is calculated by:

$$\delta CO_2 = [\sum_{i=1}^{N} \Delta XCO2(i)]^{1/2}, \quad (1)$$

where $N$ is the number of the instruments, and $\Delta XCO2(i)$ is the mean change of dry-air $CO_2$ column (XCO2) observed by the $i$th EM27 instrument when a fixed amount of CO2 gas is released in 7 days from this grid box. As one example, below plot shows increasing network sensitivity, when a EM27 instrument is added to Birmingham to construct a possible network of 6 sites:

[Figure]

Figure 1: Change of network sensitivity when one EM27 instrument in Brimingham is added to a (random) network of 5 sites (i.e., Belfast, Glasgow, Newcastle, Leeds and Cardiff).

18. Line 396: Have these apriori estimates been introduced in the paper? The inventories for the GEOS-CHEM simulation have been introduced, but if they are the synthetic "true" emissions, what is the prior?

The original inventories used in the GEOS-Chem simulation represent the synthetic "true" emissions. The prior emissions used in the inversion are generated by perturbing this true inventory to represent the initial uncertainty. Thus, the perturbed inventory serves as the prior estimate, while the original inventory acts as the synthetic truth for evaluation purposes. To clarify this point we have added this text to the manuscript:

*"The original inventories used in the GEOS-Chem simulation represent the synthetic "true" emissions. The prior emissions used in the inversion are generated by perturbing this true inventory to represent the initial uncertainty. Thus, the perturbed inventory serves as the prior estimate, while the original inventory acts as the synthetic truth for evaluation purposes."*

19. Line 407-408: It would be good to have this additional error reduction as an average over the UK land area not just their maximum values.

We have calculated the minimum, mean, and maximum error reductions over land area (UK + Ireland) for all site configurations, and these are summarized in Table 2:

Table 2. Error reduction (%) for CO2 and CH4 fluxes over the UK + Ireland for different network configurations. Results are shown for GEMINI-UK alone and GEMINI-UK combined with 5 or 7 DECC sites. Shown statistics are the minimum, mean, and maximum error reduction for January and July. All results assume a transport error of 0.5 ppm for XCO2 and 5.0 ppb for XCH4. Instrumental uncertainties were assumed to be 1 ppm for XCO2 and 11 ppb for XCH4 and were scaled by the number of observations per site per hour (N=25). The corresponding figure panels are indicated in the last column.

| Region | Spec | Network | Number of sites | Month | Error Reduction Min, % | Error Reduction Mean, % | Error Reduction, Max, % | Figure |
|---|---|---|---|---|---|---|---|---|
| UK + Ireland | CO2 | GEMINI-UK | 11 | January | 0.7 | 9.4 | 50.9 | 6a |
| UK + Ireland | CO2 | GEMINI-UK | 11 | July | 8.4 | 28.0 | 58.9 | 6b |
| UK + Ireland | CH4 | GEMINI-UK | 11 | January | 1.2 | 15.8 | 54.9 | 6c |
| UK + Ireland | CH4 | GEMINI-UK | 11 | July | 10.0 | 39.0 | 74.7 | 6d |
| UK + Ireland | CO2 | GEMINI-UK & DECC | 16 | January | 0.2 | 1.0 | 7.9 | 7a |
| UK + Ireland | CO2 | GEMINI-UK & DECC | 16 | July | 0.7 | 7.2 | 24.3 | 7b |
| UK + Ireland | CH4 | GEMINI-UK & DECC | 16 | January | 0.2 | 1.1 | 10.0 | 7c |
| UK + Ireland | CH4 | GEMINI-UK & DECC | 16 | July | 1.4 | 6.0 | 28.7 | 7d |
| UK + Ireland | CO2 | GEMINI-UK & DECC | 18 | January | 0.1 | 0.8 | 7.6 | 8a |
| UK + Ireland | CO2 | GEMINI-UK & DECC | 18 | July | 0.5 | 4.8 | 17.7 | 8b |
| UK + Ireland | CH4 | GEMINI-UK & DECC | 18 | January | 0.1 | 0.8 | 7.2 | 8c |
| UK + Ireland | CH4 | GEMINI-UK & DECC | 18 | July | 0.5 | 3.7 | 19.2 | 8d |

20. Line 413-414: How does this statement connect to the figure. It just shows one posterior, not the improvement of the posterior.

We can confirm that Fig. 9 shows the prior fluxes only, not the posterior or the error reduction directly. We refer to this figure to illustrate that the regions where the error reductions occur (shown in Figs 6-8) spatially correspond to areas with substantial prior fluxes shown in Fig. 9. To better clarify this, we improved the text in Lines 413-414:

*"The maximum error reductions were 7.6% for CO2 and 7.2% for methane in January, and 17.7% for CO2 and 19.2% for methane in July. Overall, the error reductions (shown in Fig. 6-8) correspond spatially to regions with substantial prior fluxes, as illustrated in Fig. 9. This demonstrates that the GEMINI-UK data improve knowledge particularly over areas with high net CO2 and methane fluxes."*

21. Line 441-442: The optimality criterion for the sensor placement you mentioned was the sensitivity coverage. Is that equivalent?

We used the sensitivity coverage for the site selection and then checked the random error reduction by applying perturbations and LETKF. Is it what the referee wants to clarify here?

22. Line 444-445: These are the values if there were not other measurements. You should use the values from figure 8. Please also adjust the numbers in the abstract.

We can confirm that the lower values in the reported ranges correspond to the error reductions calculated in Figures 7-8, which include other measurements (e.g., the DECC tall towers), while the higher values are from Figure 6, which reflects the impact of the GEMINI-UK sites alone. Therefore, the ranges are intended to represent both scenarios: (1) GEMINI-UK operating as a standalone network, and (2) GEMINI-UK complementing the existing measurement infrastructure. This is an informative approach to present the full extent of the network's potential impact. We have added clarification to the text where the error reduction ranges are given including the abstract:

*"Based on our results, we expect that GEMINI-UK will deliver significant error reductions in CO2 flux estimates, up to 51% in January and up to 59% in July (GEMINI-UK only), and up to 8% in January and up to 24% in July when combined with DECC sites."*

*"Despite the network being optimally designed to enhance our understanding of UK CH$_4$ fluxes, we expect, based on our calculations, that GEMINI-UK will also substantially reduce uncertainties of methane emissions, up to 55% in January and up to 75% in July (GEMINI-UK only), and up to 10% in January and up to 29% in July when combined with DECC sites."*

In addition, we also highlight the differences in error reductions arising from the transport error assumptions, using the values provided in Tables A1-A2 in the updated version of the manuscript.

23. Figure 2: The definition of R, bias and error should be mentioned in the text. Are they calculated only using the daytime hours?

We confirm that the definitions of R (Pearson correlation coefficient), bias, and error have now been included in the text in Section 3.1. These metrics were calculated over the full time series, not limited to daytime hours.

24. Figure 2: It would be good to see these fits at the different sites to get a better idea of where each boundary conditions performs better.

   We have added the figures of diurnal cycle evaluation (mod vs observed) for the individual sites for both CO2 and methane. These figures will be placed in the Appendix to avoid overloading the main text.

**Minor comments:**

1. 144: instrumentsorldwide

   Corrected.

2. 190: described **by** Deng

   Corrected.

3. 195: from

   Corrected.

4. 197: 2° x 2.5° make the space consistent (e.g. 201)

   Corrected.

5. 201:

   Corrected.

6. 213: methane**.** These

   Dot added.

7. 279:

   Corrected.

8. 286: July instead of June

   Corrected.

9. Line 81: "EM27/SUN"

Corrected.

10. Line 87: You should cite TCCON here since it is the first mentioning of it

Reference added there.

11. Line 91: "in situ" or "in-situ"? The text uses both variants

Which one is the most common to use?

12. Line 111: You could also cite https://doi.org/10.5194/amt-9-2303-2016

The reference added.

13. Line 118: You could consider to also cite the PROFFAST Software release itself. This is how the PROFFASTpylot cited it: https://gitlab.eudat.eu/coccon-kit/proffastpylot/https://gitlab.eudat.eu/coccon-kit/proffastpylot/-/blob/joss/paper.bib?ref_type=heads/blob/joss/paper.bib?ref_type=heads#L234-L246. Maybe they have a more updated way of citing it by now.

We have now added a citation to the PROFFAST software release itself in addition to the references already included.

14. Line 122: https://hitran.org/citepolicy/

The webpage link added.

15. Line 144: "instruments worldwide"

Space added.

16. Line 168: The model has not been introduced at this point

We have added clarification to Line 168:
*"For our theoretical calculations, we use the mean values of the five lowest model levels (from the GEOS-Chem 3-D Model of Atmospheric Chemistry and Transport, described in the next section), as the station heights (ranging from 56 m to 380 m above sea level) and the top inlet heights (ranging from 45 m to 248 m above ground level) fall within the altitude range of these model levels."*

17. Line 177-178: What is the nesting? Is there an inner domain? Mention Figure 1b here again.

We have revised Lines 177–178 to clarify the setup and added a reference to Figure 1b:

*"We run the GEOS-Chem model at a horizontal resolution of 0.25×0.3125° for a regional European domain (−15 to 35°E and 34 to 66°N), as shown in Figure 1b, with 47 vertical levels ranging from the surface to 0.01 hPa, described by a hybrid-sigma coordinate system. This nested simulation is driven by boundary conditions from three independent sources, described below."*

18. Line 182: Include the "CAMS-REG-v4" name in the text.

    Actually, we use TNO which became CAMS-REG-v4, but not the original CAMS_REG-v4.

19. Line 189: Lateral exchange from where to where?

    We have clarified in the manuscript that the lateral exchange represents carbon moved from production to consumption sites, primarily due to farming (e.g., crops), and updated Line 189 accordingly.

20. Line 197-198: This should be mentioned before, when presenting the "nested model".

    We respectfully disagree with this comment. Lines 197–198 simply provide a brief description of how the boundary conditions were obtained. The setup and purpose of the nested model are introduced earlier in the manuscript, and we believe that discussing the boundary conditions in detail at that earlier point would interrupt the logical flow of the model description. Therefore, we feel it is appropriate to present this information where it currently appears.

21. Line 213: Dot is missing.

    Corrected.

22. Line 326: How long is the embargo?

    The embargo period has not been finalised at this time, but it is likely to be several weeks (except in special circumstances, e.g. cal/val of MicroCarb) to ensure the public data have passed QA/QC procedures. The data will also be subject to re-assessment on an annual basis.

23. Line 462: New technologies like?

    We are currently horizon scanning to determine suitable new technologies.

24. Figure 6: Where does this checkered pattern come from? It would be helpful to have the Gemini-UK locations plotted here as well.

Please see our response to comment 10 in the major comments section, where we address the interpretation of Fig. 6. The GEMINI-UK locations are plotted in the updated version of the figure/paper.

25. Abstract line 10: these ... network. Shouldn't it be this network?

Corrected to 'this network'

26. Line 46: you could consider to cite also Bayesian inversion studies that incorporate EM27/SUN measurements, e.g.:
https://acp.copernicus.org/articles/21/13131/2021/
https://acp.copernicus.org/articles/23/6897/2023/

References added.

27. Line 388: what is considered an observation here? 15min averages or 1min averages? What is your single measurement sample frequency of the EM27/SUN?

In Line 388, the reported numbers refer to hourly averaged observations. In our error reduction calculations, we assume that each hourly average is based on 25 individual EM27/SUN measurements.

28. Line 421: You mention TCCON data is shown but it is only 3 rows later that you explain where it is shown. This is confusing.

To avoid confusion, we have removed the word "shown" from the sentence in Line 421.

29. Line 428: two times "within"

Agreed and corrected.

30. Appendix A2: For readers to get more context it could be helpful to briefly mention existing enclosure designs, e.g.
https://amt.copernicus.org/articles/11/2173/2018/

We have added the following text starting end of line 720:

*...associated changes in weather conditions. Other EM27/SUN enclosures have been developed by academic institutions including TUM (Heinle and Chen, 2018, https://doi.org/10.5194/amt-11-2173-2018), Heidelberg University (Knapp et al., 2021, https://doi.org/10.5194/essd-13-199-2021), University of Wollongong (Velazco et al., 2019, https://doi.org/10.5194/essd-11-935-2019), and LSCE (Lopez et al., 2025, https://doi.org/10.5194/egusphere-egu25-18600). Aside from the enclosure developed by Knapp et al., which is designed for ship deployment and includes additional fast slew mirrors and full waterproofing, these other enclosures are not fully weatherproof and expose the instrument to the local environment while observations are being taken. As of this year the LSCE enclosure design is being manufactured and distributed by eloneo (ELONEO - Instrumentation environnementale performante), however the other designs are not commercially available.*

**Anonymous Referee #2**

The GEMINI-UK network will measure column average dry air mole fractions of $CO_2$ and $CH_4$ with EM27/SUN spectrometers at ten sites across the UK, with the goal to reduce the uncertainty in regional net fluxes estimates. In this study, Kurganskiy et al. present its network design, co-located measurements of one of its instruments with TCCON and the networks theoretical benefits for reducing net flux uncertainties of $CO_2$ and $CH_4$. The authors determine the networks potential benefit by quantifying the uncertainty reduction in inversions of simulated measurements.

The network constitutes a substantial contribution to the overall atmospheric GHG observing system in the UK. Presenting the network design and instrument performance is a valuable step for later usage of the network data. Estimating the networks benefit does not only provide insights of the specific network but is also informative for initiatives in other regions. The methods used here are an appropriate tool to investigate this question and the criteria applied to simulate the data yield are realistically chosen. However, the results depend strongly on the uncertainty assumptions, which are only little discussed. I recommend publication of the article after a major revision, addressing the following comments.

**Major Comments**

The theoretical network performance is evaluated in a closed-loop experiment based on measurements which are simulated with the forward model used for the inversion. While this works well to evaluate the spatial structure of the uncertainty reduction, the magnitude of the uncertainty reduction depends mainly on the values chosen for a priori uncertainty and measurement/transport error. In the manuscript, both are not sufficiently justified and discussed given their importance. The observation error is not specified, only described as "scene-dependent". The sensitivity of the results to the assumptions should be presented.

We agree with this reviewer comment on the importance of prior and measurement/transport errors having the impact on the results. This point was also raised by Reviewer 1.

We clarify the choice of the specified uncertainties by citing the previous work based on our expert judgement. The settings we use correspond to the ones used in real data inversions (e.g. Feng, Palmer, Scarpelli et al., 2024) for GHG flux estimates using the GEOS-Chem atmospheric chemistry transport model and in-situ observations including the DECC network.

For our baseline OSSE calculations, we used a transport error of 0.5 ppm for XCO2, corresponding to the middle of the range quantified by McNorton et al. (2020, GMD), and 5 ppb for XCH4, consistent with the bias estimates reported by Stanevich et al. (2021, ACP). To assess the robustness of our results, we performed additional sensitivity tests by setting up the atmospheric transport uncertainty a factor of 2 lower and higher than the baseline values (0.25 and 1.0 ppm for XCO2; 2.5 and 10 ppb for XCH4), shown in Figures A3-A8. These tests confirm that while the absolute error reduction values vary with the assumed transport errors, the spatial patterns and qualitative conclusions remain robust.

[Figure]

Figure A3. Same as Figure 6 but using a factor of 2 lower uncertainties for XCO2 and XCH4 (0.25 ppm for XCO2 and 2.5 ppb for XCH4).

[Figure]

Figure A4. Same as Figure 7 but using a factor of 2 lower uncertainties for XCO2 and XCH4 (0.25 ppm for XCO2 and 2.5 ppb for XCH4).

[Figure]

Figure A5. Same as Figure 8 but using a factor of 2 lower uncertainties for XCO2 and XCH4 (0.25 ppm for XCO2 and 2.5 ppb for XCH4).

[Figure]

Figure A6. Same as Figure 6, but with XCO2 and XCH4 uncertainties increased by a factor of two (1.0 ppm for XCO2 and 10.0 ppb for XCH4).

[Figure]

Figure A7. Same as Figure 7, but with XCO2 and XCH4 uncertainties increased by a factor of two (1.0 ppm for XCO2 and 10.0 ppb for XCH4).

[Figure]

Figure A8. Same as Figure 8, but with XCO2 and XCH4 uncertainties increased by a factor of two (1.0 ppm for XCO2 and 10.0 ppb for XCH4).

Throughout the analysis, the theoretical network performance is evaluated based on the relative change of the uncertainty. However, the ultimate goal is to improve the knowledge on the net fluxes. For this, evaluating the total change in uncertainty is much more informative. Especially for evaluating the results shown in Fig. 7 and 8, which show significant spatial differences. This will also make the argument in L414 clearer. An estimate on the uncertainty reduction of the national estimate could be instructive.

Following this suggestion, as well as the comments from Reviewer 1, we have provided statistical metrics for the land-only error reduction (UK + Ireland). The table for the baseline results (see our response to Reviewer 1) is included (Table 2), and we have added two additional tables in the appendix (Tables A1 and A2) showing the same metrics for low and high transport errors for XCH4 and XCO2 (corresponding to half and twice the baseline uncertainties, respectively):

Table A1. Error reduction (%) for CO2 and CH4 fluxes over the UK + Ireland for different network configurations. Results are shown for GEMINI-UK alone and GEMINI-UK combined with 5 or 7 DECC sites. Shown statistics are the minimum, mean, and maximum error reduction for January and July. All results assume a transport error of 0.25 ppm for XCO2 and 2.5 ppb for XCH4. Instrumental uncertainties were assumed to be 1 ppm for XCO2 and 11 ppb for XCH4 and were scaled by the number of observations per site per hour (N=25). The corresponding figure panels are indicated in the last column.

| Region | Spec | Network | Number of sites | Month | Error Reduction Min, % | Error Reduction Mean, % | Error reduction, Max, % | Figure |
|---|---|---|---|---|---|---|---|---|
| UK + Ireland | CO2 | GEMINI-UK | 11 | January | 1.1 | 11.3 | 57.8 | A3a |
| UK + Ireland | CO2 | GEMINI-UK | 11 | July | 10.7 | 32.6 | 65.6 | A3b |
| UK + Ireland | CH4 | GEMINI-UK | 11 | January | 1.7 | 18.2 | 60.3 | A3c |
| UK + Ireland | CH4 | GEMINI-UK | 11 | July | 12.3 | 42.7 | 78.4 | A3d |
| UK + Ireland | CO2 | GEMINI-UK & DECC | 16 | January | 0.3 | 1.5 | 11.4 | A4a |
| UK + Ireland | CO2 | GEMINI-UK & DECC | 16 | July | 0.9 | 9.7 | 29.9 | A4b |
| UK + Ireland | CH4 | GEMINI-UK & DECC | 16 | January | 0.3 | 1.6 | 13.9 | A4c |
| UK + Ireland | CH4 | GEMINI-UK & DECC | 16 | July | 2.0 | 7.7 | 32.1 | A4d |
| UK + Ireland | CO2 | GEMINI-UK & DECC | 18 | January | 0.1 | 1.2 | 10.9 | A5a |
| UK + Ireland | CO2 | GEMINI-UK & DECC | 18 | July | 0.7 | 6.7 | 22.7 | A5b |
| UK + Ireland | CH4 | GEMINI-UK & DECC | 18 | January | 0.2 | 1.1 | 10.0 | A5c |
| UK + Ireland | CH4 | GEMINI-UK & DECC | 18 | July | 0.8 | 5.0 | 22.4 | A5d |

Table A2. Error reduction (%) for CO2 and CH4 fluxes over the UK + Ireland for different network configurations. Results are shown for GEMINI-UK alone and GEMINI-UK combined with 5 or 7 DECC sites. Shown statistics are the minimum, mean, and maximum error reduction for January and July. All results assume a transport error of 1.0 ppm for XCO2 and 10.0 ppb for XCH4. Instrumental uncertainties were assumed to be 1 ppm for XCO2 and 11 ppb for XCH4 and were scaled by the number of observations per site per hour (N=25). The corresponding figure panels are indicated in the last column.

| Region | Spec | Network | Number of sites | Month | Error reduction Min, % | Error Reduction Mean, % | Error Reduction, Max, % | Figure |
|---|---|---|---|---|---|---|---|---|
| UK + Ireland | CO2 | GEMINI-UK | 11 | January | 0.4 | 7.3 | 41.5 | A6a |
| UK + Ireland | CO2 | GEMINI-UK | 11 | July | 6.1 | 22.4 | 49.7 | A6b |
| UK + Ireland | CH4 | GEMINI-UK | 11 | January | 0.8 | 12.9 | 47.1 | A6c |
| UK + Ireland | CH4 | GEMINI-UK | 11 | July | 7.5 | 34.3 | 69.1 | A6d |
| UK + Ireland | CO2 | GEMINI-UK & DECC | 16 | January | 0.1 | 0.6 | 4.7 | A7a |
| UK + Ireland | CO2 | GEMINI-UK & DECC | 16 | July | 0.5 | 4.7 | 17.4 | A7b |
| UK + Ireland | CH4 | GEMINI-UK & DECC | 16 | January | 0.1 | 0.6 | 6.1 | A7c |
| UK + Ireland | CH4 | GEMINI-UK & DECC | 16 | July | 0.9 | 4.2 | 24.0 | A7d |
| UK + Ireland | CO2 | GEMINI-UK & DECC | 18 | January | 0.1 | 0.5 | 4.5 | A8a |
| UK + Ireland | CO2 | GEMINI-UK & DECC | 18 | July | 0.3 | 3.0 | 12.0 | A8b |
| UK + Ireland | CH4 | GEMINI-UK & DECC | 18 | January | 0.1 | 0.4 | 4.4 | A8c |
| UK + Ireland | CH4 | GEMINI-UK & DECC | 18 | July | 0.3 | 2.5 | 15.0 | A8d |

No other sources of error to the flux estimates are discussed. This underestimates the benefit of the total column measurements. For example, a misrepresentation of the boundary layer would induce errors in the flux estimates to which the total column measurements would be much less sensitive compared to in situ measurements. This and other types of systematic errors can not be represented in this closed-loop experiment. However, they should still be discussed.

This issue has been raised by Referee 1 and thus we duplicate our reply here as well: In our OSSE framework, we assess the theoretical performance of the GEMINI-UK system via the relative error reduction between the prior and posterior error covariance matrices. This allows us to evaluate the potential constraint provided by the observing system. However, other important sources of uncertainty are not fully captured, including systematic model errors. These limitations are inherent in the OSSE design, where the same model is used to generate the synthetic truth and perform the error reduction calculations by perturbing the prior fluxes and applying LETKF.

The result of section 3.1 is that "CAMS in situ" provides the best fit to DECC measurements. On what quantitative result is this conclusion based? The analysis before reads inconclusive.

We agree that "CAMS in-situ" provides the best fit to DECC measurements might read inconclusive. CAMS in-situ BCs provides ones of the lowest biases at the DECC sites and therefore that dataset was chosen for the error reduction calculations in this study.

To better clarify this, we have implemented the following changes in section 3.1 (last sentence):

*"Overall, we find that the CAMS in situ lateral boundary conditions provide one of the lowest biases at the DECC sites, particularly in comparison to CAMS-EGG4, and therefore we use them for the subsequent error reduction calculations."*

The bias and standard deviation of the $XCO_2$ and $XCH_4$ measurements by the GEMINI-UK EM27/SUN with respect to TCCON are in line with the findings of Frey et al. 2019. However, the standard deviation is larger than what is expected for the random error of EM27/SUN and TCCON measurements (Frey et al. 2019, Wunch et al. 2011). It should be made clear, that the standard deviation presented here is not caused by random but by systematic errors. Additionally, it would be helpful to set bias and scatter in relation to the expected $XCO_2$ and $XCH_4$ variability and the expected gradients though out the network. The results should be used to inform the measurement error assumed in the network performance analysis.

We will clarify here what the biases and standard deviations presented represent with respect to how the EM27/SUN and enclosure system performs with respect to the performance demonstrated by the Harwell TCCON station. In addition, further investigation of this preliminary data revealed issues with the EM27/SUN solar tracker, which have since been resolved. We will replace the figures and data shown with a more recent side-by-side comparison using observations made after the problem was fixed, which will be more representative of the anticipated performance of each EM27/SUN station in the GEMINI-UK network.

No results are shown for the network design process. Especially the results of the sensitivity analysis for the final selections would be interesting for the reader. It would be help to visualize the influence region for individual sites and the overall network sensitivity.

We thank the reviewer for this comment. The network design was a laborious process, described above in terms of the approach we took in response to an earlier comment. Visualising this approach in a way that would be useful to a reader would involve several multi-panel plots due to the brute-force perturbation process we adopted, the different combinations of sites considered during the design stage, and the varying influence of emitting regions. With respect, considering the need for brevity of the material being reported and the limited worth of these plots given we have already installed the sites, we have chosen not to include these plots in the revised manuscript.

In response to comment 17 raised by Reviewer 1, we show a subset of the results that ultimately led us to choose the University of Birmingham as a host site. The sensitivity of the network to regional emissions, when Birmingham is included, is shown, which we used to identify a range of higher education institutions (HEIs) to approach. Ultimately, we chose the University of Birmingham because we have colleagues there who understand the commitment needed to establish and maintain a measurement network, understood the needs to verify the UK progress with net zero, and because their estates team secured a suitable location for the EM27/SUN.

In some instances, we couldn't identify suitable HEIs and instead approached a suitable further education institute, e.g. Elizabeth College in Guernsey.

**Minor comments**

For the forward operator H, the modeled columns are convolved with the averaging kernels. For this, two averaging kernels are used (one for January, one for July). This implies that the SZA dependence of the averaging kernel is neglected, even though L269 names "scene-specific averaging kernels". Generally, the SZA dependence of the EM27/SUN averaging kernels for $CO_2$ and $CH_4$ are not negligible (Hedelius et al. 2016). If it is neglected here, what is the justification for this?

We agree that the EM27/SUN averaging kernels depend on scene parameters, in particular the solar zenith angle (SZA). In this study, we used two averaging kernels - one for winter (10 December 2021) and one for summer (16 June 2021) which were derived using real scene-specific retrievals, including the actual SZA on those days. While we did not explicitly vary the averaging kernel with each individual observation, the applied kernels do incorporate the SZA conditions on their respective dates. Therefore, the SZA dependence was not entirely neglected, but rather approximated using representative days from each season. We have revised the manuscript to clarify this point in L269-271:

*"To translate the resulting vertical profiles of $CO_2$ and methane to XCO2 and XCH4, respectively, we use scene-specific averaging kernels derived from the EM27/SUN instrument at UCL, London. For simplicity, we apply one kernel for January (based on retrievals from 10 December 2021) and one for July (16 June 2021). These kernels reflect the actual scene parameters, including solar zenith angle, on those days, but are representative of the values in those contrasting months."*

The weatherproof enclosure is mentioned in the manuscript and described in Appendix A. It is mentioned that the optical dome has no adverse effect on the tracking performance. Were tests performed (for example through simultaneous measurements with a standard EM27/SUN) that demonstrate the absence of adverse spectral effects, i.e. on the retrieved columns? This would be especially interesting, since integration into COCCON is planned.

Laboratory tests were performed on samples of the glass used in the optical dome to check the transmittance of the material at the appropriate wavelengths. These showed that although the transmittance was reduced by around 10%, the reduction had low wavelength dependence. Side-by-side measurements of the atmosphere performed with and without the optical dome showed that any effects on the XCO2 and XCH4 retrievals were within the measurement uncertainty, although there is a small measurable impact on XCO. These findings will be the subject of a future publication, describing the use of these enclosures in an urban EM27/SUN network around Edinburgh (Morrison et al., 2025).

It should be addressed that transport/representativeness error is typically different (lower) for total column measurements than for in situ measurements, since they are less sensitive to very local effects.

Thanks for this helpful suggestion! To address this point, we have added the following clarification at Line 282:

*"We use a transport error of 3 ppm for CO2 and 15 ppb for CH4 at the in situ DECC sites. For the total column observations from the GEMINI-UK network, we adopt smaller transport errors (0.5 ppm for CO2 (McNorton et al. (2020, GMD), and 5 ppb for XCH4, consistent with the bias estimates reported by Stanevich et al. (2021, ACP), reflecting their reduced sensitivity to local variability near the surface. In addition, we assess the impact of uncertainty assumptions by repeating the analysis with twice lower and twice higher errors for XCO2 and XCH4. "*

In section 2.2, the term "theoretical calculations" is unclear. For which part of the study is real or simulated in situ data used?

In this study, the term *"theoretical calculations"* refers to our use of **simulated** GEMINI-UK and DECC observations in the error reduction experiments. Real observational data are not used in the assimilation or error analysis presented here. Section 2.2 includes descriptions of existing in-situ networks to highlight their relevance for **future** real-data inversions. To clarify this, we have modified the text in Section 2.2 as follows:

*"In this study, we perform theoretical error reduction calculations using simulated GEMINI-UK and DECC observations. Real observational data are not assimilated in the results presented here but will be used in future inversion experiments.*
*For these theoretical calculations, we also consider continuous in-situ concentration measurements of CO2 and methane collected at a fixed elevation as part of the UK DECC network (Stanley et al., 2018), which currently includes five sites. These data have been used to produce data-driven UK estimates that supplement inventory estimates reported annually to the UNFCCC.*
*To simulate these observations, we use the mean values of the five lowest model levels, as the station heights (ranging from 56 m to 380 m above sea level) and the top inlet heights (ranging from 45 m to 248 m above ground level) fall within the altitude range of these levels. We also include the surface measurement site at Mace Head, western Ireland (a few metres above local terrain), and new tall towers at Jodrell Bank in northwest England and at Invergowrie in east Scotland.*
*For future inverse calculations using real data, we will consider only tall tower measurements collected at the highest inlet heights, typically 90–248 m above ground, during local hours of 10:00-17:00 to avoid the influence of the nocturnal boundary layer, when measurements may be skewed or localized due to thinner and more stratified conditions.*
*When we analyse real data in future work, we will also use CO2 and methane data collected across mainland Europe as part of the Integrated Carbon Observing System*

*(ICOS; Heiskanen et al., 2022), which comprises 170 sites across 16 European countries (Figure 1), to provide lateral boundary conditions for the UK."*

The term "baseline calculation/model" is unclear. What is this baseline contrasted by?

We have clarified in the manuscript (Line 196) that the "baseline calculations" refer to the prior model simulations obtained by spinning up the nested model using lateral boundary conditions from the equivalent global model. This baseline serves as the reference state before assimilation:

*"We spin up the nested model using lateral boundary conditions from the equivalent global version of the model run to form our baseline calculations, which represent the prior model state before applying perturbations to prior emissions and before data assimilation."*

L196 and L330 imply that the lateral boundary condition is given by the global GEOMS model. In L379 it is specified that "CAMS in situ" is used for subsequent calculation. Does this refer to future studies or to the subsequent sections of this study?

In our study, we use three different lateral BC datasets - GEOS-Chem global, CAMS in situ, and CAMS EGG4 - to test sensitivity in the forward model. For the network design and site selection calculations described in L330, we use the nested GEOS-Chem simulation driven by GEOS-Chem global BCs. In the perturbed ensemble runs used for error reduction calculations (L379), we chose the CAMS in situ product, which had one of the lowest mean biases at the DECC sites (Figure 4 in the original paper draft). The phrase "subsequent calculations" in L379 refers to the ensemble error reduction analysis presented in this paper, not to future studies. We have clarified this throughout the manuscript.

Please use clear names for the lateral boundary conditions. Does CAMS in situ refer to vCAMS-73? Additionally, it is not clear what vCAMS-73 refers to. The references (Chevallier, 2023; Segers 2023) do not contain a DOI or URL, and the Atmosphere Data Store does not list anything under that reference.
In L214 it is stated that the threshold for well-mixed boundary layer is "analyzing the observed time series". For what is the time series analyzed?

We thank the Referee for pointing this out. We confirm that CAMS in situ refers to the vCAMS-73 product. To improve clarity, we have updated the manuscript to use the full name of the product and provided the appropriate link to the documentation (https://atmosphere.copernicus.eu/sites/default/files/2020-06/CAMS73_2018SC2_%20D5.2.1-2020_202004_%20CO2%20inversion%20production%20chain_v1.pdf).
The references to Chevallier (2023) and Segers (2023) have been removed.

Regarding Line 214, we have clarified in the text that the observed time series at each site were analysed using a set of candidate thresholds for well-mixed boundary layer filtering. The evaluation involved examining how each threshold affected the number of

records retained, and the final threshold was selected based on expert judgement to best reflect local conditions and data characteristics:

*"We consider conditions to be well-mixed when the standard deviation of concentrations across the lowest five model vertical layers - approximately the lowest 600 metres - is less than 5 ppm for CO2 and less than 25 ppb for methane. These threshold values were chosen based on expert judgement, guided by an analysis of the observed time series and the number of data points retained under different candidate thresholds."*

In L229, why is the a priori termed "background"?

In the context of LETKF and broader data assimilation frameworks, the term *background* is commonly used to refer to the *a priori* estimate of the state vector, and the two terms are often used interchangeably in the literature.

In L290-L297, the notation is quite confusing. Are the cursive and bold "a" in the exponent the same? If so, what is the difference between (6) and (7). If not please change the notation. Additionally, what is N?

Thank you for pointing this out. You are correct that the notation was inconsistent and potentially confusing. In our revised version, we distinguish between the *local* posterior error covariance matrix used within the LETKF update step and the *ensemble-derived* posterior error covariance matrix used to quantify a posteriori uncertainty. To clarify this distinction, we now denote the local posterior error covariance matrix as $\widetilde{P}^a$ (Eq. 6), and the ensemble posterior error covariance matrix as $P^a$ (Eq. 7). We confirm that N refers to the ensemble size and is the same in both equations, and we have replaced n with N in Eq. 6 for consistency.

In L335, what does "this contribution" refer to?

To improve clarity, we have combined the sentence in Line 335 with the preceding one into a single paragraph clarifying that "this contribution" refers to the influence of $CO_2$ fluxes from each grid box on simulated $XCO_2$ values.

In L344, how is the optimal subset defined? Is this the subset with maximum S, or do you have other criteria (i.e. related to the spatial distribution)?

As we mentioned in our reply to Reviewer 1, we aim to provide a good spatial coverage over UK and Ireland, to provide reliable CO2/CH4 emission estimates for UK major administrative regions. There could be other possible criteria to decide which sites are the best, but they are usually computationally too expensive to explore those possible site combinations.

In Figure 2 and 3, the different panels are hard to distinguish on one glance. Please simplify the title and indicate the lateral boundary condition used in the respective panel or row.

Thank you for the suggestion. We have revised Figures 2 and 3 to make the boundary conditions more immediately clear. Specifically, we now indicate the lateral boundary condition used in each individual panel title. This allows readers to identify both the simulation month and the boundary condition at a glance. The figure captions have also been updated accordingly.

A checkerboard pattern is visible in Fig. 6. What causes this?

Thank you for pointing this out. The checkerboard pattern in Fig. 6 arises from the resolution mismatch between the ensemble perturbations used to define the prior uncertainty and the model grid used in the simulations. Specifically, the prior flux uncertainties were generated at 0.5° × 0.625° resolution, while the model simulations and the error reduction calculations were performed at 0.25° × 0.3125° resolution. This mismatch produces structured artefacts in the a posteriori error reduction fields, particularly when expressed as relative reductions. Following this comment, and a similar one from Reviewer 1, we have repeated the calculations using prior fluxes reaggregated to 0.5° × 0.625° resolution. This update leads to revised Figs. 6–9, but the main conclusions of the study remain unchanged (see our response to Reviewer 1, comment 10).

**Technical corrections**

- L81 EM27 Sun

  Replaced with EM27/SUN

- L144 instrumentsorldwide

- L195 frmo

- L213 methane. These

- L231 condtions

- L388 substantally

- L428 within within

- Words missing in the sentence L370-L372

We agree with all technical corrections, and we have implemented them in the revised version of the manuscript.  The sentence in Lines 370–372 was indeed missing some

words and has been corrected. The updated text is provided in our response to the major comment from Referee 2 regarding the sentence "The result of section 3.1 is that 'CAMS in situ…'", as discussed in the major comments section of this response.

**References**
Frey et al. 2019, https://doi.org/10.5194/amt-12-1513-2019
Hedelius et al. 2016, https://doi.org/10.5194/amt-9-3527-2016
Wunch et al. 2011, https://doi.org/10.1098/rsta.2010.0240

We thank the reviewer for the reference suggestions. These references are already included in the draft manuscript, and we will also add the additional ones cited in our current response document.